# Genomic landscape of virus-associated cancers

Yoonhee Nam [1,9], Karen Gomez[1,9], Jean-Baptiste Reynier [1,2], Cole Khamnei[1], Michael Aitken[1,3], Vivian Zheng [1], Tenzin Lhakhang[1], Milena Casula [4], Giuseppe Palmieri [4,5], Antonio Cossu[6], Arnold Levine [7], Enrico Tiacci[8] & Raul Rabadan [1,2] ✉

It has been estimated that 15%-20% of human cancers are attributable to infections, mostly by carcinogenic viruses. The incidence varies worldwide, with a majority affecting developing countries. Here, we conduct a comparative analysis of virus-positive and virus-negative tumors in nine cancers linked to five viruses. We observe a higher frequency of virus-positive tumors in males, with notable geographic differences in incidence. Our genomic analysis of 1971 tumors reveals a lower somatic burden, distinct mutation signatures, and driver gene mutations in virus-positive tumors. Compared to virus-negative cases, virus-positive cases have fewer mutations of *TP53, CDKN2*A, and deletions of 9p21.3/*CDKN2*A-*CDKN1A* while exhibiting more mutations in RNA helicases *DDX3X* and *EIF4A1*. Furthermore, an analysis of clinical trials of PD-(L)1 inhibitors suggests an association of virus-positivity with higher treatment response rate, particularly evident in gastric cancer and head and neck squamous cell carcinoma. Both cancer types also show evidence of increased CD8 + T cell infiltration and T cell receptor clonal selection in virus-positive tumors. These results illustrate the epidemiological, genetic, and therapeutic trends across virus-associated malignancies.

An estimated 15-20% of cancers are attributable to infections[1,2], and 8–10% are caused by viruses[3,4]. To date, seven viruses are known to be associated with the development of cancers in humans (oncoviruses): human gammaherpesvirus 4 (HHV-4, also known as Epstein-Barr virus [EBV]), human herpesvirus 8 (HHV-8), human papillomavirus (HPV), human T-cell lymphotropic virus type 1 (HTLV-1), hepatitis B virus (HBV), hepatitis C virus (HCV), and Merkel cell polyomavirus (MCPyV)[5]. In addition, previous studies have observed potential associations of adeno-associated virus 2 (AAV2) with hepatocellular carcinomas (HCC)[6], cytomegalovirus (CMV) with glioblastoma multiforme (GBM)[7],

and a "hit-and-run" mechanism of viral involvement in classical Hodgkin lymphoma (CHL)[8].

While the mechanisms of malignant transformation caused by oncoviruses differ, there are some general patterns that are observed[9]. First, oncoviruses cause a persistent, long-term infection, and tumors develop years after the initial infection. For example, most individuals are infected with EBV by early childhood (in developing countries) or adolescence (in developed countries)[10], but an EBV-associated cancer may not develop until old age. Hepatocellular carcinoma develops 10–30 years after infection with HBV or HCV[11], and cervical cancer

[1]Program for Mathematical Genomics and Department of Systems Biology, Columbia University, New York, NY, USA. [2]Department of Biomedical Informatics, Columbia University, New York, NY, USA. [3]Department of Physics, Columbia University, New York, NY, USA. [4]Unit of Cancer Genetics, Institute of Genetic Biomedical Research (IRGB), National Research Council (CNR), Sassari, Italy. [5]Immuno-Oncology & Targeted Cancer Biotherapies, University of Sassari, Sassari, Italy. [6]Department of Medicine, Surgery and Pharmacy, University of Sassari, Sassari, Italy. [7]Simons Center for Systems Biology, Institute for Advanced Study, Princeton, NJ, USA. [8]Institute of Hematology and Center for Hemato-Oncology Research, Department of Medicine and Surgery, University and Hospital of Perugia, Perugia, Italy. [9]These authors contributed equally: Yoonhee Nam, Karen Gomez. ✉e-mail: rr2579@cumc.columbia.edu

develops 25–30 years following infection with HPV[12]. Second, oncoviruses encode proteins that directly contribute to malignant transformation. In HPV infected cells, the E6 and E7 oncoproteins inhibit the tumor suppressors p53 and Rb, respectively[13]. The vGPCR protein encoded by HHV8 induces angiogenesis and promotes cell transformation[14]. EBV expresses different genes depending on the viral latency program. Type I latency is characterized by the expression of EBV nuclear antigen 1 (EBNA1), essential for viral DNA replication and potentially inhibiting apoptosis, along with EBV-encoded small RNAs (EBERs) and BamHI-A rightward transcript (BART) miRNAs[15]. In contrast, type II latency involves the expression of latent membrane proteins (LMP1 and LMP2), which activate the NF-κB and PI3K/AKT pathways, in addition to the markers of latency I[15]. Latency I is commonly observed in Burkitt lymphoma (BL), while latency II is seen in classical Hodgkin lymphoma (cHL) and nasopharyngeal carcinoma, with EBV-positive gastric cancer being associated with either latency type I or II[15,16]. Third, viral infection is necessary, but not sufficient, for malignant transformation. Many oncoviruses are highly prevalent in the general population: 90–95% of people worldwide are infected with EBV[10], 80% of individuals will acquire an HPV infection by age 45[17], and MCPyV is detected in 80% of individuals in the general population by age 50[18]. However, only a small fraction of those infected with oncoviruses will develop cancer, suggesting additional genetic and/or environmental factors are required. The factors that contribute to the malignant transformation of virus-infected cells remain incompletely understood, but are known to include a combination of environmental, immune, inherited, and somatic components. While many of these components have been described for individual cancer types, relatively little has been reported about the clinical and genetic factors that are common across virus-associated cancers.

In this work, we utilize previously reported datasets in addition to newly acquired data (Kaposi sarcoma) to include all seven known oncoviruses identified in 923 patients across 14 viral-positive cancer types. We identify patterns of common phenotypic characteristics, somatic drivers, and therapeutic responses among these malignancies. This study provides a comprehensive analysis of human cancers that develop in the context of viral infection and key factors related to their pathogenesis.

## Results

### Virus-associated cancer show unique epidemiological trends

Virus-associated cancers are known to follow unique epidemiological patterns compared to non-virus-associated cancers. For example, EBV-positive HL is more frequently observed in cases with mixed cellular histology, males, children, older adults, and in developing countries. In these regions, HL incidence shows an earlier peak, primarily in children under 15 years, and is characterized by a high prevalence of EBV positivity. In contrast, young-adult onset nodular sclerosis (NS) HL, typical of the 15 to 39 age group in industrialized nations, is usually EBV-negative[19,20]. BL has been traditionally classified into two clinical variants, i.e., endemic BL (eBL) and sporadic BL (sBL), that present different demographic (eBL tend to be younger), geographic (most eBL are from equatorial Africa), virus status (nearly all eBL are EBV +, while a small fraction of sBL are) and somatic mutation characteristics[21] (this should be taken into account in BL comparisons along the manuscript).

In order to illustrate other common demographic characteristics of virus-associated malignancies, we analyzed data from the Global Cancer Observatory (GLOBOCAN 2020)[22] and published incidence rates in 48 studies of 11 cancer types linked to 5 viruses and 13 non-virus-associated cancers[23–104]. First, we compared the number of viral cancers in males versus females (M/F) reported in select published studies[23–104] (Fig. 1A and Supplementary Data 1). We found that the M/F ratio was greater overall in virus-associated cancers compared to nonviral cancers ($p = 2.2e-16$, Fisher's exact test). Among studies that

reported M/F ratio for virus-positive and virus-negative tumors specifically, virus-positive cases tended to have a greater M/F ratio than virus-negative cases ($p = 5.23e-8$, Fig. 1B and Supplementary Data 2). This trend was consistent in gastric cancer (GC, $p = 1.04e-10$) and HL ($p = 0.015$), both of which have been reported previously[105–107]. In contrast, no difference in the M/F ratio and viral status was observed for BL, despite higher M/F incidence ratios of 2:1 to 4:1 being reported in both eBL and sBL[108]. A lower M/F ratio was observed in MCPyV-positive Merkel cell carcinoma (MCC) compared to virus-negative MCC, as noted previously[4]. Digital papillary adenocarcinoma, which has been recently associated to HPV42, is reported to be more frequent in males compared to females at a ratio of 4:1[109].

To examine how the incidence of virus-associated cancers differs by geographic location, we compared the age-standardized incidence rates (ASR) of 4 cancers in 185 countries reported in GLOBOCAN 2020. In HL, EBV-positive cases occur most frequently in North Africa, the Middle East, and South America, with the lowest incidence occurring in East Asia (Fig. 1C, D). In contrast, most cases of EBV-positive nasopharyngeal carcinoma (NPC) occur in China and Southeast Asia (Fig. 1E, F). Similarly, Kaposi sarcoma and cervical cancer (nearly all of which are virus-positive) show disparities in incidence by geographic location (Supplementary Fig. S1). These results illustrate that the locations of global hot spots of virus-positive tumor incidence vary by virus and even among cancers associated with the same virus. These disparities reflect differences in risk factors for virus-positive tumors among human populations, both genetic (e.g., inherited susceptibility polymorphisms[110,111]) and environmental (e.g., oncovirus prevalence[112], and lifestyle factors such as smoking or diet, which affect overall cancer risk[113]).

### Virus-positive tumors have fewer somatic mutations than virus-negative tumors

We aggregated somatic mutation data from 1971 tumors in published studies of 9 cancers subjected to whole-exome sequencing/WES (plasmablastic lymphoma [PBL][114] [$n = 15$], cHL[115,116] [$n = 69$], cervical cancer [CC][117] [$n = 178$], hepatocellular carcinomas [HCC][118] [$n = 196$], GC[105] [$n = 440$], and head and neck squamous cell carcinoma [HNSCC][119] [$n = 487$]), targeted DNA sequencing (BL[21] [$n = 29$], PBL[114,120] [$n = 36$], primary central nervous system lymphoma [PCNSL][121] [$n = 58$], MCC[122] [$n = 71$], and cHL[123] [$n = 293$]), and/or whole genome sequencing/WGS (BL[21] [$n = 91$] and cHL[115] [$n = 24$]) (Supplementary Data 3). In general, virus-positive tumors had a lower count of nonsynonymous mutations than virus-negative tumors, as reported in individual cancer types[116,120,124–127] (Figs. 2A, 2B, Supplementary Data 3 and Supplementary Data 5), including in particular PCNSL[125] (Wilcoxon test $p = 1.4e-7$), cHL (targeted, $p = 2.4e-6$; WES, $p = 0.032$; WGS, $p = 0.045$), PBL[120] ($p = 0.044$), HNSCC[126] ($p = 6.5e-6$) and, as a trend, GC[127] ($p = 0.086$), CC ($p = 0.17$), and MCC[124] ($p = 0.072$) (Fig. 2A).

Conversely, in HCC and BL, virus-positive cases had a greater mutation load than virus-negative cases ($p = 0.026$ and $p = 5.4e-4$, respectively), but the count of nonsynonymous mutations in genes previously described as BL drivers[21] still trended towards lower in virus-positive BL cases ($p = 0.12$, Fig. 2B and Supplementary Fig. S2 and Supplementary Data 5), consistent with that report[21]. Furthermore, when restricting the analysis to driver genes (Supplementary Data 4), the lower mutation count in virus-positive cases became statistically significant in MCC ($p = 0.015$, Supplementary Fig. S2), while virus-positive GC and HCC had more driver gene mutations than their virus-negative counterparts (GC: $p = 0.036$; HCC: $p = 8.8e-4$). Overall, the total mutation count and/or driver mutation count was lower in virus-positive compared to virus-negative tumors in most cancers studied (Fig. 2B).

In addition, we observed that several cancer subtypes and virus strains differed in mutation load (Supplementary Data 6). The NS subtype, the most common subtype of cHL in our datasets (targeted,

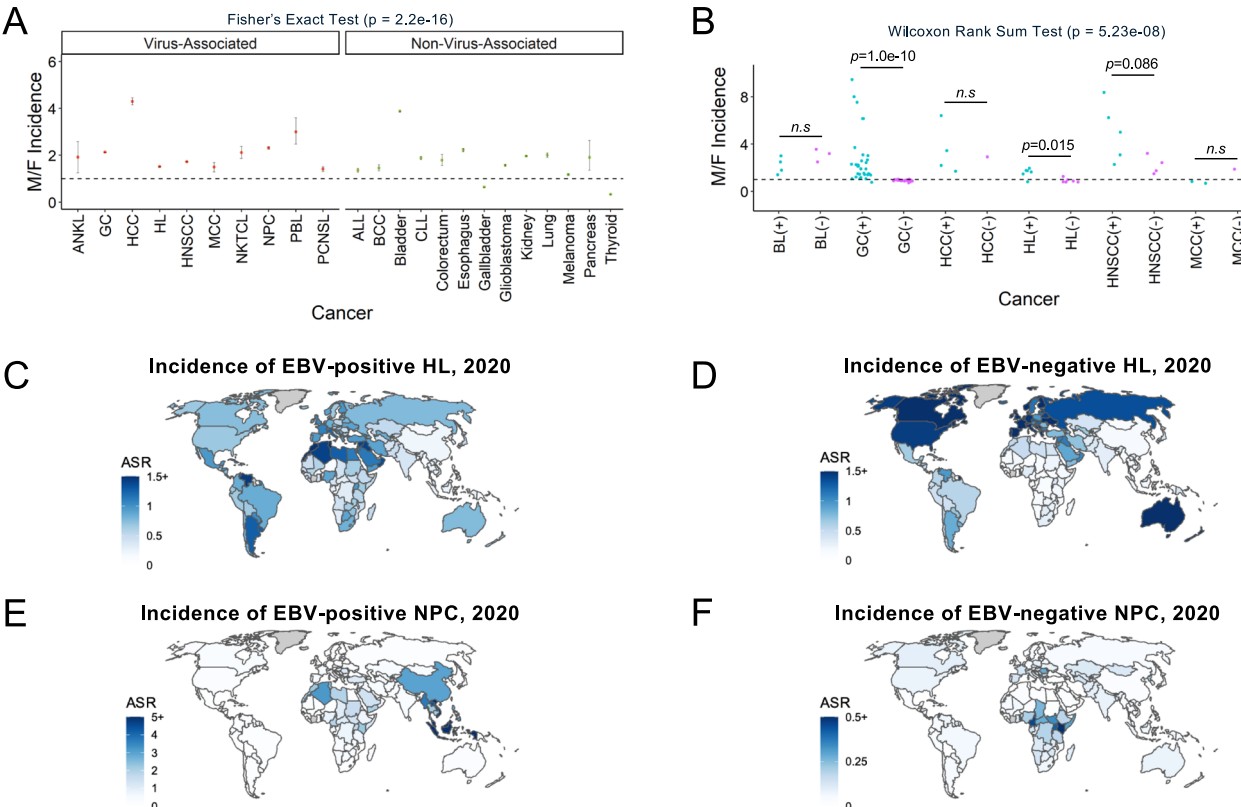

**Fig. 1 | Epidemiological trends of virus-associated cancers. A** Incidence ratios of virus-associated and non-virus-associated cancers analyzed using two-sided Fisher's Exact Test. Data are presented as point estimates (M/F incidence ratios) with error bars indicating 95% confidence intervals. ANKL (92/48), GC (659302/309048), HCC (16091/3744), HL (60913/40219), HNSCC (266342/154358), MCC (521/352), NKTCL (787/370), NPC (29769/12855), PBL (443/148), PCNSL (1814/1286), ALL (3239/2380), BCC (121701/51529), Bladder (487885/125906), CLL (9908/5249), Colorectum (622/348), Esophagus (13067/5857), Gallbladder (47920/74542), Glioblastoma (31071/19801), Kidney (287986/146433), Lung (2861/1416), Melanoma (21469/18141), Pancreas (105/55), Thyroid (207549/613624). **B** Virus-positive and virus-negative tumors in virus-associated cancers in males compared to females (M/F) reported in selected published studies, analyzed using two-sided Wilcoxon Rank Sum Test. Each point corresponds to an incidence ratio reported in a published study in Supplementary Data 2. ANKL, aggressive NK-cell leukemia; GC, gastric cancer; HCC, hepatocellular carcinoma; HL, Hodgkin lymphoma; HNSCC, head and neck squamous cell carcinoma; MCC, Merkel cell carcinoma; NKTCL, Natural killer/T-cell lymphoma; NPC, nasopharyngeal carcinoma; PBL, plasmablastic lymphoma; PCNSL, primary central nervous system lymphoma; ALL, acute lymphoblastic leukemia; BCC, basal cell carcinoma; CLL, chronic lymphocytic leukemia; BL, Burkitt lymphoma. **C**, **D** Estimated incidence rates of EBV-positive HL (**C**) and EBV-negative HL (**D**) by country. **E**, **F** Estimated incidence rates of EBV-positive NPC (**E**) and EBV-negative NPC (**F**) by country. Map data from Natural Earth (https://www.naturalearthdata.com/, public domain), produced by rnaturalearth R package[194]. ASR, age-standardized rate. Source data are provided as a Source Data file.

84.6%, 204/241; WES, 73.4%, 47/64), exhibited the highest mutation load relative to other subtypes in both targeted and WES cohorts ($p = 1.6E-06$ and 0.010, respectively). However, after adjusting for subtypes, sex, and age, only EBV status remained significantly associated with a lower mutation load in the targeted and WES cohorts, consistent with a previous report[116] (coefficient = −0.44, −2.1; $p = 0.009, 0.001$). As expected[21], eBL had a higher total mutation load compared to sBL, reflecting a similar trend with EBV status ($p = 0.00029$). Since EBV-positivity was highly associated with eBL ($p = 7.68e-12$), when both subtype and EBV status were considered simultaneously, neither variable showed a significant independent association with total mutation load.

In CC, there was no significant difference in mutation load between squamous carcinoma and adenocarcinoma subtypes. However, age at diagnosis positively correlated with mutation load (Spearman's $\rho = 0.31$, $p = 4.1E-05$, Supplementary Data 6), consistent with a previous study[128]. Notably, multiple HPV strains in CC differed significantly in mutation load, with HPV31 exhibiting the highest levels (Fold change = 7.07, $p = 0.02$, Supplementary Fig. S3 and Supplementary Data 6). To account for age, HPV strains, viral status, on mutation load, we used generalized linear model regression model (GLM) and found that HPV31 and virus-positive status remained significantly

associated with elevated mutation load (coefficient = 1.81 and 1.09, $p = 0.013$ and $< 0.001$, respectively).

In HNSCC, females exhibited higher driver gene mutation counts than males ($p = 0.0058$), and age correlated positively with both total and driver mutation counts (Spearman's $\rho = 0.19$ and 0.27, $p = 2.4E-05$ and 1.2E-09, respectively; Supplementary Data 6), consistent with previous findings[129]. Unlike CC, there were no significant differences in mutation load between HPV strains in HNSCC (Supplementary Fig. S4), as reported in a prior study[130].

In HCC, no difference in mutation load was evident between HBV and HCV related HCC. Neither Activated B-cell (ABC) nor Germinal Center B-cell (GCB) subtypes of PCNSL differed in their mutation loads.

**Virus-associated cancers display unique mutation signatures**

To detect and quantify the relative contribution of COSMIC mutation signatures[131] within the virus-associated cancers, we applied SigProfilerExtractor[132] to extract de novo mutational signatures and decompose them into COSMIC signatures across 7 available cancers. Virus-positive tumors exhibited different activities of mutation signatures compared to virus-negative tumors of the same cancer type (Fig. 3, S5 and Supplementary Data 7). UV light is known to be the major etiological agent of MCC tumors in the absence of viral

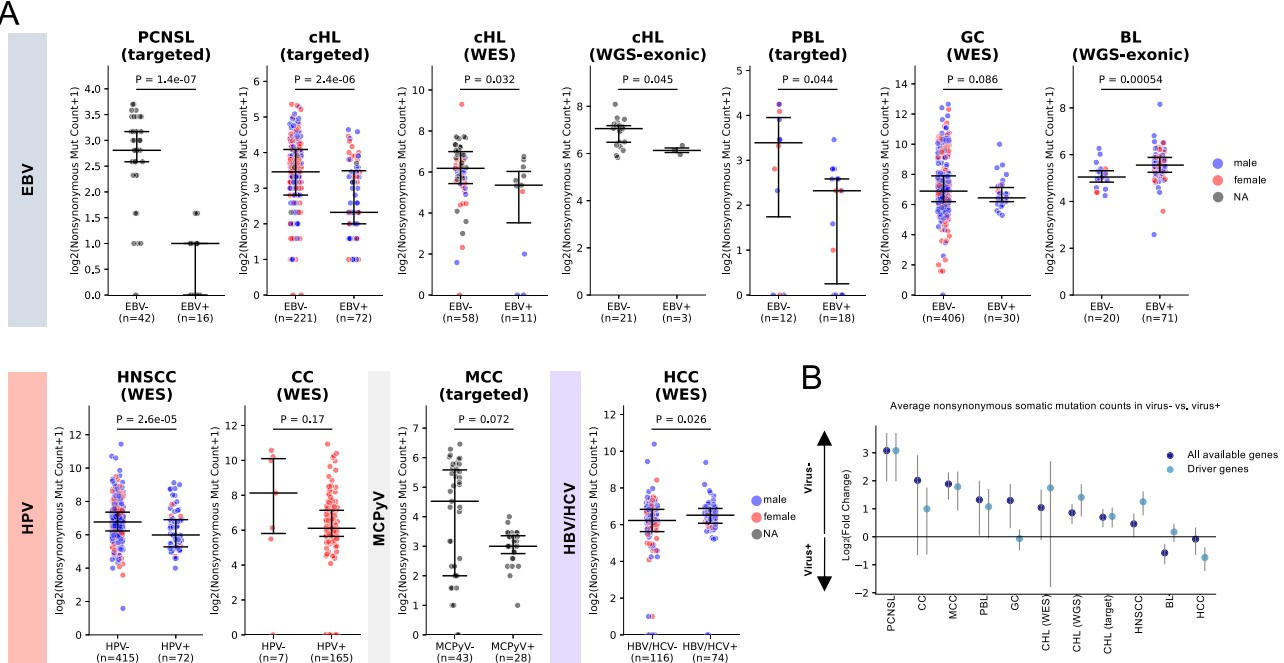

**Fig. 2 | Mutation burden of virus-positive and virus-negative tumors in 9 cancers. A** Counts of somatic nonsynonymous mutations in virus-positive and virus-negative tumors in the same cancers. Data are presented as median values with interquartile range (25th–75th percentile). Actual median values for virus-negative and positive samples in the shown cancer types are PCNSL: 6 and 1, cHL (targeted): 10 and 4, cHL (WES): 126.5 and 65, cHL (WGS): 132 and 69, BL: 32 and 46, PBL: 9.5 and 4, GC: 117.5 and 86, CC: 279 and 68, HNSCC: 108 and 62.5, MCC: 22 and 7, HCC: 74 and 90.5. *P*-values are calculated by a two-sided Wilcoxon rank-sum test. **B** Log$_2$(fold change) of average number of somatic nonsynonymous mutations (all genes: dark blue, driver genes: light blue) in virus-negative tumors compared to virus-positive tumors. PCNSL (*n* = 58), CC (*n* = 172), MCC (*n* = 71), GC (*n* = 436), PBL (*n* = 51), cHL (target, *n* = 293; WES, *n* = 69; WGS, *n* = 24), HNSCC (*n* = 487), BL (*n* = 91 for all genes; 68 EBV-positive eBL, 6 EBV-negative eBL, 3 EBV-positive sBL, 14 EBV-negative sBL), BL (*n* = 120 for driver genes) and HCC (*n* = 190). GC, gastric cancer; HCC, hepatocellular carcinoma; cHL, classical hodgkin lymphoma; HNSCC, head and neck squamous cell carcinoma; MCC, Merkel cell carcinoma; PBL, plasmablastic lymphoma; PCNSL, primary central nervous system lymphoma; CC, cervical cancer; BL, Burkitt lymphoma. Data are presented as log$_2$(fold change) with error bars indicating 95% confidence intervals. Source data are provided as a Source Data file.

infection. Accordingly, the relative contribution of mutations attributed to SBS7a/7b/UV-light was lower in MCPyV-positive compared to MCPyV-negative, as reported previously[122] (Fig. 3A, B and Supplementary Data 7).

In GC, the proportion of mutations associated with SBS15/mismatch-repair (MMR) deficiency was higher in EBV-negative than EBV-positive cases (Fig. 3C, D and Supplementary Data 7). Microsatellite instability (MSI), as assessed by standard methods, is a defining characteristic of a GC subtype that is exclusively EBV-negative and comprised 73/406 (18%) of EBV-negative patients in the TCGA cohort. Accordingly, the relative contribution of mutations attributed to SBS15 and SBS20/MMR-deficiency were higher in the conventionally defined MSI subtype compared to other EBV-negative cases (Supplementary Fig.S6 and Supplementary Data 8). This suggests that the difference in MMR signatures between EBV-positive and EBV-negative GC are largely driven by the MSI subtype.

In HNSCC, HPV-positive HNSCC had the higher relative contribution of SBS2/APOBEC mutations (Fig. 3E, F and Supplementary Data 7), as reported in a previous study[2]. This is consistent with the hypothesis that HPV oncoproteins may increase APOBEC3A and APOBEC3B expression and mutagenic activity[133,134]. In a multivariate analysis of HPV-positive HNSCC, which accounted for sex, HPV strain, age, and total mutation count, HPV33-associated HNSCCs showed significantly lower SBS2/APOBEC mutation counts, while total mutation count was positively associated with SBS2/APOBEC mutation counts, both findings consistent with previous reports[130,135] (coefficient = −1.37 and 0.012, *p* = 0.003 and < 0.001, respectively; Supplementary Data 9, Supplementary Fig. S7).

The mutation counts associated with the SBS5/clock-like signature were higher in HPV-negative cervical cancer (CC) compared to HPV-positive CC (*p* = 0.013, Supplementary Data 7), consistent with our observation that total mutation counts are positively correlated with age and tend to be higher in HPV-negative cases. Notably, the levels of SBS2/13/APOBEC-associated signatures varied between HPV strains, with HPV31 significantly exhibiting the highest mutation counts (Supplementary Fig. S8 and Supplementary Data 8). A similar trend observed in HNSCC was also apparent in CC, where the HPV33 strain exhibited lower APOBEC signature levels compared to HPV16, although this difference was not statistically significant in CC. These strain-specific differences aligned with our findings on total mutation load. Importantly, these differences appear to be primarily driven by variations in total mutation load, as HPV strain-specific effects were no longer significant in a multivariable model that accounted for virus strain, virus status, total nonsynonymous mutation load, and age. In this model, virus-positive status and total nonsynonymous mutation load were positively associated with APOBEC signature mutations (Supplementary Data 10), consistent with previous findings[117]. In BL exomes[21] (Supplementary Fig. S5B), a de novo mutational signature, SBS96B, was identified. Although SBS96B did not meet the cosine similarity threshold for decomposition into known COSMIC signatures, potentially related COSMIC signatures included SBS46, SBS5, SBS17, and SBS15. This signature showed a higher proportion in EBV-positive BL samples compared to EBV-negative BL samples (Supplementary Data 7), consistent with the original genome-wide mutation signature analysis on SBS5, SBS17, and SBS15 in the same cohort[21]. However, in partial contrast to the original analysis, we

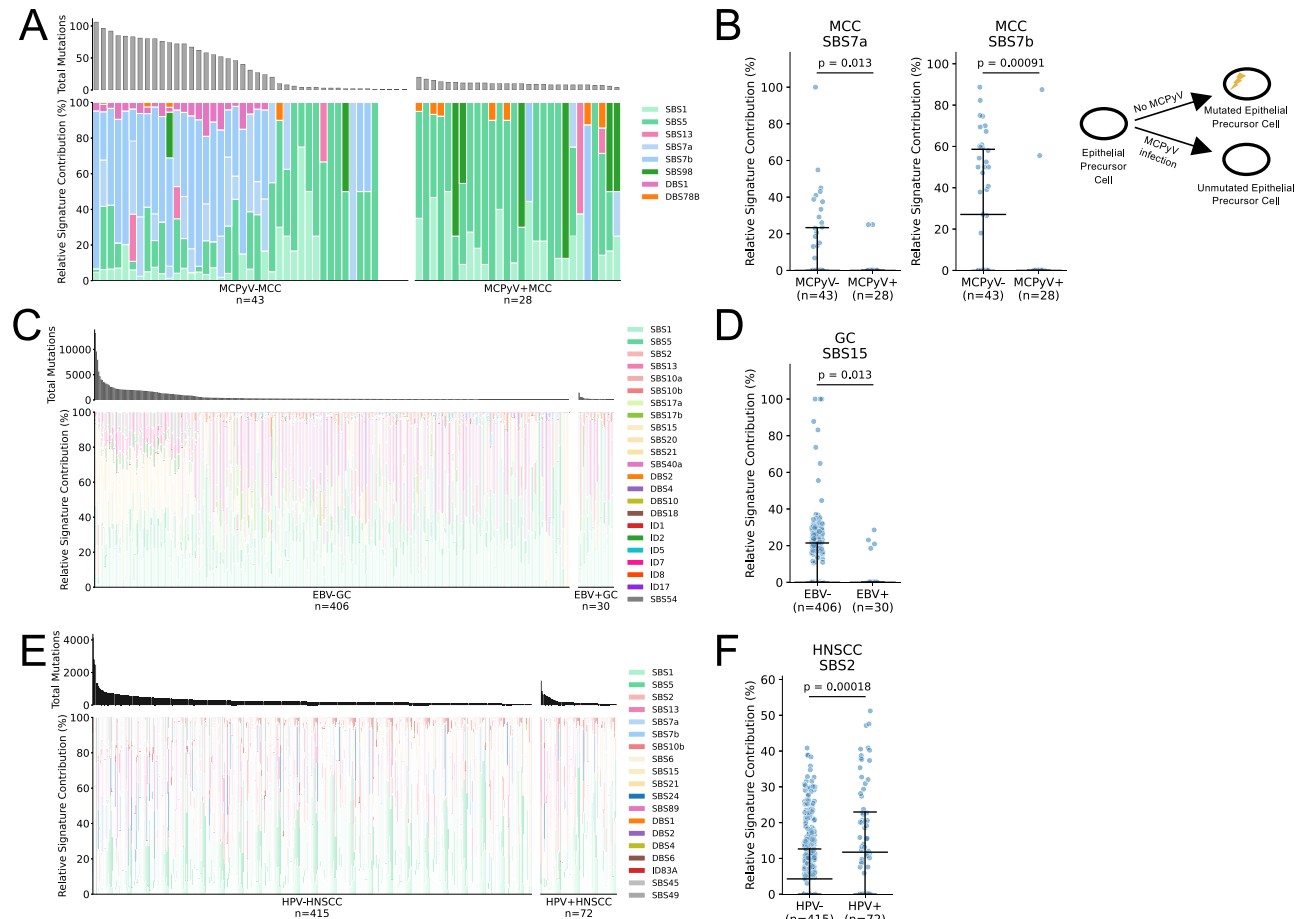

**Fig. 3 | Mutation signatures in virus-associated cancers. A**, **B** MCC ($n = 71$), (**C**, **D**) GC ($n = 436$), (**E**, **F**) HNSCC ($n = 487$). Total mutations (top bar plot) and proportion of mutations associated with each signature (bottom stacked plot) in virus-positive compared to virus-negative cases. Signatures identified from SigProfilerExtractor are shown. Signature names with SBS96, DBS78, or ID83 followed by a capital letter are de novo mutational signatures identified from the cancer cohort, which could not be decomposed into known COSMIC (v.3.4) signatures. The rest are decomposed COMSIC signatures. Schematic representation of the effect of the absence of processes behind key mutation signatures in virus-associated cases is shown in (**B**) MCC (MCPyV-positive). MCC, Merkel cell carcinoma; GC, gastric cancer; HNSCC, head and neck squamous cell carcinoma. **B**, **D**, **F** Data are presented as median values with interquartile range (25th–75th percentile). *P* values are calculated by a two-sided Wilcoxon rank-sum test. Source data are provided as a Source Data file.

did not detect the SBS9/polη signature, which is associated with non-canonical AID activity and was previously identified in BL genomes[21].

In HCC, there was no difference in absolute count of mutations attributed to mutation signatures (Supplementary Fig. S5C), consistent with the similar mutation burden in virus-positive and -negative HCC overall. However, HCC tumors positive for HBV had a greater proportion of mutations due to SBS24/aflatoxin, an environmental carcinogen known to predispose to HBV-mediated cirrhosis, as observed in previous studies[136–138] ($p = 0.010$) (Supplementary Data 8). There was no significant difference in the absolute counts or proportions of signatures in EBV-positive versus EBV-negative PBL, likely due to the limited number ($n = 1$) of EBV-negative cases (Supplementary Fig. S5D). Overall, these results illustrate that the signatures of somatic mutation processes vary depending on infection status for each cancer, highlighting differing selective pressures on the cancer genomes in the presence or absence of viral oncoproteins.

To identify differences in structural variant (SV) and copy number (CN) signatures between virus-positive and virus-negative tumors, we conducted a signature analysis using SigProfilerExtractor[132], following the same methodology as in SBS analysis. This analysis was performed on WGS samples from HCC, HNSCC, CC, and GC obtained from PCAWG, as well as BL and cHL (Supplementary Figs. S9, 10 and

Supplementary Data 11). Among SV signatures, no significant differences were observed between virus-positive and virus-negative tumors, except for an enrichment of the SV2 signature, which consists of non-clustered translocations (COSMIC v3.4), in HPV-positive HNSCC ($q = 0.0025$, Supplementary Fig. S9G). In the CN signature analysis, several de novo signatures that could not be decomposed into known COSMIC CN signatures were identified as differing in multiple cancer types (Supplementary Fig. S10).

Furthermore, we compared differences in chromosomal instability (CIN) signatures between virus-positive and virus-negative tumors using CIN signature data from a recent study on TCGA cohorts, including HCC, HNSCC, CC, and GC (Supplementary Data 3)[139]. Notably, the CX1 signature, which involves whole-arm or whole-chromosome changes and is associated with chromosome mis-segregation due to defective mitosis and/or telomere dysfunction[139], showed a significantly higher relative contribution in virus-positive cases compared to virus-negative cases in HNSCC, GC, and HCC ($p = 7.9$e-6, 1.3e-4, 0.0032, respectively; Supplementary Fig. S11, 12 and Supplementary Data 12, 13). Although CC exhibited a similar trend, the difference was not statistically significant, likely due to the small number of HPV-negative cases ($n = 4$, Supplementary Fig. S12). This observation is consistent with studies showing that HPV's E6 and E7

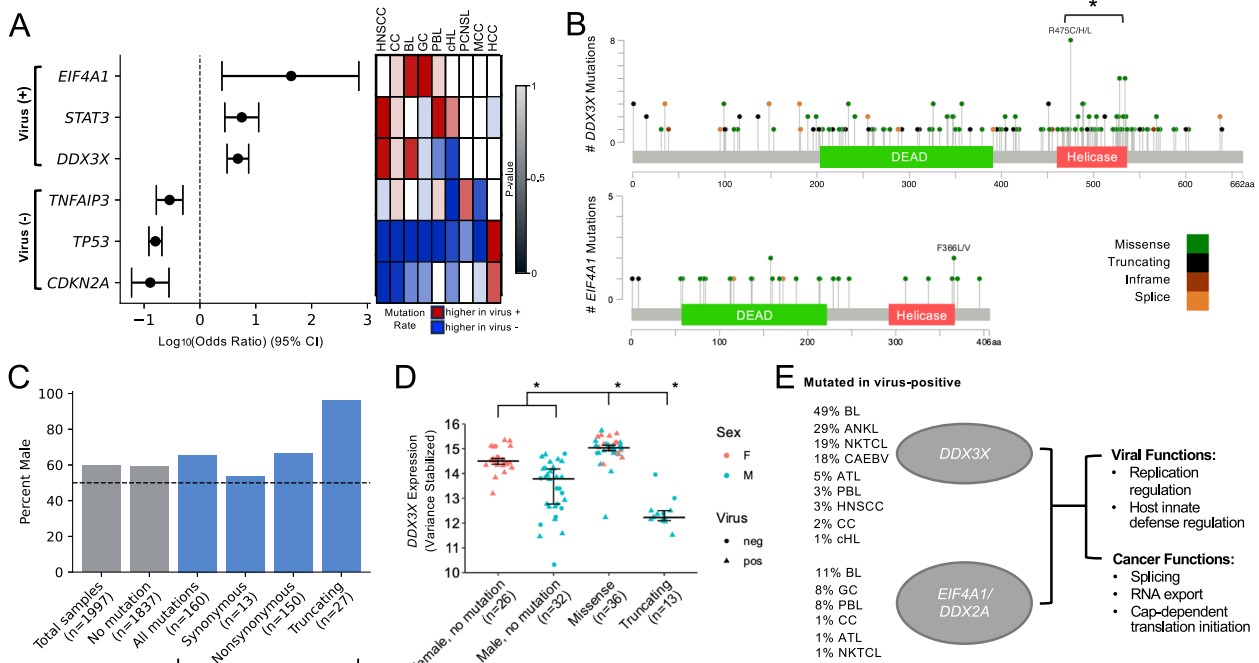

**Fig. 4 | Somatic mutations in *EIF4A1* and *DDX3X*, both RNA helicases of the DEAD (Asp-Glu-Ala-Asp) box protein family, are recurrent genetic lesions associated with virus-positive status. A** Combined $\log_{10}$(odds ratio) of mutation in genes associated with virus-positive (top) and virus-negative (bottom) status ($q < 0.005$) from pooled data of 1971 tumors across 9 virus-associated cancers. Data are presented as $\log_{10}$(odds ratio) values with error bars indicating 95% confidence intervals. The heatmap on the right displays the cancer cohorts included in the pooled data for the calculation of each gene, with colors representing mutation rate trends in each cohort (red: higher in virus-positive; blue: higher in virus-negative) and shades indicating the two-sided Fisher's exact test *p*-value. HNSCC, head and neck squamous cell carcinoma; CC, cervical cancer; BL, Burkitt lymphoma; GC,

gastric cancer; PBL, plasmablastic lymphoma; cHL, classical Hodgkin lymphoma; PCNSL, primary central nervous system lymphoma; MCC, Merkel cell carcinoma; HCC, hepatocellular carcinoma. **B** Mutations in *DDX3X* and *EIF4A1* in 2488 tumors. * $p < 0.05$, two-sided binomial test. **C** Fraction of patients that are male by *DDX3X* mutation status. **D** *DDX3X* expression by *DDX3X* mutation status and sex in Burkitt lymphoma ($n = 117$). * $p < 0.05$, two-sided MWU test. Data are presented as median values with interquartile range (25th–75th percentile). **E** Frequencies of mutation of *DDX3X* and *EIF4A1* in virus-positive tumors overall and summary of key biological functions. ANKL, aggressive NK-cell leukemia; NKTCL, Natural killer/T-cell lymphoma; CAEBV, chronic active Epstein-Barr virus disease; ATL, Adult T-cell leukemia/lymphoma. Source data are provided as a Source Data file.

proteins disrupt normal cell cycle regulation and mitotic progression, leading to chromosomal abnormalities[140].

## Virus-positive tumors harbor frequent mutations in RNA helicases *DDX3X* and *EIF4A1*

To identify genomic loci preferentially mutated in virus-positive tumors, we compared the rate of nonsynonymous mutations and copy number aberrations in the pooled cohort of 602 virus-positive and 1369 virus-negative tumors from 9 cancers (1,971 total cases; Supplementary Data 3 and Supplementary Fig. S13; see Methods). We found that three genes had significantly elevated odds of mutation in virus-positive tumors: *EIF4A1* (OR = 43.07, 95% CI = 2.48-701.39, $q = 1.65e-3$, Fisher's exact test, BH corrected), *STAT3* (OR = 5.61, 95% CI = 2.79-11.28, $q = 4.80e-7$), and *DDX3X* (OR = 4.78, 95% CI = 3.06-7.46, $q = 5.57e-9$) (Fig. 4A, S14, and Supplementary Data 14).

When looking at individual cancer types, *EIF4A1* mutations were significantly more frequent in EBV-positive GC compared to EBV-negative GC ($p = 5.95e-3$). This trend was consistently observed across all other cancer types analyzed, including BL, PBL, and CC (Supplementary Fig. S14 and Supplementary Data 14). Similarly, *DDX3X*, an RNA helicase in the same family as *EIF4A1*, was more frequently mutated in EBV-positive HNSCC ($p = 0.044$), as previously reported[141], and a similar trend was observed in BL and CC, though not statistically significant (Supplementary Fig. S14 and Supplementary Data 14). Specifically for BL, we were able to increase the statistical power by extending the analysis to incorporate data on *DDX3X* from six different genomic studies of BL[21,142–146] including eBL ($n = 144$) and sBL ($n = 177$) cases, of whom 142 known to be EBV-positive and 119 EBV-negative

(Supplementary Data 15). Notably, we found that *DDX3X* mutation was strongly (though not exclusively) associated with EBV-positive status ($p = 3.78e-6$), which has been discussed in previous studies but without statistical significance[21,142–146], and endemic subtype ($p = 4.25e-7$, Supplementary Data 14, Supplementary Data 16). No differences were observed between EBV type 1 and type 2 (Supplementary Data 16). *STAT3* mutations were significantly more frequent in virus-positive PBL ($p = 0.050$), as noted in previous studies[147], and in virus-positive HNSCC ($p = 0.0052$), with cHL and CC showing similar trends compared to their virus-negative cases (Supplementary Data 14).

Furthermore, an analysis of recurrent copy number aberrations in 6 available cancers (HCC, PBL, CC, BL, HNSCC, GC) revealed 14q32.33 loss (OR = 2.79, 95% CI = 2.14-3.65, $q = 2.44e-13$) and 11q23.3 loss (OR = 2.55, 95% CI = 1.98-3.30, $q = 2.57e-12$) more frequent in virus-positive tumors (Supplementary Fig. S16). A recent study reported that EBV's protein EBNA1 binds to a specific region on human chromosome 11q23.3, inducing breakage and structural variations[148]. However, the loss of 11q23.3 was enriched in virus-positive HNSCC, CC, and HCC, all of which associated with non-EBV virus, while the EBV-associated BL, GC and PBL showed the opposite trend (Supplementary Fig. S16B).

Meanwhile, virus-negative tumors had higher odds of *TP53* mutations (OR = 8.20, 95% CI = 4.79–8.20, $q = 4.60e-52$), which were significant and known for most cancer types individually, including HNSCC[2], GC[105], MCC[18], BL[21], PBL[120], among others. However, HCC exhibited the opposite trend, with *TP53* mutations being significantly associated with HBV/HCV-positive HCC, as previously reported[149]. Mutations in *CDKN2A* were also more frequently observed in virus-negative tumors (OR = 7.75, 95% CI = 3.59–16.76, $q = 9.42e-12$), and

were significant in HNSCC individually, as noted in a prior study[2]. Consistently, loss of 9p21.3/*CDKN2A,CDKN1A* were observed in virus-negative cancers (OR = 4.96, 95% CI = 3.65–6.73, $q$ = 1.77e-25) (Figure S16C). Lastly, *TNFAIP3* mutations were enriched in virus-negative cases (OR = 3.49, 95% CI = 2.02–6.04, $q$ = 3.72e-7), and were significant in cHL individually, as reported previously[150] (Fig. 4A, S15, and Supplementary Data 14).

*EIF4A1* and *DDX3X* are RNA helicases of the DEAD (Asp-Glu-Ala-Asp) box protein family, which are known to play a role in splicing, RNA export, and cap-dependent translation initiation[151]. To further explore the role of these genes in virus-associated cancers, we expanded the analysis from the 1971 initial cases to include 201 cases from the extended BL cohort mentioned above (Supplementary Data 15) and also 316 cases from tumors that are virus-associated in almost 100% of instances: Kaposi sarcoma (KS, 10 newly sequenced cases), aggressive NK cell lymphoma (ANKL)[152] ($n$ = 14), adult T cell leukemia/lymphoma (ATL)[153] ($n$ = 81), extranodal NK T-cell lymphomas (NKTCL)[154] ($n$ = 100), and NPC[155] ($n$ = 111) (Supplementary Data 2), for a total of 2,488 cases. Overall, we identified 29 *EIF4A1* nonsynonymous mutations (25 in virus-positive, 4 in virus-negative cases) and 150 *DDX3X* nonsynonymous mutations (93 in virus-positive, 38 in virus-negative, 11 in both, 8 in virus status unknown cases) (Supplementary Data 17). Focusing on the more numerous *DDX3X* mutations, among virus-positive tumor samples they were detected in 49% (67/136) BL, 29% (4/14) ANKL, 19% (19/100) NKTCL, 1.2% (1/86) cHL, 5% (4/81) ATL, 3% (1/30) PBL, 3% (2/60) HNSCC, and 2% (3/144) of CC. *DDX3X* mutations occurred in the helicase domain more frequent than expected by chance ($p$ = 1.97e-16 overall; $p$ = 4.56e-9 for virus-positive only and $p$ = 2.90e-6 for virus-negative only cases, binomial test) (Fig. 4B and Supplementary Data 17). The *DDX3X* gene is located on the X chromosome and was previously reported to escape X inactivation in females[156]. Both truncating events and at least some missense mutations in *DDX3X* have been previously described as causing functional loss of protein activity[146]. While only 60% of patients were male (1193/1997), truncating *DDX3X* mutations occurred almost exclusively in males (26/27, 96%; 14/14 virus-positive, 11/12 virus-negative, 1 male with unknown virus status) (Fig. 4C), consistent with a previous study in BL[157]. As most of *DDX3X*-mutated cases were from the BL cohort, we focused on this disease to evaluate the relationship between mutation status and *DDX3X* expression, using published RNA-sequencing data[21]. We observed a significantly lower *DDX3X* expression in unmutated male versus female cases (median = 13.79 and 14.50, $p$ = 2.4e-6, MWU test), consistent with escape from X inactivation[156]. Cases with *DDX3X* missense mutations had an elevated *DDX3X* expression irrespective of sex compared to unmutated cases (median = 15.04 and 14.24, $p$ = 1.2e-9), potentially suggesting that overexpression of missense mutants may favor their ability to decrease *DDX3X* function, while cases with truncating mutations (9 EBV-positive; 4 EBV-negative) had a lower expression (median = 12.23 and 14.24, $p$ = 1.79e-5) (Fig. 4D), consistent with them being loss-of-function events. Similarly, in the TCGA study of HNSCC, *DDX3X* expression was lower in unmutated male versus female cases (median = 12.58 and 12.99, $p$ < 2.2e-16), and was also lower in cases (3 HPV + ; 2 HPV-) with truncating mutations compared to unmutated cases (median = 10.37 and 12.71, $p$ = 1.5e-4) (Supplementary Fig. S17). Together, these results suggest that mutations in *DDX3X* and *EIF4A1* may play a role in virus-positive tumors in various types of cancer (Fig. 4E).

### Virus-associated cancers exhibit more frequent responses to immunotherapy

*PD-L1* overexpression has been linked to better overall survival in patients treated with immune checkpoint inhibitors (ICI) in several tumor types, including GC[158], HNSCC[159], and MCC[160]. *PD-L1* expression has been associated with infection by oncoviruses including EBV[161], HPV[162], HBV[163], and MCPyV[160]. To determine whether virus positivity

might be a useful marker for response to ICI therapy, we evaluated the correlation of viral status with response to ICI therapy with anti-PD(L)1 in 32 cohorts reported on ClinicalTrials.gov that had available therapy response and virus infection status data, representing four virus-linked cancers (Supplementary Data 18 and Supplementary Data 19).

Virus positivity was significantly associated with ICI therapy response in GC (OR = 2.27, 95% CI = 1.17-4.29, $p$ = 0.011, Fisher's exact test) and HNSCC (OR = 1.85, 95% CI = 1.24-2.74, $p$ = 0.0018), with a similar trend in HCC (OR = 1.30, 95% CI = 0.95-1.78, $p$ = 0.098), but not MCC ($p$ = 0.85) (Fig. 5A). The same cancer types displayed significant association between PD-L1 expression and ICI therapy response, including GC ($p$ = 1.55e-6), HCC ($p$ = 0.0068), and HNSCC ($p$ = 0.034), and a trend was observed also in MCC ($p$ = 0.15). Higher tumor mutational burden (TMB) was associated with ICI therapy response in GC ($p$ = 1.55e-6), exhibiting a similar pattern in HNSCC ($p$ = 0.14), the only two cancer types with such data available (Supplementary Data 19).To determine the relationship between virus positivity and other prognostic markers of ICI therapy response, we compared the expression of PD-L1 [CD274] in TCGA's studies of GC[105], HCC[118], and HNSCC[119]. CD274 expression was higher in virus-positive GC compared to virus-negative GC (median = 6.74 and 5.21, $p$ = 1.6e-07), but this was not the case in HCC or HNSCC (Fig. 5B). Using results of CIBERSORT[164,165] deconvolution of the same TCGA samples, we found evidence of higher CD8 + T cell infiltration in EBV-positive versus EBV-negative GC (median = 0.19 and 0.09, $p$ = 7.4e-09), and in HPV-positive versus HPV-negative HNSCC (median = 0.13 and 0.07, $p$ = 4.3e-10), but not between virus-positive and virus-negative HCC (Fig. 5C). These results were replicated in all other deconvolution approaches tested[166–169] (Supplementary Fig. S18). Similarly, analysis of TRUST4[170,171]-extracted T cell receptors showed higher T cell β receptor clonal selection, measured by counts per clonotypes per thousand reads (CPK), in both virus-positive GC (median = 185.16 and 310.73, $p$ = 7.8e-10) and HNSCC (median = 345.72 and 444.44, $p$ = 0.017) but not in HCC (Fig. 5D). We observed similar trends for the α chain, as well as with the metric of clonality (Supplementary Fig. S19). The effect of virus positivity appears independent from TMB, as virus-positive GC and HNSCC have fewer mutations than virus-negative GC and HNSCC, respectively (Fig. 2A), while in HCC, virus-positive and negative cases have similar mutation load ($p$ = 0.12). These results suggest that virus-positive status may be a positive prognostic marker for patients undergoing ICI therapy for GC and HNSCC, which may be correlated with higher CD8 + T cell infiltration, T cell receptor clonal selection, and T cell exhaustion (for GC only).

## Discussion

This study builds upon and extends prior research while providing insights into the epidemiological, somatic, and immune components commonly implicated in the pathogenesis of oncovirus-associated cancers. The observed higher male incidence of virus-associated cancers confirms earlier epidemiological findings in certain malignancies, such as EBV-positive HL and GC[105–107]. Likewise, the finding of a reduced mutation burden in virus-positive tumors relative to virus-negative counterparts is consistent with previous reports in individual tumor types[116,120,124–127], including HNSCC[126] and PCNSL[125]. We also identified frequent RNA helicase mutations (*DDX3X* and *EIF4A1*) in virus-positive tumors spanning multiple cancer types; although *DDX3X* mutations have been reported in HNSCC[141], our results demonstrate their broader relevance across a pan-cancer cohort. Moreover, by aggregating data from 252 BL cases, we found a significant association between *DDX3X* mutations and EBV-positive BL, an association that had been noted previously but not shown to be statistically significant[21,142–146]. Finally, our analysis of immunotherapy response suggests that virus-positive status is associated with enhanced response to checkpoint blockade in gastric cancer and head and neck squamous cell carcinoma.

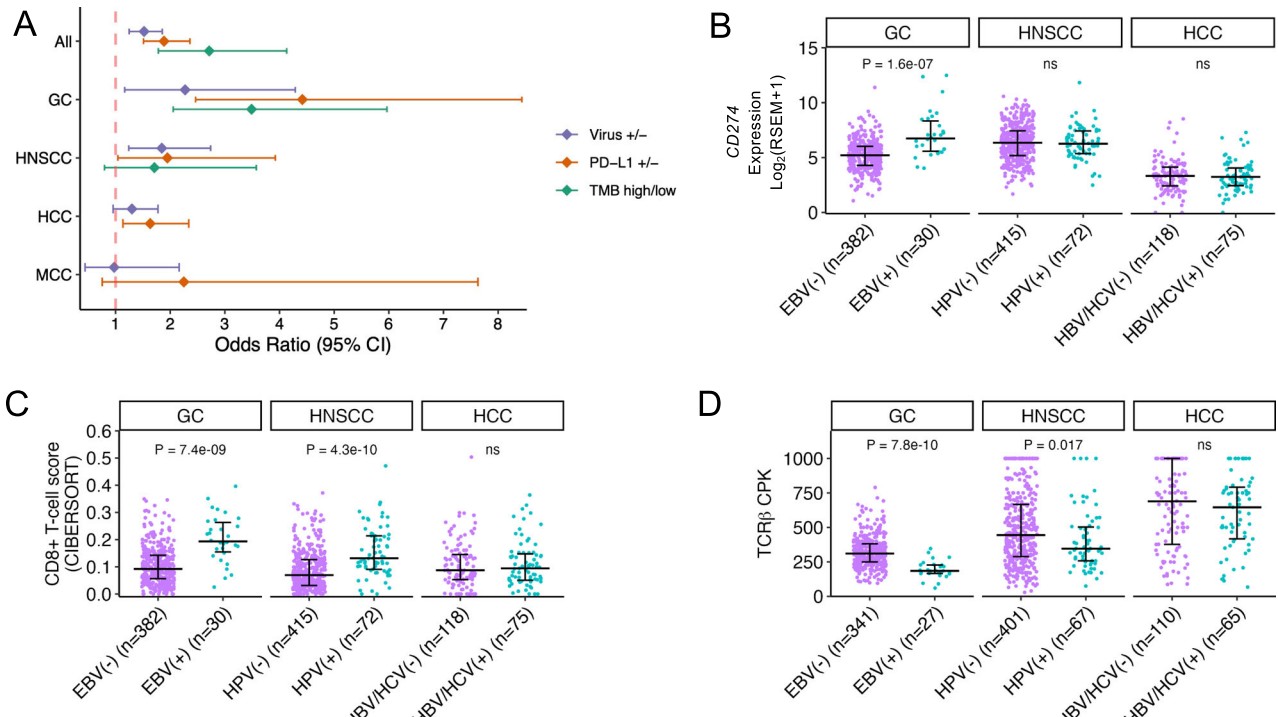

**Fig. 5 | Analysis of biomarkers for immunotherapy response in virus-associated cancers. A** Odds ratio of positive response to treatment with PD-1/PD-L1 inhibitors with virus-positive status, PD-L1 positive status, and/or high tumor mutation burden (TMB) in 32 studies representing four types of cancer, Fisher's exact test. Data are presented as odds ratio values with error bars indicating 95% confidence intervals. **B** Log$_2$(RSEM + 1) expression of *PD-L1* (*CD274*), (**C**) CIBERSORT CD8 + T cell infiltration score, and (**D**) TCRβ clonotypes per thousand reads (CPK), versus viral status of tumors in TCGA studies of GC (TCGA-STAD), HCC (TCGA-LIHC), and HNSCC (TCGA-HNSC). GC, gastric cancer; HCC, hepatocellular carcinoma; HNSCC, head and neck squamous cell carcinoma; MCC, Merkel cell carcinoma. *P* values are calculated by a two-sided Wilcoxon rank-sum test. Data are presented as median values with interquartile range (25th–75th percentile). Source data are provided as a Source Data file.

Through analysis of cancer incidence rates reported in a selection of published studies, we noted virus-associated cancers display greater incidence in males compared to females relative to non-virus-associated cancers. This may be caused in part by immunologic predisposition towards viral infection in male compared to females. In general, females have a more robust immune response to infection, which has been attributed to X-chromosome inactivation and regulation of the immune response by genetic, hormonal, and environmental mediators[172,173].

By a large-scale analysis of DNA sequencing data from 1971 tumors collected from different studies, we found that virus-positive tumors generally display a lower mutation load compared to virus-negative tumors. It has been hypothesized that the oncogenic activity of virus-encoded proteins removes selective pressure for somatic mutations. However, unlike the other virus-associated cancers, virus-positive HCC and BLs (comprised mostly but not exclusively of eBL) have a greater mutation load compared to virus-negative cases. In HCC, this may reflect the activity of HCV oncoproteins that inhibit DNA repair and induce double-stranded breaks in the DNA[174], while in EBV-positive BL, it is more difficult to propose hypotheses on the underlying reason(s). Yet, EBV-positive BL had a lower driver mutation load compared to EBV-negative BL, which may be attributed to the activity of viral proteins such as EBNA1 that reduce selective pressure for the acquisition of driver genetic alterations seen commonly in virus-negative BL[21].

Our analysis noted HPV strain-specific differences in mutation patterns across CC and HNSCC, with consistent trends observed in both cancers. APOBEC signatures were strongly correlated with total mutation load in both cancers[117,135], and HPV strains exhibited distinct APOBEC-associated mutation loads. In particular, HPV33 consistently demonstrated lower APOBEC-associated mutation loads compared to HPV16 in both HNSCC[130] and CC. In CC, the strain-specific differences in APOBEC signature levels appeared to be largely driven by total mutation load. In contrast, in HNSCC, HPV33 retained significantly lower APOBEC-associated mutation counts[130] even after adjusting for total mutations, suggesting additional strain-specific mechanisms beyond overall mutational burden. These findings indicate that APOBEC-mediated mutagenesis plays a central role in shaping the mutational landscape of HPV-positive CC and HNSCC, with HPV strain-specific effects potentially reflecting shared viral-host interaction mechanisms across these cancer types. However, the small sample size for certain HPV strains, such as HPV31 (*n* = 2) in CC, limits the generalizability of these findings. Larger studies are needed to confirm these strain-specific effects and to determine their potential implications for tumor progression and clinical outcomes.

Consistent with Zapatka et al. [2], we observed a lack of TP53 mutations in HPV-positive HNSCC. In addition, we observed that somatic mutation of the RNA helicase protein *DDX3X* was more frequent in virus-positive tumors compared to virus-negative tumors. *DDX3X* additionally functions as a component of the innate immune signaling pathway and is known to inhibit replication of viruses such as HBV by activating production of IFN-beta[156,175]. Some RNA viruses, including HCV and HIV, exploit functions of *DDX3X* to aid in viral replication[156,175]. In cancer, *DDX3X* has been described as both a tumor suppressor and an oncogene in different cancer types and even among different tumors of the same cancer type[176]. *DDX3X* is expressed in many tissues of the body and escapes X chromosome inactivation[156,176]. The relatively high frequency of mutations in *DDX3X* in virus-positive tumors and the near-exclusive male bias for truncating mutations suggests that loss of function of *DDX3X* may contribute to the pathogenesis of some virus-associated cancers, particularly BL, which had

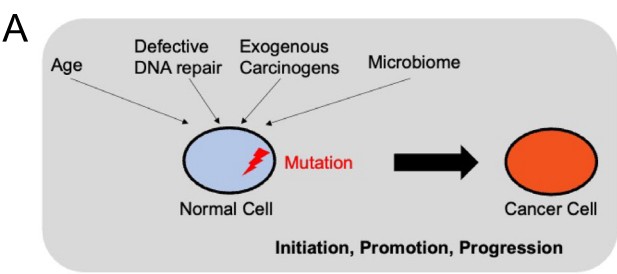

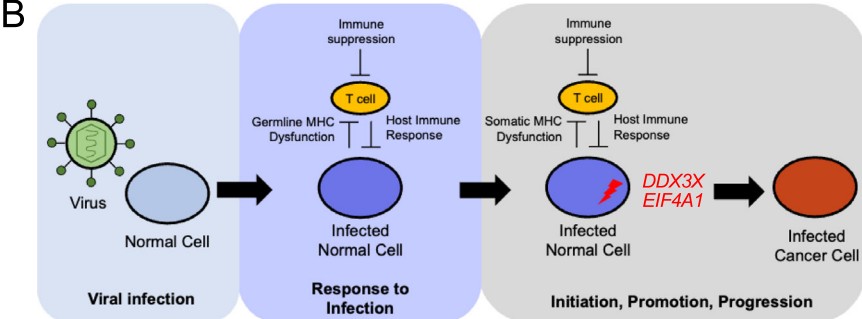

**Fig. 6 | Models of oncogenesis for virus-associated and non-virus-associated cancers. A** Model for oncogenesis in the absence of viral infection. A normal cell accumulates driver mutations as a result of age, defective DNA repair, exogenous carcinogens, or microbiome interactions, leading under selective pressure to initiation, promotion, and progression that ends in the malignant transformation of the cell. **B** Model for oncogenesis in the presence of viral infection. A normal cell is infected with a virus, and a latent infection is established as a result of inadequate host immune response, potentially associated with germline MHC dysfunction or other inherited risk factors. The infected normal cell acquires somatic mutations in specific genes, such as chromatin modifiers like RNA helicases *DDX3X* and *EIF4A1*, leading to initiation, promotion, and progression that ends in the malignant transformation of the infected cell.

the highest frequency of *DDX3X* mutations in this study and for which similar findings were recently reported in another study[157]. It is worth noting that *DDX3X* mutations, although enriched, did not occur exclusively in EBV-positive BLs. Gong and colleagues reported[146] the role of *DDX3X* and its Y-chromosome paralog *DDX3Y* in facilitating MYC-driven lymphomagenesis. Given the pleiotropic role of *DDX3X* in viral recognition and RNA processing and its higher mutational frequency in virus-associated tumors beyond MYC-driven lymphomas, it will be interesting to molecularly dissect the dual role of these mutations in viral and cell processes leading to tumor development. Lastly, while our findings on *EIF4A1* and *DDX3X* remain robust and underscore the potential importance of these RNA helicases in virus-positive tumors, they are derived from a limited number of cancer types. Future studies incorporating larger and more diverse cohorts will help expand our understanding and validate their roles across a broader range of malignancies.

In this study, we focused on the most significantly differentially mutated genes across all virus-positive versus virus-negative cancers, which may have led to the omission of well-established somatic mutation associations in specific cancer types. For instance, the depletion of *CTNNB1* mutations in HBV-positive HCC cases, a known association[2,149], was not highlighted because the *p*-value for *CTNNB1* mutations in the combined cohort of all eligible cancer types did not meet our significance threshold. In contrast, HCV-positive HCC cases are associated with *CTNNB1* mutations[118], and grouping HBV- and HCV-positive cases together as virus-positive masked the HBV-specific trend. Similarly, *TERT* promoter mutations, well-documented in virus-positive HNSCCs[2,177], were not captured in our study because the analysis focused on exonic somatic mutations, excluding promoter regions.

Analysis of ICI clinical trials revealed that virus-positive status could represent a positive biomarker for ICI therapy response in GC and HNSCC[178]. The improved response to immunotherapy of EBV-positive GC patients compared to EBV-negative GC patients is hypothesized to be due to increased expression of *PD-L1*, potentially through activation of the NF-κB pathway by viral protein LMP2A[179]. However, the association between HPV infection and *PD-L1* expression is less clear: some studies report a link between HPV status and *PD-L1* expression[180,181], while others find no association[182,183], the latter of which is consistent with our results from the TCGA HNSCC dataset. In both GC and HNSCC, we observed a marked increase in CD8 + T cell infiltration as well as an increase in T cell receptor clonality in virus-positive tumors. The high immunogenicity of viral antigens expressed in these tumors might elicit a bigger and more clonal T cell response, which could be reactivated through ICI therapy.

While this study does not focus on epigenetic alterations, previous studies have highlighted significant differences in the epigenetic landscapes of virus-associated cancers. For example, HPV-positive CC exhibit higher promoter methylation and increased gene silencing compared to virus-negative cases[117]. Similarly, HPV-positive HNSCC is widely reported to be hypermethylated compared to HPV-negative cases[184,185], with genes such as *CDKN2A*, *RASSF1*, and *CCNA1*, which are involved in cell cycle regulation and apoptosis, frequently affected[184,186,187]. In GC, EBV-positive cancers are characterized by DNA hypermethylation[105]. These prior findings underscore the important role of epigenetic alterations in the pathogenesis of virus-associated cancers.

Our integrative analysis highlights two distinct routes to oncogenesis, in the absence and presence of viral infection (Fig. 6A, B). Further studies will be needed to understand how this model may be incorporated into the development of targeted therapies.

## Methods

### Experimental model and subject details

The Kaposi sarcoma patients (*n* = 10) were enrolled for study at the University of Sassari, Italy. Skin lesion tumor samples and adjacent non-neoplastic cells were surgically resected for DNA sequencing. All biological samples from Kaposi patients (tissue and blood specimens),

along with clinical information including sex (based on self-report) and age, were obtained after written consent from the patients and the sample IDs are anonymized. The study was approved by the Committee for the Ethics of the Research and Bioethics of the National Research Council (CNR n.12629). In addition, clinical and genomic data of 2293 cancer patients was obtained from 17 published studies[21,105,114–117,119–123,152–155,188]. The combined cohort consists of 788 females, 1101 males, and 414 individuals of unknown or unreported sex. The ages range from 1 year to 90 years.

### Epidemiological analysis
Sex ratio in incidence rates of virus-associated and non-virus associated cancers, as well as virus-positive and virus-negative cases of virus-associated cancers, were obtained from studies listed in Supplementary Data 1 and Supplementary Data 2[23–104]. Global age-standardized incidence rates of cancers by country in 2020 were obtained from GLOBOCAN 2020 Cancer Today online portal (https://gco.iarc.fr/today/home). Attributable fraction of cancer cases for each region were obtained from de Martel et al. [3].

### Virus infection status calling
Virus-infection status of patients were reported in the original studies[17,21,105,114–123,152–155,188], or obtained via cBioportal (https://www.cbioportal.org/) for TCGA cervical[117], gastric[105], and head and neck squamous cell carcinoma[119] data sets.

### Single nucleotide and indel variant calling pipeline
WES data from 10 Kaposi sarcoma samples were aligned to GRCh37 using the Burrows-Wheeler aligner. Samples were pre-processed by indel realignment, duplicate removal, and base recalibration with GATK following the GATK best practices workflow. SAVI-v2[189] was used to call somatic variants. The variant list was filtered for variants with a minimum total depth 10 and maximum total depth 700 in both tumor and normal, strand bias $p$-value > 0.001 in tumor and normal and called as significant somatic variants by SAVI ($p$-value < 0.05, and confidence interval for the significance of the tumor/normal comparison > 0). Variants were excluded if they were found in an in-house supernormal created from 186 normal samples from the TCGA, if they were in the cohort supernormal constructed from variants in the ten normal samples, or if they were common SNPs found at a frequency ≥ 5% in the 1000 Genomes Project.

### Mutation load analysis
The mutations in each tumor were obtained from the variant lists reported in the original studies (for previously published cases[21,105,114–117,119–123,152–155,188]), or from the mutation calling pipeline described above (for newly sequenced Kaposi sarcoma cases). Mutations in driver genes were defined as mutations that occurred within genes described as cancer-specific drivers and/or recurrently mutated genes in the original studies[21,105,114,116,117,119–122,152–155,188].

### Mutation signature analysis
Mutation signatures were called from somatic variants separately for each cancer type using SigProfilerExtractor (v.1.1.24)[132], an NMF-based mutational signature extraction tool. The analysis was performed with default parameters, with the minimum and maximum number of signatures set to 1 and 10, respectively. The tool performed a de novo extraction of mutational signatures and decomposed them into COSMIC (v3.4) signatures. Suggested solutions from SigProfilerExtractor were used in our analysis, which included decomposed COSMIC signatures as well as de novo signatures in cases where the reconstruction did not achieve a cosine similarity above 0.8. Signatures were obtained from somatic variants called from whole genome sequencing data when available (Burkitt) or whole exome sequencing data (other cancers).

### Structural variant signature and copy number signature analysis
Structural variant (SV) and copy number (CN) data were obtained from the International Cancer Genome Consortium Accelerating Research in Genomic Oncology (ICGC ARGO) data platform (Legacy ICGC 25 K Data, https://platform.icgc-argo.org/) for the Pan-Cancer Analysis of Whole Genomes (PCAWG) WGS cases, including HCC, HNSCC, CC, and GC. For BL and cHL, SV and CN data were retrieved from their respective publications[21,115]. SV and CN signatures were identified using SigProfilerExtractor (v1.1.24), following the same approach as the mutational signature analysis with default parameters.

### Copy number segmentation and variant calling
Copy number segmentation of Kaposi sarcoma samples was conducted using Sequenza[190] for each pair of tumor of normal samples sequenced by whole-exome sequencing for each case. Copy number segmentation of plasmablastic lymphoma samples was performed with Oncoscan[114]. Copy number segmentation data of other cancers was obtained from the original published studies (Burkitt[21], NKTCL[154]) or cBioportal (TCGA samples[105,117–119]).

### GISTIC analysis
In order to define significant regions of recurrent CNAs across all available virus-associated cancers, GISTIC 2.0[191] was applied to pooled copy number segmentation data of 1557 tumors, using a significance threshold of $q = 0.1$, a maximum segmentation threshold of 10,000, and all other parameters default via the GenePattern server (https://www.genepattern.org/) (Supplementary Data 20). GISTIC peaks of gain or amplification were counted as present in a patient if the maximum inferred tumor copy number within the wide peak limit region was > 2.3 (gain) or > 3.6 (amplification). GISTIC peaks of deletion were counted as present in a patient if the minimum inferred tumor copy number within the wide peak limit region was < 1.7 (heterozygous loss) or < 0.8 (homozygous loss). Arm and whole-chromosome level CNAs were defined as lesions of the same type (i.e., gain or loss) that covered > 75% of the chromosome arm or chromosome, respectively.

### Analysis of recurrently mutated genes in virus-associated cancers
Nonsynonymous mutations in protein-coding genes were analyzed in 1971 patients with available DNA sequencing data. A cancer type or sequencing cohort was included in the combined analysis of a gene only if the gene was covered by the respective sequencing panel and was mutated in at least one sample from that cohort. To minimize noise from hypermutated samples, only cases with fewer than 300 mutations were included across all cancer types.

Significant genes were identified based on an odds ratio (OR) greater than 1 for mutations in virus-positive versus virus-negative cases in the combined dataset, a Benjamini-Hochberg (BH) corrected $p$-value below 0.005, and recurrence in at least two virus-positive or virus-negative cases across at least three unique cancer types. In addition, the final combined direction of mutation prevalence had to be consistently supported by at least half of the included cancer types.

Significant copy number alterations were identified using the pan-cancer GISTIC analysis (see "GISTIC Analysis" section) and required an odds ratio greater than 1 for mutations in virus-positive versus virus-negative cases, with a BH corrected $p$-value below 0.0001.

### Analysis of immunotherapy trials
The comparative analysis of response to immunotherapy was performed using data from ClinicalTrials.gov and did not include any unpublished clinical trial data. Nine checkpoint inhibitors targeting either PD-1 or PD-L1 were included: Nivolumab, Pembrolizumab, Cemiplimab, Atezolizumab, Avelumab, Durvalumab, Camrelizumab, Sintilimab, Toripalimab. On September 22nd 2022, trials were collected using the following query: "((NOT NOTEXT) [CITATIONS]) AND (< ICI

drug name 1 > OR < ICI drug name 2 > OR …)", where ICI drug names correspond to the ones listed above as well as any known synonyms (e.g., Nivolumab: Opdivo, ONO-4538, BMS-936558, MDX1106). Only single-arm, interventional studies where the RECIST1.1 objective response rate was available specifically for PD-1/PD-L1 inhibitors (with no other non-ICI combination therapies) were included. Whenever possible, response stratified by virus status, PD-L1 expression and tumor mutational burden was collected. "PD-L1 positive" refers to patients that were classified as PD-L1 positive in the original study (most often, ≥1% of all cells). "Tumor mutational burden (TMB) high" refers to patients classified as TMB high in the original study (with different cut-offs used depending on the study and/or tumor type). Due to the lack of individual patient information, each biomarker for response was evaluated independently (Fisher's exact test).

### Immune infiltration and T cell receptor analysis

TCGA RSEM expression data was obtained from cBioportal. TCGA deconvolution results were downloaded from the TIMER2.0 website (http://timer.cistrome.org/)[164]. For the T cell receptor (TCR) clonality analysis, we focused on the complimentary determining region 3 (CDR3) of both the α and β chains, which is the most highly variable sequence of the TCR and the most important for determining antigen affinity[192]. We obtained the previously published TRUST4 TCR calls from the authors, and we applied the two diversity metrics from the original study[171], namely the number of clonotypes per thousand CDR3 reads (CPK), and clonality, defined as (1 – normalized Shannon entropy). All comparisons were performed using the Wilcoxon rank sum test / Mann-Whitney U test.

### Statistics

Analyses of the significance of mutation counts and frequencies were performed using the two-sided Wilcoxon rank sum test (Mann-Whitney U test) and two-sided Fisher's exact test, respectively. Odds ratios of mutation by virus status were computed with Haldane-Anscombe correction when applicable. The 95% confidence interval of odds ratios were estimated using the normal approximation (Wald). Multiple hypothesis corrections were applied using the Benjamini-Hochberg Procedure and reported as q-values. Combined p-values were computed using Fisher's method (unweighted, based on one-sided p-values) and wFisher method (weighted by sample size, based on two-sided p-values) as implemented in the metapro R package[193]. No data were excluded from the statistical tests, except in the analysis of recurrently mutated genes in virus-associated cancers, where only cases with fewer than 300 mutations were included across all cancer types to minimize noise from hypermutated samples.

### Reporting summary

Further information on research design is available in the Nature Portfolio Reporting Summary linked to this article.

## Data availability

WES raw data (FASTQ and BAM files) of Kaposi sarcoma generated in this study have been deposited in the European Nucleotide Archive (ENA) at EMBL-EBI database under accession code PRJEB76508. The TCGA (HNSCC, CC and GC) cohort's clinical data, mutation and copy number alteration calls are available at cBioportal (HNSCC:https://www.cbioportal.org/study/summary?id=hnsc_tcga, CC:https://www.cbioportal.org/study/summary?id=cesc_tcga_pan_can_atlas_2018, GC:https://www.cbioportal.org/study/summary?id=stad_tcga), raw and normalized chromosomal instability (CIN) signature activities are available within the supplementary information of Drews et al.'s publication[139]. SV and CN data of PCAWG WGS cases are available from the International Cancer Genome Consortium Accelerating Research in Genomic Oncology (ICGC ARGO) data platform (Legacy ICGC 25 K Data, https://docs.icgc-argo.org/docs/data-access/icgc-25k-data#open-release-data-object-bucket-details; SV: s3://icgc25k-open/PCAWG/consensus_sv/; CN: s3://icgc25k-open/PCAWG/consensus_cnv/). Clinical and genomic data including SNV, CN, and/or SV calls of other cancer types are available within the supplementary information of their published studies[21,105,114–117,119–123,152–155,188]. EBV status of the Hodgkin lymphoma cohort from Alig et al.[123] has been acquired from the authors directly. Source data are provided in this paper.

## Code availability

All analyses were performed using publicly available software: for alignment of Kaposi sarcoma sample WES data to GRCh37, Burrows-Wheeler aligner v.0.7.17 (https://github.com/lh3/bwa); for calling somatic variants, SAVI version 2 (https://github.com/WinterLi1993/SAVI); for calling mutation signatures from somatic variants, SigProfilerExtractor v.1.1.24 (https://github.com/AlexandrovLab/SigProfilerExtractor); for copy number segmentation of Kaposi sarcoma samples, Sequenza (https://sequenzatools.bitbucket.io/); for defining significant regions of recurrent CNAs, GISTIC version 2.0 (https://broadinstitute.github.io/gistic2/); for combining p-values using the weighted Fisher method (wFisher), metapro (http://github.com/unistbig/metapro). For general coding, R (version 4.4.1) and Python (version 3.9.21) were used. Detailed information on the software used is also provided in the respective sections of the Methods.

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

## Acknowledgements

We would like to express our sincere thanks to Laura Pasqualucci for her invaluable suggestions, insightful contributions and expertise in the field of lymphoma, which significantly enriched this study. This work was funded by the National Institutes of Health, National Cancer Institute grants R35CA253126 and U01CA243073 (R.R.), Fondazione AIRC (Investigator Grant no. 23732 to E.T.) and SU2C Convergence Program (K.G., Y.N., J.R. and R.R.).

## Author contributions

Conceptualization, Y.N., K.G. and R.R.; Methodology, K.G. and R.R.; Investigation, Y.N., K.G., J.R., C.K., M.A., V.Z. and T.L.; Formal Analysis, Y.N., K.G. and J.R.; Writing – Original Draft, K.G.; Writing – Review & Editing, Y.N., K.G., J.R. and R.R.; Resources, M.C., A.L., G.P., A.C., E.T. and R.R.; Supervision, R.R.

## Competing interests

R. Rabadan is the founder of Genotwin, a member of the advisory board of Diatech Pharmacogenetics and Flahy. None of these activities are related to the results in the current manuscript. The remaining authors declare no competing interests.
