## [Peer Review File · Nature Communications]

Genomic landscape of virus-associated cancers

Corresponding Author: Professor Raul Rabadan

Version 0:

Reviewer comments:

Reviewer #1

(Remarks to the Author)

In this work, Nam et al. analyze some epidemiological, genomic and clinical features of virus-associated cancers. Although the effort is commendable, the data sets gathered for the genomic analysis are heterogenous in terms of sequencing technology and size. Only a minority of cases were analyzed by whole genome sequencing, and larger series were previously analyzed for most cancer types. As a result, many of the described associations between genomic features and viral infections were previously published, and several robust known associations are missed in the present work. Thus, although this study is statistically sound and clearly written, it lacks focus in my opinion and misses the objective of proposing a comprehensive characterization of the genomic features of virus-associated cancers, beyond previously published studies.

More specific comments are listed below :

- 1) In the Introduction, the authors should mention AAV2 as associated with the development of cancers.
- 1) It is difficult in Fig. 1A to understand which individual cancers have an increased male/female ratio in virus-positive cases. I would suggest placing HBV+/- data from a same cancer adjacent in the figure. Also, considering each series as a single point considerably limits statistical power. I understand that retrieving patient-level data from each cohort goes beyond the scope of this study, but I am not sure this incomplete epidemiological analysis should be included in this manuscript dedicated to genomic features.
- 2) In Fig. 1B,C, does the incidence of EBV-positive cases correlate with a higher rate of EBV infection, or the presence of specific strains?
- 3) The authors retrieved directly mutation lists from original studies, and did not start from raw data to ensure homogeneous processing. Thus, how did the authors account for technical and computational discrepancies between studies? In particular, targeted, whole exome and whole genome data were used so some genes will be covered in some data sets and not others. Finally, the number of cases per cancer type (which should be shown as a main figure) is very heterogeneous, so more represented cancers (e.g. HL) will have more impact in the results. How did the authors account for this? I would suggest generating statistics per cancer type first and combining p-values afterwards.
- 4) How were data sets selected? It is of course difficult to be exhaustive, but much broader whole genome sequencing data sets are available, e.g. ICGC data for HCC.
- 5) Mutation burden analysis is shown as log₂ (nonsynonymous mutations) in Fig. 2. It would be more informative to provide data in mutations per megabase, allowing absolute comparisons across cancers.
- 6) Please include the list of driver genes considered for each cancer as supplementary table.
- 7) The authors should thoroughly compare their results with previous work to state what is new and explain why previously described associations are missed. For example, the enrichment of TP53 mutations in HBV-related HCC (Amaddeo, Hepatology 2015), or the depletion of TP53 and CDKN2A alterations in HPV-positive HNSCC (Zapatka, Nat Genet 2020) are well known. Depletion of CTNBN1 mutations in HBV-related HCC (Amaddeo) or TERT alterations in HPV-positive HNSCC (Zapatka) are missed...

- 8) The authors state they performed mutational signature analysis "informed by de novo signature calling". What does it mean? Did the authors identify new signatures? In Fig. 3 it is difficult to see the mutation counts for tumors with the lowest mutation burden. I would suggest cutting the axes to increase readability. Also in the boxplots please define "signature activity".
- 9) In addition to mutational signatures, it would be a great addition to study structural variant and copy-number signatures, which were previously associated with viral insertions.
- 10) Again, several results of the mutational signature analysis, e.g. association of SBS24 with HBV-positive HCC (Schulze, Nat Genet 2015) or association of SBS2 with HPV-positive HNSCC (Zapatka, Nat Genet 2020) so it should be clearly stated what results are new to this study.
- 11) The association of DDX3X and EIF4A1 mutations with virus-positive cancers seems strongly driven by only a couple of cancer types. The authors should either increase sample size to obtain stronger statistics in other cancers, or clearly state this limitation.
- 12) The association of viral status with immune checkpoint inhibitor response is valuable. However I don't understand the strategy used to determine whether viral status is independent from PD-L1 expression. Why not use directly the data shown in Fig. 5A to see if viral status and PD-L1 expression are significant in multivariate models?

Reviewer #2

(Remarks to the Author)

The paper examines the genomic characteristics of virus-positive and virus-negative tumors across nine different cancer types. The primary objective is to delineate the differences in genomic features, such as mutation frequencies, between these two groups. The study builds on over a century of research into tumor viruses, aiming to provide new insights into the genomic landscapes shaped by viral infections. However, its impact is diminished by the lack of clarity regarding novel findings and insufficient citation of previous works.

Major comments are listed below.

1. Cite additional previous works to provide context and support. Additionally, clarify whether the findings are novel. Without this clarification, it is difficult to gauge the contribution of this work. Many results seem have been reported previously. For example, TCGA GC study [PMID: 25079317] has reported that "most EBV-positive cases were male (81%, $P = 0.037$), as previously reported [PMID: 19445939]." Immunotherapy Efficacy in Virus-associated Cancers have been reviewed [EJMO 2022;6(2):100–110]. Additionally, virus-associated Merkel cell carcinomas are known to carry a low mutational burden [PMID: 26238782]. It may be beneficial to comment more frequently if a finding from this work agrees with previous work or is a novel finding. Without this clarification, it is challenging to assess the unique contributions of the study. This ambiguity undermines the paper's impact and may lead to the perception that it offers little new information.
2. DDX3 and eIF4A results are aggregated across different cancer types. This approach can be misleading because it masks the variability and specific associations that may be present in individual cancer types. For example, eIF4A is only significantly elevated odds of mutation in virus-positive tumors in GC ($q=0.04$) but not in other cancer types. However, when combining cancer types, the adjusted P value becomes much more significant ($q= 3.46E-05$). Aggregating data could lead to incorrect conclusions about the generalizability of these findings across all cancer types. It's of interest to examine if the association between eIF4A mutation and virus status is still significant after adjusting different cancer types as the confounder.
3. Investigate cancer subtypes. Cancer is a heterogeneous disease, and different subtypes can have distinct genomic characteristics and responses to viral infections. By not analyzing subtypes, the study may overlook important differences.
4. Investigate WGS samples. The study seems limited to WES and targeted sequencing. WGS provides a more comprehensive view of the genome, including non-coding regions and large structural changes that WES and targeted sequencing might miss. Including WGS samples (e.g., ones from PCAWG [PMID: 32025001]) could examine chromosomal instability (CIN) and structural variants (SV) between virus-positive and virus-negative tumors.
5. Mutational signatures were analyzed by Palimpsest. Would the same conclusions be hold when using the state-of-art methods SigProfilerExtractor [PMID: 36388765] or MuSiCal [PMID: 38361034].
6. Viral sub-strains. Is it possible to stratify the analysis by viral sub-strains.

Minor:

1. Fig1 a. show confidence interval.
2. Define GC in the main text.
3. The title is "Genomic landscape of virus-associated cancers". But the first section is about epidemiological trends which seems unrelated to the major theme of the manuscript.

Reviewer #3

(Remarks to the Author)

This study provides a comprehensive analysis of the genomic changes associated with virus driven cancers. It confirms previous epidemiological analyses showing gender and geographical variations in the incidence rates of these virus-associated tumours. The overall conclusion of fewer mutations and deletions in virus-positive versus virus-negative cases, particularly in certain genes such as p53, is consistent with previous studies where individual virus-associated cancers have been examined. The novel observation of increased mutations in the RNA helicases DDX3X and EIF4A is interesting and provides an important focus for more detailed molecular studies.

Specific comments.

1. Some of the references are old and not relevant. For instance, more recent references covering the different patterns of virus gene expression in EBV-associated malignancies should be included.
2. The finding that EBV-positive Hodgkin lymphoma (HL) occurs 'most frequently in North Africa, the Middle East and South America' is surprising and raises concerns about the methodology used to ascribe EBV positivity in the included studies. It would also be useful to include information about the age profile of these tumours in the different countries.
3. Previous studies have also found that the total mutation count is lower in virus-positive compared to virus-negative tumours while highlighting important differences in the epigenetic landscape in these tumours. While understandingly not the focus of the current manuscript, it is important to mention the contribution of epigenetic alterations to the pathogenesis of virus-associated cancers.
4. Mutations in DDX3X appear to be more common in virus-associated lymphomas rather than carcinomas. How is this explained?
5. These results suggest that EBV-positive status may be a positive prognostic marker for patients undergoing ICI therapy for GC and HNSCC, which may be correlated with PD-L1 expression in GC but may represent an independent marker in HNSCC.' This statement lines 275-278 is confusing. What EBV-positive HNSCC is being referred to? Comparisons with EBV-positive NPC are difficult as all these tumours are virus positive.
6. The response to ICI therapy could be examined in the context of the tumour microenvironment as the TCGA includes transcriptional data related to infiltrating lymphocytes etc.

Version 1:

Reviewer comments:

Reviewer #1

(Remarks to the Author)

The authors performed an in-depth revision of their manuscript and satisfactorily answered to all my comments. They considerably improved the clarity of the manuscript by providing many additional details and improving the clarity of the figures.

They added new data (PCAWG data sets) and analyses (structural variant and copy-number signatures).

The manuscript now clearly states which findings are novel and how the results compare with previous work.

I find the revised manuscript very convincing and I now support its acceptance for publication in Nature Communications.

Reviewer #2

(Remarks to the Author)

The authors have addressed most of my previous comments. I have two minor comments for further improvement:

1. Clarification of Novel Findings:

The revised manuscript has cited more prior works and highlights four key findings: higher male incidence, reduced mutation burden, frequent RNA helicase mutations, and an enhanced immunotherapy response in virus-associated cancers. It would be beneficial to clarify in the Discussion which of these findings confirm previous reports and which represent novel observations. Clearly distinguishing these points will help readers understand the significance of the findings and identify areas for further study.

2. Specification of Data Sources:

Although the paper cites 23 studies involved in this analysis and the supplementary table provides links to the corresponding publications, it is not clear where the actual datasets used in this analysis were obtained. For enhanced transparency and reproducibility, please specify the direct links or identifiers for the datasets employed in this study.

Reviewer #3

(Remarks to the Author)

The authors have comprehensively addressed my comments and have revised their manuscript accordingly.

Point-by-Point Response to Reviewer Comments

Reviewer #1 (Remarks to the Author): Expert in cancer genomics and virus-associated cancers

Reviewer's summary) *"In this work, Nam et al. analyze some epidemiological, genomic and clinical features of virus-associated cancers. Although the effort is commendable, the data sets gathered for the genomic analysis are heterogenous in terms of sequencing technology and size. Only a minority of cases were analyzed by whole genome sequencing, and larger series were previously analyzed for most cancer types. As a result, many of the described associations between genomic features and viral infections were previously published, and several robust known associations are missed in the present work. Thus, although this study is statistically sound and clearly written, it lacks focus in my opinion and misses the objective of proposing a comprehensive characterization of the genomic features of virus-associated cancers, beyond previously published studies."*

Response) We thank the reviewer for the thoughtful feedback. We believe that integrating multiple data sources enables a broader and more comprehensive analysis of virus-associated cancers across tumor types, providing insights that individual tumor-type studies alone cannot achieve. The novelty of our study lies in this integrative approach.

A prior study, "The landscape of viral associations in human cancers" by Zapatka et al., published in *Nature Genetics* in 2020, systematically investigated viruses in tumors as part of the Pan-Cancer Analysis of Whole Genomes Consortium¹. It identified trends in somatic mutations in certain virus-positive tumors. Building upon this foundational work, our study significantly expands the scope by analyzing a larger dataset that includes more oncoviruses, virus-positive tumors, and cancer types. Specifically, we examined 2,488 cancer samples encompassing all seven known oncoviruses and 14 virus-associated cancers, nine of which include both virus-positive and virus-negative tumors for direct comparison. This comprehensive approach reveals common patterns and provides insights that extend beyond individual tumor types.

From our analysis, we highlight the following key findings:

1. **Higher Male Incidence:** Virus-associated cancers exhibit a significantly higher incidence in males than in females, exceeding the gender disparity observed in non-virus-associated cancers.
2. **Reduced Mutation Burden:** Virus-positive tumors generally harbor fewer nonsynonymous somatic mutations compared to virus-negative tumors of the same cancer type.
3. **Frequent RNA Helicase Mutations:** Virus-positive tumors frequently harbor mutations in RNA helicases, such as DDX3X and EIF4A1, many of which appear to reduce protein function in a sex-specific manner. These mutations are recurrently observed across various virus-related tumors, including natural killer/T-cell lymphoma, aggressive NK-cell lymphoma, and Burkitt lymphoma (BL), among others. Notably, by aggregating data from 252 BL cases, we identified a significant association between DDX3X mutations and EBV-positive BL, an association previously reported but not statistically significant.
4. **Enhanced Immunotherapy Response:** Virus-positive tumors, particularly gastric cancer (GC) and head and neck squamous cell carcinoma (HNSCC), demonstrate higher response rates to

immune checkpoint inhibitors. These tumors also exhibit increased CD8+ T-cell infiltration and show evidence of T-cell receptor clonal expansion.

Additionally, in the following responses, we have detailed how we mitigated the effects of data heterogeneity, included analysis from additional whole-genome sequencing (WGS) cases, and addressed why some well-established associations may not have been captured in our findings.

Specific comments:

Comment 1) *“In the Introduction, the authors should mention AAV2 as associated with the development of cancers.”*

Response 1) We have included the association of AAV2 with hepatocellular carcinomas along with CMV in GBM as below in the introduction section.

Modified in Introduction section) *“In addition, previous studies have observed potential associations of adeno-associated virus 2 (AAV2) with hepatocellular carcinomas (HCC)², cytomegalovirus (CMV) with glioblastoma multiforme (GBM)³, and a "hit-and-run" mechanism of viral involvement in classical Hodgkin lymphoma (CHL)⁴.”*

Comment 2) *“It is difficult in Fig. 1A to understand which individual cancers have an increased male/female ratio in virus-positive cases. I would suggest placing HBV+/- data from a same cancer adjacent in the figure. Also, considering each series as a single point considerably limits statistical power. I understand that retrieving patient-level data from each cohort goes beyond the scope of this study, but I am not sure this incomplete epidemiological analysis should be included in this manuscript dedicated to genomic features.”*

Response 2) We appreciate the reviewer’s comments and have revised Fig. 1A for clarity by separating it into two sub-figures: Fig. R1A now compares the sex incidence of virus-associated versus non-virus-associated cancers, while Fig. R1B focuses specifically on the sex incidence of virus-positive and virus-negative cases within virus-associated cancers. We have adjusted the layout of Fig. R1B so that virus-positive and virus-negative data points within each cancer type appear adjacent for easier comparison. Additionally, we went through each cohort in Fig. R1A retrieving the sample sizes and merged them into one point per cancer type with confidence interval lines to increase clarity and statistical power. The cohorts with no sample size information were taken out. The updated figures show significant difference of male-to-female ratio between virus-associated versus non-virus-associated cancers (Fisher’s Exact P = 2.2e-16, Fig. R1A) and between virus-positive and virus-negative cases within virus-associated cancers (Wilcoxon Rank Sum Test P = 5.23e-08, Fig. R1B). We included Fig. R1 as Fig. 1A-B in the revised manuscript.

One of the manuscript’s primary objectives is to identify the genomic differences of virus-associated cancers. Among our main findings, we identified recurrent mutations in the *DDX3X* gene, located on the sex chromosome, which the truncating mutations occur predominantly in males and are more frequent in virus-positive cases (Figures 4A, C, and D). Although it may not be the sole factor, these observations

suggest that virus-associated, sex chromosome-linked mutations could be associated to the observed male-biased incidence of these cancers.

Therefore, we decided it as important to highlight the differences in incidence between males and females (M/F ratio) in these malignancies. For instance, in gastric cancer, the known discrepancy can be striking: EBV-positive gastric tumors have a mean M/F ratio of 2.7 (n=33 studies), in contrast to EBV-negative tumors, which have a mean M/F ratio of 0.93 (n=31 studies; Wilcoxon p-value = 1.0e-10; Fig. R1B). In a meta-analysis by Lee et al.⁵, encompassing multiple populations from Chile, Colombia, The Netherlands, China, and Japan (totaling 9,738 patients), the M/F ratio in EBV-positive cohorts reached as high as 12.67 (van Beek et al. cohort), while the highest M/F ratio in EBV-negative groups was only 2.0 (Moritani et al. cohort)⁵. A similar trend is observed in head and neck squamous cell carcinoma (HNSCC), where the M/F ratio is 8.37 in HPV-positive cases⁶, compared to 3.22 in HPV-negative cases⁷. These pronounced sex differences in virus-positive cancers underscore the significant influence of viral factors on gender disparities in cancer incidence.

Fig. R1. Epidemiological trends of virus-associated cancers. A) Incidence ratios of virus-associated and non-virus-associated cancers, analyzed using Fisher's Exact Test. Error bars represent 95% confidence intervals for each M/F incidence ratio, and B) Virus-positive and virus-negative tumors in virus-associated cancers in males compared to females (M/F) reported in selected published studies, analyzed using Wilcoxon Rank Sum Test. Each point corresponds to an incidence ratio reported in a published study in Table S2. Virus-associated cancers: ANKL, aggressive NK-cell leukemia; GC, gastric cancer; HCC, hepatocellular carcinoma, HL, Hodgkin lymphoma; HNSCC, head and neck squamous cell carcinoma, MCC, Merkel cell carcinoma; NKTCL, Natural killer/T-cell lymphoma; NPC, nasopharyngeal carcinoma; PBL, plasmablastic lymphoma; PCNSL, primary central nervous system lymphoma. Non-virus associated cancers: ALL, acute lymphoblastic leukemia; BCC, basal cell carcinoma; CLL, chronic lymphocytic leukemia. Virus-positive tumors: BL, Burkitt lymphoma.

Comment 3) “In Fig. 1B,C, does the incidence of EBV-positive cases correlate with a higher rate of EBV infection, or the presence of specific strains?”

Response 3) EBV prevalence is consistently high worldwide, with over 90% of adults infected across both developed and developing countries⁸. The relationship between EBV infection rates and the incidence of EBV-positive classical Hodgkin lymphoma (cHL) and nasopharyngeal carcinoma (NPC) remains unclear. However, timing of infection and the initial immune response to EBV vary significantly by region. In

developing countries, infection typically occurs early in childhood, potentially leading to a distinct immune response and a higher likelihood of persistent EBV in cells—an established risk factor for certain cancers⁹. In contrast, in developed nations, primary infection is more often delayed until adolescence or early adulthood, where it frequently presents symptomatically as infectious mononucleosis¹⁰. In regions with earlier childhood infections, cHL cases are more likely to be EBV-positive, whereas in regions with delayed infection, EBV-positive cHL is less common. This suggests a potential link between infection timing and the likelihood of developing EBV-positive cHL⁹.

The two major types of EBV, known as EBV-1 and EBV-2, show distinct geographic distributions. EBV-1 is prevalent among populations in Europe, the Americas, China, and South Asia, while EBV-2 is rarer in these regions and more frequently found in Central Africa, Alaska, and Papua New Guinea, where it appears at rates comparable to EBV-1^{11,12}. Instances of dual infections with both EBV types have also been documented in these areas^{13,14}. EBV-1 is the more commonly observed type in EBV-positive cases of cHL and NPC; however, it is still uncertain whether specific EBV strains directly contribute to these cancers. Further research is necessary to clarify the roles of individual strains in the development of cHL and NPC^{15,16}.

Comment 4) *“The authors retrieved directly mutation lists from original studies, and did not start from raw data to ensure homogeneous processing. Thus, how did the authors account for technical and computational discrepancies between studies? In particular, targeted, whole exome and whole genome data were used so some genes will be covered in some data sets and not others. Finally, the number of cases per cancer type (which should be shown as a main figure) is very heterogeneous, so more represented cancers (e.g. HL) will have more impact in the results. How did the authors account for this? I would suggest generating statistics per cancer type first and combining p-values afterwards.”*

Response 4) We appreciate the reviewer’s concerns regarding data processing consistency and the representation of cancer types. Below are the approaches we implemented to ensure that differences across datasets had minimal impact on our results.

Cohort-specific analysis

To address potential cohort discrepancies, we focused our mutation burden (nonsynonymous mutation counts) and mutational signature analyses primarily within each study and each cohort independently (e.g., within the discovery cohort of each study). Among the cancer types in Figure 2, only classical Hodgkin lymphoma by whole-exome sequencing (CHL_WES) and plasmablastic lymphoma (PBL) originally included multiple combined cohorts (2 cohorts for CHL_WES and 3 cohorts for PBL). Most cohorts within these groups showed a consistent trend, with virus-positive cancers having a lower mutation burden (Fig. R2 and Table R1). When combining p-values for each cohorts (with and without weighting by sample size; Fisher’s method and wFisher¹⁷), the results remained statistically significant (Table R2). However, given that Liu 2020 WES cohort and Liu 2020 targeted sequencing had only one EBV-negative sample, we decided to use only Ramis-Zaldivar 2021 targeted sequencing cohort to show in Figure 2. Figure 2 has been updated accordingly. For all other cancer types, analyses were conducted within single cohorts. This approach helped minimize cross-study variability and ensured reliable, consistent mutation burden comparisons.

Fig. R2. Tumor mutation burden on each cohort within classical Hodgkin lymphoma by whole-exome sequencing (CHL_WES) and plasmablastic lymphoma (PBL) cancer types. A) CHL WES cohorts (n = 2). B) PBL cohorts (n = 3). For all PBL cohorts, mutation counts were calculated based on the overlapping targeted genes from the targeted sequencing panels (Liu 2020 and Ramis-Zaldivar 2021). Wilcoxon rank sum test p-values are shown.

Cancer group	cohort	total_n	n_virus_neg	virus_neg_median	n_virus_pos	virus_pos_median	Wilcoxon Statistic (two-sided)	Wilcoxon P-value (two-sided)	Wilcoxon Statistic (one-sided, virus_pos_isless)	Wilcoxon P-value (one-sided, virus_pos_isless)
CHL_WES	Tiacci 2018	34	30	48.5	4	1.5	-2.645886	0.008148	-2.645886	0.004074
CHL_WES	Maura 2023	35	28	125.0	7	55.0	-2.37126	0.017728	-2.37126	0.008864
PBL	Liu 2020 WES	15	1	5.0	14	2.5	-1.157275	0.24716	-1.157275	0.12358
PBL	Liu 2020 targeted-seq	6	1	0.0	5	1.0	1.17108	0.241567	1.17108	0.879217
PBL	Ramis-Zaldivar 2021 targeted-seq	30	12	4.0	18	1.0	-2.264836	0.023523	-2.264836	0.011761

Table R1. Statistics for each cohort of classical Hodgkin lymphoma by whole-exome sequencing (CHL_WES) and plasmablastic lymphoma (PBL) from Fig.R2.

Cancer group	pvalue combination method	direction	Combined P-value
CHL_WES	unweighted Fisher	Virus pos is lower	0.000405
CHL_WES	Weighted Fisher	Virus pos is lower	0.000812
PBL	unweighted Fisher	Virus pos is lower	0.038155
PBL	Weighted Fisher	Virus pos is lower	0.029619

Table R2. Combined p-values for cohorts of classical Hodgkin lymphoma (CHL_WES) and plasmablastic lymphoma (PBL) analyzed using unweighted and weighted Fisher's methods.

Gene coverage and mutation considered across datasets

In our analysis of genes recurrently mutated in virus-positive versus virus-negative tumors, we pooled cases from multiple cancer types for each gene to compare the overall mutation rate between virus-positive and virus-negative cases. In this approach, we included each cancer type or sequencing cohort in the combined pool only if the gene of interest was covered by the respective sequencing panel. Additionally, to ensure

that each cancer type contributed meaningfully to the combined results and to provide a clearer statistical framework, we applied two additional criteria for analyzing and selecting genes of interest:

1. A cancer type was included in the analysis only if the gene was mutated in at least one sample (virus-positive or virus-negative).
2. A gene was considered a candidate of interest only if the final combined direction was consistently supported by at least half of the included cancer types.

As a result of these additional criteria, *STAT3* (associated with virus-positive tumors) and *TNFAIP3* (associated with virus-negative tumors) have been included to recurrently mutated genes in virus-positive versus virus-negative tumors, alongside *EIF4A1* and *DDX3X* (virus-positive-associated) and *TP53* and *CDKN2A* (virus-negative-associated).

To clarify which cancer types and how many cases were considered for each gene, we included per-cancer-type statistics in Fig. S14 and Fig. S15 (Fig. R3 and Fig. R4) and in Table S14. Moreover, a heatmap has been added to the right side of Fig. 4A to illustrate the mutation rate trends across cancer types and clarify which cancer types were included in the analysis (Fig. R5).

Fig. R3. Combined and individual cancer-type trends in virus-positive-associated genes. A–C) Boxplots of bootstrapped ($n = 10,000$) one-sided Fisher's exact p-values (direction: higher mutation rates in virus-positive cases) for each cancer type included in the combined analysis of each gene. Number of samples used in the analysis for each cancer type are shown in parenthesis. D–F) Differences in mutation proportions between virus-positive and virus-negative cases for each cancer type. * MWU test, $p < 0.05$.

Fig. R4. Combined and individual cancer-type trends in virus-negative-associated genes. A–C) Boxplots of bootstrapped ($n = 10,000$) one-sided Fisher's exact p-values (direction: higher mutation rates in virus-negative cases) for each cancer type included in the combined analysis of each gene. Number of samples used in the analysis for each cancer type are shown in parenthesis. D–F) Differences in mutation proportions between virus-positive and virus-negative cases for each cancer type. * MWU test, $p < 0.05$.

Fig. R5. Combined \log_{10} (odds ratio) of mutation in genes associated with virus-positive (top) and virus-negative (bottom) status ($q < 0.005$) from pooled data of 1,971 tumors across 9 virus-associated cancers. The heatmap on the right displays the cancer cohorts included in the pooled data for the calculation of each gene, with colors representing mutation rate trends in each cohort (red: higher in virus-positive; blue: higher in virus-negative) and shades indicating the two-sided Fisher's exact test p-value.

Showing the number of cases per cancer type and the rationale for combining cancer types in a pan-cancer analysis

Following the reviewer's suggestion, we have included a figure showing the number of cases per cancer type used in the mutated gene comparison analysis (Fig. R6). This figure now appears as Fig. S13 in the revised manuscript.

We understand the reviewer’s concern that cancer types with larger sample sizes might exert a stronger influence on the results when examining mutations in virus-positive versus virus-negative tumors. However, we perform a pan-cancer analysis by combining all available samples to uncover trends that may remain undetected in individual cancer types due to limited sample sizes. For example, when we analyze all 603 cases together for the EIF4A1 gene, the resulting p-value is more significant than in any individual cancer type, indicating a collective signal rather than one driven by a single cancer type (Fig. R3A). Likewise, in all six genes selected, multiple cancer types jointly support the overall direction and strength of the finding. In addition, as suggested by the reviewer, we conducted a p-value combination analysis for each gene, both with and without weighting by sample size (using Fisher’s method and wFisher¹⁷, respectively). The results remained statistically significant for most genes except *DDX3X* (Table R3). However, when we incorporated the extended Burkitt lymphoma dataset, which specifically includes *DDX3X* mutation data aggregated from 252 cases in previously published studies, we observed a significant association between *DDX3X* mutations and EBV-positive BL (p=3.78e-6). This association was not significant in the original BL cohort (n=120, p=0.19) used for the initial *DDX3X* analysis. When using p-values from the extended BL cohort to calculate the combined p-value for the *DDX3X* gene, both unweighted and weighted methods produced significant results (p=0.001 and p=2.57e-4, respectively; Table R4). These findings have now been included in Table S14 of the revised manuscript.

Fig. R6. Number of cases by each cancer types used in virus-positive/negative associated mutated gene analysis colored by sequencing technologies.

Gene	Cancer	Mutated Virus+	Mutated Virus-	Not Mutated Virus+	Not Mutated Virus-	OR (>1 towards virus+)	OR lower	OR upper	Fisher’s Exact P value	Adjusted P value (BH)	Unweighted Fisher Combined P value	wFisher Combined P value
EIF4A1	Combined	15	0	246	342	43.073022	2.480079	701.388973	2.8e-06	1.7e-03	4.6e-02	7.4e-03
STAT3	Combined	25	12	412	1110	5.612864	2.794126	11.275170	4.8e-07	3.7e-04	6.9e-02	2.9e-02
DDX3X	Combined	60	32	393	1002	4.780534	3.064676	7.457071	2.1e-12	5.6e-09	0.53 (with Burkitt cohort), 0.001 (with extended BL cohort)	0.32 (with Burkitt cohort), 2.57e-04 (with extended BL cohort)
TNFα	Combined	15	130	388	963	0.286380	0.165674	0.495030	3.7e-07	3.3e-04	3.1e-03	3.6e-04
TP53	Combined	74	593	501	640	0.159411	0.121881	0.208497	4.6e-52	2.5e-48	7.9e-41	2.1e-42
CDKN2A	Combined	7	112	516	1065	0.128997	0.059680	0.278826	9.4e-12	1.7e-08	1.3e-03	2.5e-04

Table R3. Statistics for each virus-positive and virus-negative associated genes from their respective pooled (Combined) datasets.

Gene	Cancer	Mutated Virus+	Mutated Virus-	Not Mutated Virus+	Not Mutated Virus-	OR (>1 towards virus +)	OR lower	OR upper	Fisher's Exact P value	Adjusted P value (BH)	Unweighted Fisher Combined P value	wFisher Combined P value
DDX3X	gastric	0	3	27	314	1.6	0.095	39.70	1			
DDX3X	cervical	4	0	146	4	0.28	0.0098	4.88	1			
DDX3X	PBL	1	1	30	12	0.4	0.023	6.93	0.51			
DDX3X	Burkitt	51	10	43	16	1.90	0.78	4.61	0.19			
DDX3X	CHL	1	15	85	284	0.22	0.029	1.71	0.14			
DDX3X	HNSCC	3	3	62	372	6	1.18	30.40	0.044			
DDX3X	extended BL cohort	68	25	68	91	3.64	2.09	6.35	3.78E-06			
DDX3X	Combined	60	32	393	1002	4.78	3.06	7.46	2.1e-12	5.6e-09	0.53 (with Burkitt cohort), 0.001 (with extended BL cohort)	0.32 (with Burkitt cohort), 2.57e-04 (with extended BL cohort)

Table R4. Statistics for each cancer type included in the pooled (combined) dataset for *DDX3X*. Note that the combined *DDX3X* statistics represent the pooled dataset, which includes the Burkitt cohort (n=120), except for the last two columns. These columns show the combined p-values under two conditions: (1) with the Burkitt cohort included and (2) with the extended BL cohort included.

Addressing mutation calling discrepancies

We acknowledge the variability in mutation calling methods across datasets. To assess the potential impact of these differences, we compared mutation calls from cBioPortal (original mutation data generated by individual TCGA sequencing centers on GRCh37, as used in our manuscript) with those processed by the GDC method (mutations called on GRCh38 using four variant callers: MuTect, VarScan2, MuSE, and Pindel) for four TCGA cohorts included in our analysis: STAD (gastric cancer), HNSC (head and neck squamous carcinoma), CESC (cervical cancer), and LIHC (hepatocellular carcinoma). The comparison revealed a strong correlation in total mutation load (Fig. R7) and overlapping single nucleotide polymorphisms (SNPs) in driver genes (Table R5) between the cBioPortal and GDC datasets. This suggests that, despite differences in mutation-calling methods and filtering, these variations may have minimal impact on the overall mutation trends observed in our study.

Fig. R7. Comparison of total somatic SNP counts between GDC and cBioPortal data for A) Gastric Cancer, B) Head and Neck Cancer, C) Cervical Cancer, and D) Hepatocellular Carcinoma. *r*, Pearson correlation coefficient.

Cohort	Pearson correlation coefficient	P-value
TCGA_STAD	0.99	0
TCGA_HNSC	0.94	7.2e-235
TCGA_CESC	0.96	1.0e-87
TCGA_LIHC	0.91	1.6e-68

Table R5. Comparison of somatic SNP counts in driver genes between GDC and cBioPortal data.

Discussion on study limitations

To address the current limitations of our approach in combining cancer types to identify virus-associated genes, we have updated the discussion section as follows.

Included in Discussion section) *“Lastly, while our findings on EIF4A1 and DDX3X remain robust and underscore the potential importance of these RNA helicases in viral positive tumors, they are derived from a limited number of cancer types. Future studies incorporating larger and more diverse cohorts will help expand our understanding and validate their roles across a broader range of malignancies.”*

Comment 5) *“How were data sets selected? It is of course difficult to be exhaustive, but much broader whole genome sequencing data sets are available, e.g. ICGC data for HCC.”*

Response 5) We thank the reviewer for this valuable suggestion. We selected datasets based on the availability of comprehensive genomic information among the virus associated cancers. In the revised manuscript, we have included structural variant and copy-number signature analyses using whole-genome sequencing data from four virus-associated cancer types (cervical cancer, gastric cancer, HCC, and head and neck squamous cell carcinoma) from the ICGC/TCGA Pan-Cancer Analysis of Whole Genomes (PCAWG). For results, please refer to our response to comment 10 below.

Comment 6) *“Mutation burden analysis is shown as log₂ (nonsynonymous mutations) in Fig. 2. It would be more informative to provide data in mutations per megabase, allowing absolute comparisons across cancers.”*

Response 6) Fig. R8 displays mutations per mega base (TMB) for the cancer types. To account for highly hypermutated outliers in certain cohorts, we used log₂(TMB+1) values in the graphs. However, due to the use of customized panels targeting frequently mutated genes specific to each cohort, the resulting TMB values in targeted sequencing cohorts are elevated. For instance, the TMB for cHL in the targeted sequencing cohort was nearly double that of the WES cohort. This could potentially be misleading and limits the feasibility of making absolute comparisons across different cancer types. Based on this consideration, we have chosen to retain the original log₂-transformed nonsynonymous mutation counts in the y-axis for clarity.

Fig. R8. Mutations per mega base (TMB) for each cancer cohorts. P-value calculated by MWU test.

Comment 7) “Please include the list of driver genes considered for each cancer as supplementary table.”

Response 7) The list of driver genes is now included in supplementary Table S4 of the revised manuscript.

Comment 8) “The authors should thoroughly compare their results with previous work to state what is new and explain why previously described associations are missed. For example, the enrichment of TP53 mutations in HBV-related HCC (Amaddeo, Hepatology 2015), or the depletion of TP53 and CDKN2A alterations in HPV-positive HNSCC (Zapatka, Nat Genet 2020) are well known. Depletion of CTNNB1 mutations in HBV-related HCC (Amaddeo) or TERT alterations in HPV-positive HNSCC (Zapatka) are missed...”

Response 8) We thank the reviewer for raising these important points and for providing specific examples from previously published work. We have acknowledged these findings and clearly stated them in the revised manuscript. Additionally, we have included a note in the discussion section explaining why some well-known associations may not have been identified or were missed in our analysis. Please find our detailed modifications outlined below.

Modified in Results Section)

“Meanwhile, virus-negative tumors had higher odds of TP53 mutations (OR 8.20, 95% CI 4.79-8.20, $q=4.60e-52$), which were significant and known for most cancer types individually, including HNSCC¹, GC¹⁸, MCC¹⁹, BL²⁰, PBL²¹, among others. However, HCC exhibited the opposite trend, with TP53 mutations being significantly associated with HBV/HCV-positive HCC, as previously reported²². Mutations in CDKN2A were also more frequently observed in virus-negative tumors (OR 7.75, 95% CI 3.59-16.76, $q=9.42e-12$), and were significant in HNSCC individually, as noted in a prior study¹. Consistently, loss of 9p21.3/CDKN2A, CDKN1A were observed in virus-negative cancers (OR 4.96, 95% CI 3.65-6.73, $q=$

1.77e-25) (**Fig. S16C**). Lastly, *TNFAIP3* mutations were enriched in virus-negative cases (OR 3.49, 95% CI 2.02-6.04, $q=3.72e-7$), and were significant in cHL individually, as reported previously²³ (**Fig. 4A, S15, and Table S14**).”

Added in Discussion Section)

“In this study, we focused on the most significantly differentially mutated genes across all virus-positive versus virus-negative cancers, which may have led to the omission of well-established somatic mutation associations in specific cancer types. For instance, the depletion of CTNNB1 mutations in HBV-positive HCC cases, a known association^{1,22}, was not highlighted because the p-value for CTNNB1 mutations in the combined cohort of all eligible cancer types did not meet our significance threshold. In contrast, HCV-positive HCC cases are associated with CTNNB1 mutations²⁴, and grouping HBV- and HCV-positive cases together as virus-positive masked the HBV-specific trend. Similarly, TERT promoter mutations, well-documented in virus-positive HNSCCs^{1,25}, were not captured in our study because the analysis focused on exonic somatic mutations, excluding promoter regions.”

Comment 9) *“The authors state they performed mutational signature analysis “informed by de novo signature calling”. What does it mean? Did the authors identify new signatures? In Fig. 3 it is difficult to see the mutation counts for tumors with the lowest mutation burden. I would suggest cutting the axes to increase readability. Also in the boxplots please define “signature activity”.”*

Response 9) We thank the reviewer for the opportunity to clarify our approach and improve the figures. Our mutational signature analysis using Palimpsest consisted of two main steps:

1. **De novo (unsupervised) signature extraction:** We performed unsupervised non-negative matrix factorization (NMF) to extract de novo mutational signatures for each cancer type. These de novo signatures were then compared to known COSMIC v3 signatures to identify the most similar COSMIC signatures based on cosine similarity.
2. **Supervised signature fitting:** We subsequently used a supervised NMF approach, focusing on a refined set of COSMIC signatures identified in the first step as being most similar to the de novo signatures.

By “informed by de novo signature calling,” we meant that the initial de novo results guided the selection of COSMIC signatures included in the supervised fitting step. However, we re-analyzed the data using SigProfilerExtractor (v.1.1.24)²⁶ to perform de novo signature extraction and decompose them into COSMICv3.4 signatures. The results were similar to those obtained with Palimpsest. Most of the signatures identified by Palimpsest were also detected by SigProfilerExtractor, and both the direction (i.e., whether virus-positive or -negative samples had a higher proportion of mutations in each signature) and the level of significance were very similar (Table R6). For consistency with our structural variation and copy number signature analyses, which were also conducted using SigProfilerExtractor, we have chosen to present and use the SigProfilerExtractor results for mutational signature analysis in our updated manuscript. Results, figures, and methods are updated accordingly.

Regarding the mutational signature figures which now shows results from SigProfilerExtractor, we have updated the visualizations based on the reviewer’s suggestions. The updated figures now display relative

signature contribution (%) at the bottom and the total number of mutations at the top to improve readability. We have also replaced the term "signature activity" with "relative signature contribution (%)" for clarity (Fig. R9, R10). Fig. R9 and Fig. R10 is now included as Fig. 3 and Fig. S5 in the revised manuscript, respectively.

Signature	Palimpsest		SigProfilerExtractor	
	direction	Wilcoxon_ranksum_p	direction	Wilcoxon_ranksum_p
SBS3_HNSCC	higher in virus negative	4.70831E-09		
DBS1_MCC	higher in virus negative	0.000542058	higher in virus negative	7.71082E-05
SBS15_GC	higher in virus negative	0.003387619	higher in virus negative	0.012578035
SBS7b_MCC	higher in virus negative	0.005107194	higher in virus negative	0.000907045
ID7_GC	higher in virus negative	0.0071948	higher in virus negative	0.02291929
ID1_GC	higher in virus negative	0.014997676	higher in virus negative	0.026676527
ID2_GC	higher in virus negative	0.032323036	higher in virus negative	0.060021893
SBS45_HNSCC	higher in virus negative	0.035164661	higher in virus negative	0.033055838
ID3_HNSCC	higher in virus negative	0.042422919		
SBS7a_MCC	higher in virus negative	0.070003179	higher in virus negative	0.013045501
SBS6_HCC	higher in virus negative	0.122073766		
SBS40_HCC	higher in virus negative	0.127387733		
DBS2_HNSCC	higher in virus negative	0.12977748	higher in virus negative	0.075468469
SBS20_GC	higher in virus negative	0.141163209	higher in virus negative	0.809561888
ID2_CC	higher in virus negative	0.152776644	higher in virus negative	0.330767687
DBS4_HNSCC	higher in virus negative	0.176184771	higher in virus negative	0.148040278
SBS21_GC	higher in virus negative	0.27633481	higher in virus negative	0.839367232
ID5_BL	higher in virus negative	0.289578841		
SBS2_MCC	higher in virus negative	0.292334552		
DBS9_GC	higher in virus negative	0.321689434		
SBS40_PBL	higher in virus negative	0.35453948		
DBS1_CC	higher in virus negative	0.362520759		
SBS5_BL	higher in virus negative	0.372755503	higher in virus negative	0.204121407
ID2_HNSCC	higher in virus negative	0.385303924		
DBS9_CC	higher in virus negative	0.415815157		
SBS6_BL	higher in virus negative	0.420781234		
ID8_PBL	higher in virus negative	0.48745329		
SBS15_CC	higher in virus negative	0.487942581	higher in virus negative	0.617183459
ID8_HCC	higher in virus negative	0.521408175	higher in virus negative	0.405462061
SBS84_BL	higher in virus negative	0.594782083	higher in virus negative	0.298396713
SBS17b_GC	higher in virus negative	0.676372773	higher in virus negative	0.871769236
DBS1_HNSCC	higher in virus negative	0.753610605	higher in virus positive	0.770202592
ID2_PBL	higher in virus negative	0.816961322	higher in virus negative	0.48745329
DBS2_HCC	higher in virus negative	0.855101298		
DBS9_BL	higher in virus negative	0.93126201		
SBS4_CC	higher in virus negative	0.938229358		
DBS5_BL	higher in virus negative	0.961779496		
SBS2_HNSCC	higher in virus positive	6.22405E-06	higher in virus positive	0.000181837
SBS5_MCC	higher in virus positive	0.000111101	higher in virus positive	0.002913912
SBS1_HNSCC	higher in virus positive	0.000600509	higher in virus positive	0.001460776
SBS1_MCC	higher in virus positive	0.000604362	higher in virus positive	0.035716058
SBS24_HCC	higher in virus positive	0.005741363	higher in virus positive	0.066016865
SBS40_GC	higher in virus positive	0.007178642		
ID5_GC	higher in virus positive	0.024258799	higher in virus positive	0.012366693
SBS17b_BL	higher in virus positive	0.06366359		
ID3_HCC	higher in virus positive	0.070310894	higher in virus positive	0.277983931
ID8_CC	higher in virus positive	0.100402599		
SBS26_HCC	higher in virus positive	0.179199558		
SBS2_CC	higher in virus positive	0.201011164	higher in virus positive	0.237282435
SBS1_GC	higher in virus positive	0.265874302	higher in virus positive	0.127314855
SBS46_BL	higher in virus positive	0.266241925		
SBS13_MCC	higher in virus positive	0.279062458	higher in virus positive	0.99061269
SBS6_CC	higher in virus positive	0.338532109		
SBS1_CC	higher in virus positive	0.391817781	higher in virus positive	0.442959274
SBS22_HCC	higher in virus positive	0.392605584		
ID8_HNSCC	higher in virus positive	0.439306092		
SBS13_CC	higher in virus positive	0.478272364	higher in virus positive	0.542962124
SBS15_PBL	higher in virus positive	0.48745329	higher in virus positive	0.417886964
DBS11_BL	higher in virus positive	0.502291337	higher in virus positive	0.125162496
ID8_GC	higher in virus positive	0.551612202	higher in virus positive	0.528281601
ID15_GC	higher in virus positive	0.562704236		
DBS7_BL	higher in virus positive	0.618220052		
DBS4_HCC	higher in virus positive	0.631081031		
ID1_BL	higher in virus positive	0.722879775	higher in virus positive	0.961779496
DBS1_BL	higher in virus positive	0.773711285		
ID17_GC	higher in virus positive	0.78118271	higher in virus positive	0.867041473

SBS10b_GC	higher in virus positive	0.850532803	higher in virus negative	0.74455586
DBS4_GC	higher in virus positive	0.86054859	higher in virus positive	0.854065121
ID2_BL	higher in virus positive	0.923646643	higher in virus negative	0.431927028
DBS9_HCC	higher in virus positive	0.937465197		
DBS5_HCC	higher in virus positive	0.943923037		
SBS7b_HNSCC	higher in virus positive	0.983353352	higher in virus negative	0.511313572

Table R6. Mutational signature identification and comparison results using the Palimpsest method and the SigProfilerExtractor method. The proportion of mutations attributed to each signature and cohort is compared between virus-positive and virus-negative cases using both methods. The direction of the statistical test is indicated in red (higher in virus-positive) or blue (higher in virus-negative). Empty rows under SigProfilerExtractor indicate signatures identified exclusively through the Palimpsest method and not by SigProfilerExtractor. Signatures highlighted in yellow correspond to those shown in Fig. 3.

Fig. R9. Mutation signatures in virus-associated cancers: A,B) MCC (n=71), C,D) GC (n=436), E,F) HNSCC (n=487). Total mutations (top bar plot) and proportion of mutations associated with each signature (bottom stacked plot) in virus-positive compared to virus-negative cases. Signatures identified from SigProfilerExtractor are shown. Signature names with SBS96, DBS78, or ID83 followed by a capital letter are de novo mutational signatures identified from the cancer cohort which could not be decomposed into known COSMIC(v.3.4) signatures. The rest are decomposed COMSIC signatures. Schematic representation of the effect of the absence of processes behind key mutation signatures in virus-associated cases is shown in B) MCC (MCPyV-positive).

Fig. R10. Mutation signatures in virus-associated cancers: A) CC (n=172), B) BL (n=91; 68 EBV-positive eBL, 6 EBV-negative eBL, 3 EBV-positive sBL, 14 EBV-negative sBL), C) HCC (n=190), D) PBL (n=23).

Comment 10) “In addition to mutational signatures, it would be a great addition to study structural variant and copy-number signatures, which were previously associated with viral insertions.”

Response 10) We thank the reviewer for the great suggestion. We performed structural variant and copy-number signature analyses using SigProfilerExtractor²⁶, following the same methodology as in SBS analysis. This analysis was performed on WGS samples from HCC, HNSCC, CC, and GC obtained from PCAWG, as well as BL and cHL (Fig. R11–12). Among SV signatures, no significant differences were observed between virus-positive and virus-negative tumors, except for an enrichment of the SV2 signature, which consists of non-clustered translocations (COSMIC v3.4), in HPV-positive HNSCC ($q = 0.0025$, Fig. R11G). In the CN signature analysis, several de novo signatures that could not be decomposed into known COSMIC CN signatures were identified as differing in multiple cancer types (Fig. R12). Fig. R11-12 are now included as Fig. S9-10 in the modified manuscript.

Added in Results Section) “To identify differences in structural variant (SV) and copy number (CN) signatures between virus-positive and virus-negative tumors, we conducted a signature analysis using SigProfilerExtractor²⁶, following the same methodology as in SBS analysis. This analysis was performed on WGS samples from HCC, HNSCC, CC, and GC obtained from PCAWG, as well as BL and cHL (Fig. S9–10 and Table S11). Among SV signatures, no significant differences were observed between virus-positive and virus-negative tumors, except for an enrichment of the SV2 signature, which consists of non-clustered translocations (COSMIC v3.4), in HPV-positive HNSCC ($q = 0.0025$, Fig. S9G). In the CN signature analysis, several de novo signatures that could not be decomposed into known COSMIC CN signatures were identified as differing in multiple cancer types (Fig. S10).”

A**B****C**

Fig. S11. Structure variant signatures in A) hepatocellular carcinoma (HCC), B) burkitt lymphoma (BL), C) classical hodgkin lymphoma (cHL), D) gastric cancer (GC), E) cervical cancer (CC), F) head and neck squamous cell carcinoma (HNSCC). G) Significant structure variant signature in HNSCC. Q value by MWU test.

Fig. R12. Copy number signatures in A) GC, B) CC, C) cHL, D) HCC, E) HNSCC, F) BL. G-I) Significant copy number signatures in G) HCC, H) HNSCC, and I) BL. Q value by MWU test.

In addition to SV and CNV signature analysis, we compared differences in chromosomal instability (CIN) signatures between virus-positive and virus-negative tumors using CIN signature data from a recent study on TCGA cohorts, including HCC, HNSCC, CC, and GC²⁷. Notably, the CX1 signature, which involves whole-arm or whole-chromosome changes and is associated with chromosome missegregation due to defective mitosis and/or telomere dysfunction²⁷, showed a significantly higher relative contribution in virus-

positive cases compared to virus-negative cases in HNSCC, GC, and HCC ($p=7.9e-6$, $1.3e-4$, 0.0032 , respectively; Fig. R13–14). Although CC exhibited a similar trend, the difference was not statistically significant, likely due to the small number of HPV-negative cases ($n = 4$, Fig. R14). This observation is consistent with studies showing that HPV’s E6 and E7 proteins disrupt normal cell cycle regulation and mitotic progression, leading to chromosomal abnormalities²⁸. Fig. R13-14 are included as Fig. S11-12 in the modified manuscript.

Added in Results Section) “Furthermore, we compared differences in chromosomal instability (CIN) signatures between virus-positive and virus-negative tumors using CIN signature data from a recent study on TCGA cohorts, including HCC, HNSCC, CC, and GC (Table S3)²⁷. Notably, the CX1 signature, which involves whole-arm or whole-chromosome changes and is associated with chromosome missegregation due to defective mitosis and/or telomere dysfunction²⁷, showed a significantly higher relative contribution in virus-positive cases compared to virus-negative cases in HNSCC, GC, and HCC ($p=7.9e-6$, $1.3e-4$, 0.0032 , respectively; Fig. S11–12, Table S12–13). Although CC exhibited a similar trend, the difference was not statistically significant, likely due to the small number of HPV-negative cases ($n = 4$, Fig. S12). This observation is consistent with studies showing that HPV’s E6 and E7 proteins disrupt normal cell cycle regulation and mitotic progression, leading to chromosomal abnormalities²⁸.”

Fig. R13. Chromosomal instability signatures in A) HNSCC. B-F) Significant chromosomal instability signatures in HNSCC. P value by MWU test.

Fig. R14. Chromosomal instability signatures in A) GC, B) HCC, C) CC. B-D) Significant chromosomal instability signatures in GC. F-H) Significant chromosomal instability signatures in HCC. J) Significant chromosomal instability signatures in CC. P value by MWU test.

Comment 11) “Again, several results of the mutational signature analysis, e.g. association of SBS24 with HBV-positive HCC (Schulze, *Nat Genet* 2015) or association of SBS2 with HPV-positive HNSCC (Zapatka, *Nat Genet* 2020) so it should be clearly stated what results are new to this study.”

Response 11) We thank the reviewer for pointing this out. We have referenced previous studies, including the association of SBS24 with HBV-positive HCC²⁹ and SBS2 with HPV-positive HNSCC¹, wherever relevant. To highlight new findings from our study, we have used phrases such as “notably” and “we found” to make the distinction clear.

Modified in Results Section) “To detect and quantify the relative contribution of COSMIC mutation signatures³⁰ within the virus-associated cancers, we applied SigProfilerExtractor²⁶ to extract de novo mutational signatures and decompose them into COSMIC signatures across 7 available cancers. Virus-positive tumors exhibited different activities of mutation signatures compared to virus-negative tumors of the same cancer type (**Fig. 3, S5 and Table S7**). UV light is known to be the major etiological agent of MCC tumors in the absence of viral infection. Accordingly, the relative contribution of mutations attributed to SBS7a/7b/UV-light was lower in MCPyV-positive compared to MCPyV-negative, as reported previously³¹ (**Fig. 3A,B and Table S7**).

In GC, the proportion of mutations associated with SBS15/mismatch-repair-(MMR)-deficiency was higher in EBV-negative than EBV-positive cases (**Fig. 3C,D and Table S7**). Microsatellite instability (MSI), as assessed by standard methods, is a defining characteristic of a GC subtype that is exclusively EBV-negative and comprised 73/406 (18%) of EBV-negative patients in the TCGA cohort. Accordingly, the relative contribution of mutations attributed to SBS15 and SBS20/MMR-deficiency were higher in the conventionally defined MSI subtype compared to other EBV-negative cases (**Fig. S6 and Table S8**). This suggests that the difference in MMR signatures between EBV-positive and EBV-negative GC are largely driven by the MSI subtype.

In HNSCC, HPV-positive HNSCC had higher relative contribution of SBS2/APOBEC mutations (**Fig. 3E,F and Table S7**), as reported in previous study¹. This is consistent with the hypothesis that HPV oncoproteins may increase APOBEC3A and APOBEC3B expression and mutagenic activity^{32,33}. In a multivariate analysis of HPV-positive HNSCC, which accounted for sex, HPV strain, age, and total mutation count, HPV33-associated HNSCCs showed significantly lower SBS2/APOBEC mutation counts, while total mutation count was positively associated with SBS2/APOBEC mutation counts, both findings consistent with previous reports^{34,35} (coefficient = -1.37 and 0.012, $p = 0.003$ and <0.001 , respectively; **Table S9, Fig. S7**).

The mutation counts associated with the SBS5/clock-like signature were higher in HPV-negative cervical cancer (CC) compared to HPV-positive CC ($p=0.013$, **Table S7**), consistent with our observation that total mutation counts are positively correlated with age and tend to be higher in HPV-negative cases. Notably, the levels of SBS2/13/APOBEC-associated signatures varied between HPV strains, with HPV31 significantly exhibiting the highest mutation counts (**Fig. S8 and Table S8**). A similar trend observed in HNSCC was also apparent in CC, where the HPV33 strain exhibited lower APOBEC signature levels compared to HPV16, although this difference was not statistically significant in CC. These strain-specific differences aligned with our findings on total mutation load. Importantly, these differences appear to be primarily driven by variations in total mutation load, as HPV strain-specific effects were no longer significant in a multivariable model that accounted for virus strain, virus status, total nonsynonymous mutation load, and age. In this model, virus-positive status and total nonsynonymous mutation load were positively associated with APOBEC signature mutations (**Table S10**), consistent with previous findings³⁶.

In BL exomes²⁰ (**Fig. S5B**), a de novo mutational signature, SBS96B, was identified. Although SBS96B did not meet the cosine similarity threshold for decomposition into known COSMIC signatures, potentially related COSMIC signatures included SBS46, SBS5, SBS17, and SBS15. This signature showed a higher proportion in EBV-positive BL samples compared to EBV-negative BL samples (**Table S7**), consistent with the original genome-wide mutation signature analysis on SBS5, SBS17, and SBS15 in the same cohort²⁰. However, in partial contrast to the original analysis, we did not detect the SBS9/pol η signature, which is associated with non-canonical AID activity and was previously identified in BL genomes²⁰.

In HCC, there was no difference in absolute count of mutations attributed to mutation signatures (Fig. S5C), consistent with the similar mutation burden in virus-positive and -negative HCC overall. However, HCC tumors positive for HBV had a greater proportion of mutations due to SBS24/aflatoxin, an environmental carcinogen known to predispose to HBV-mediated cirrhosis, as observed in previous studies^{29,37,38} (p=0.010) (Table S8)."

Comment 12) *"The association of DDX3X and EIF4A1 mutations with virus-positive cancers seems strongly driven by only a couple of cancer types. The authors should either increase sample size to obtain stronger statistics in other cancers, or clearly state this limitation."*

Response 12) We thank the reviewer for the helpful comment. As noted, we observed a significant association between *DDX3X* mutations and EBV-positive BL (p=3.78e-6) by utilizing the extended Burkitt lymphoma dataset, which includes *DDX3X* mutation data from 252 cases aggregated from previously published studies. However, we acknowledge that *DDX3X* and *EIF4A1* mutations are observed in a limited number of cancer types. We have now included a clear statement in the revised manuscript to address this limitation.

Included in Discussion Section) *"Lastly, while our findings on EIF4A1 and DDX3X remain robust and underscore the potential importance of these RNA helicases in viral positive tumors, they are derived from a limited number of cancer types. Future studies incorporating larger and more diverse cohorts will help expand our understanding and validate their roles across a broader range of malignancies."*

Comment 13) *"The association of viral status with immune checkpoint inhibitor response is valuable. However I don't understand the strategy used to determine whether viral status is independent from PD-L1 expression. Why not use directly the data shown in Fig. 5A to see if viral status and PD-L1 expression are significant in multivariate models?"*

Response 13) We thank the reviewer for the suggestion. We agree that a multivariate approach using both PD-L1 expression and viral status would further confirm that the association of viral status with ICI response still holds after accounting for PD-L1 expression. However, the data analyzed in Fig. R15A (Fig. 5A in manuscript) is sourced from clinical trials, for which individual patient information is not available. Therefore, we can only make comparisons of ICI response at the cohort-level, and a multivariate model with all three patient variables (TMB, PD-L1 expression and viral status) is not possible. Hence the TCGA analysis in Fig. R15B (Fig. 5B in manuscript), which shows higher PD-L1 expression in viral GC versus non-viral GC, and no difference in expression between viral and non-viral tumors in HCC and HNSCC.

Fig. R15. Analysis of biomarkers for immunotherapy response in virus-associated cancers. A) Odds ratio of positive response to treatment with PD-1/PD-L1 inhibitors with virus-positive status, PD-L1 positive status, and/or high tumor mutation burden (TMB) in 32 studies representing four types of cancer, Fisher's exact test. B) $\text{Log}_2(\text{RSEM}+1)$ expression of *PD-L1* (*CD274*). P value by MWU test. Lines show median and quartile distributions (upper: 75th, lower: 25th percentiles).

Reviewer #2 (Remarks to the Author): Expert in cancer genetics and genomics, biostatistics, virus-associated cancers, and epidemiology

Reviewer's summary) *"The paper examines the genomic characteristics of virus-positive and virus-negative tumors across nine different cancer types. The primary objective is to delineate the differences in genomic features, such as mutation frequencies, between these two groups. The study builds on over a century of research into tumor viruses, aiming to provide new insights into the genomic landscapes shaped by viral infections. However, its impact is diminished by the lack of clarity regarding novel findings and insufficient citation of previous works."*

Response) We thank the reviewer for the thoughtful summary. We have ensured extensive citation of relevant previous works and have clarified the novel findings throughout the revised manuscript. In the responses below, we address these concerns in detail and outline the revisions made.

Major comments)

Comment 1) *"Cite additional previous works to provide context and support. Additionally, clarify whether the findings are novel. Without this clarification, it is difficult to gauge the contribution of this work. Many results seem have been reported previously. For example, TCGA GC study [PMID: 25079317] has reported that "most EBV-positive cases were male (81%, P = 0.037), as previously reported [PMID: 19445939]." Immunotherapy Efficacy in Virus-associated Cancers have been reviewed [EJMO 2022;6(2):100–110]. Additionally, virus-associated Merkel cell carcinomas are known to carry a low mutational burden [PMID: 26238782]. It may be beneficial to comment more frequently if a finding from this work agrees with previous work or is a novel finding. Without this clarification, it is challenging to assess the unique contributions of the study. This ambiguity undermines the paper's impact and may lead to the perception that it offers little new information."*

Response 1) We thank the reviewer for this valuable suggestion. In response, we have added additional citations throughout the manuscript to provide proper context and support for our findings. To clearly highlight the new contributions of our work, we have used phrases such as “notably” and “we found” to distinguish new findings from those already established in the literature. Below are examples of specific sections in the manuscript that have been revised to address this comment.

Modified in Results Section)

“In order to illustrate other common demographic characteristics of virus-associated malignancies, we analyzed data from the Global Cancer Observatory (GLOBOCAN 2020)³⁹ and published incidence rates in 48 studies of 11 cancer types linked to 5 viruses and 13 non-virus-associated cancers⁴⁰⁻¹²¹. First, we compared the number of viral cancers in males versus females (M/F) reported in select published studies⁴⁰⁻¹²¹ (Fig. 1A, Table S1). We found that the M/F ratio was greater overall in virus-associated cancers compared to nonviral cancers ($p=2.2e-16$, Fisher’s exact test). Among studies that reported M/F ratio for virus-positive and virus-negative tumors specifically, virus-positive cases tended to have a greater M/F ratio than virus-negative cases ($p=5.23e-8$, Fig. 1B, Table S2). This trend was consistent in gastric cancer (GC, $p=1.04e-10$) and HL ($p=0.015$), both of which have been reported previously^{18,122,123}. In contrast, no difference in the M/F ratio and viral status was observed for BL, despite higher M/F incidence ratios of 2:1 to 4:1 being reported in both eBL and sBL¹²⁴. A lower M/F ratio was observed in MCPyV-positive Merkel cell carcinoma (MCC) compared to virus-negative MCC, as noted previously¹²⁵. Digital papillary adenocarcinoma, which has been recently associated to HPV42, is reported to be more frequent in males compared to females at a ratio of 4:1¹²⁶.”

“In general, virus-positive tumors had a lower count of nonsynonymous mutations than virus-negative tumors, as reported in individual cancer types^{21,127-131} (Fig. 2A, 2B and Table S3, S5), including in particular PCNSL¹²⁹ (Wilcoxon test $p=1.4e-7$), cHL (targeted, $p=2.4e-6$; WES, $p=0.032$; WGS, $p=0.045$), PBL²¹ ($p=0.044$), HNSCC¹³⁰ ($p=6.5e-6$) and, as a trend, GC¹³¹ ($p=0.086$), CC ($p=0.17$), and MCC¹²⁷ ($p=0.072$) (Fig. 2A).”

“In CC, there was no significant difference in mutation load between squamous carcinoma and adenocarcinoma subtypes. However, age at diagnosis positively correlated with mutation load (Spearman’s $\rho = 0.31$, $p = 4.1E-05$, Table S6), consistent with a previous study¹³². Notably, multiple HPV strains in CC differed significantly in mutation load, with HPV31 exhibiting the highest levels (Fold change = 7.07, $p = 0.02$, Fig. S3, Table S6). To account for age, HPV strains, viral status, on mutation load, we used generalized linear model regression model (GLM) and found that HPV31 and virus-positive status remained significantly associated with elevated mutation load (coefficient = 1.81 and 1.09, $p = 0.013$ and <0.001 , respectively).”

“Using results of CIBERSORT^{133,134} deconvolution of the same TCGA samples, we found evidence of higher CD8+ T cell infiltration in EBV-positive versus EBV-negative GC (median 0.19 and 0.09, $p=7.4e-09$), and in HPV-positive versus HPV-negative HNSCC (median 0.13 and 0.07, $p=4.3e-10$), but not between virus-positive and virus-negative HCC (Fig. 5C). These results were replicated in all other deconvolution approaches tested¹³⁵⁻¹³⁸ (Fig. S18). Similarly, analysis of TRUST4^{139,140}-extracted T cell receptors showed higher T cell β receptor clonal selection, measured by counts per clonotypes per thousand reads (CPK), in both virus-positive GC (median 185.16 and 310.73, $p=7.8e-10$) and HNSCC (median 345.72 and 444.44, $p=0.017$) but not in HCC (Fig. 5D). We observed similar trends for the α chain, as well as with the metric of clonality (Fig. S19).”

We thank the reviewer for sharing the article *Review of Immunotherapy Efficacy in Virus-associated Cancers*¹⁴¹. While the review highlights some clinical trials we used in our analysis, the authors do not provide a statistical evaluation of virus positivity as a marker for immunotherapy response. In our study, we aggregated the results from 33 immunotherapy clinical trial cohorts. Focusing on responses to PD-1 and PD-L1 inhibitors evaluated by RECIST 1.1, we directly compared the efficacy of these therapies in virus-positive versus virus-negative patients. We similarly analyzed the effects of PD-L1 expression and tumor mutational burden, to emphasize the predictive value of our novel biomarker in contrast to the two classic markers for immunotherapy response. We have now cited the review in the discussion section as outlined below.

Modified in Discussion Section) *“Analysis of ICI clinical trials revealed that virus-positive status could represent a positive biomarker for ICI therapy response in GC and HNSCC¹⁴¹. The improved response to immunotherapy of EBV-positive GC patients compared to EBV-negative GC patients is hypothesized to be due to increased expression of PD-L1, potentially through activation of the NF- κ B pathway by viral protein LMP2A¹⁴². However, the association between HPV infection and PD-L1 expression is less clear: some studies report a link between HPV status and PD-L1 expression^{143,144}, while others find no association^{145,146}, the latter of which is consistent with our results from the TCGA HNSCC dataset. In both GC and HNSCC, we observe a marked increase in CD8+ T cell infiltration as well as an increase in T cell receptor clonality in virus-positive tumors. The high immunogenicity of viral antigens expressed in these tumors might elicit a bigger and more clonal T cell response, which could be reactivated through ICI therapy.”*

Moreover, we believe that integrating multiple data sources provides a broader analysis of virus-associated cancers across tumor types, offering unique insights that individual tumor-type studies cannot achieve. Specifically, we analyzed 2,488 cancer samples encompassing all seven known oncoviruses and 14 virus-associated cancers, nine of which include both virus-positive and virus-negative tumors, enabling direct comparisons. This comprehensive approach uncovered common patterns and novel insights that extend beyond individual tumor types. From our analysis, we emphasize the following key findings:

1. **Higher Male Incidence:** Virus-associated cancers exhibit a significantly higher incidence in males than in females, exceeding the gender disparity observed in non-virus-associated cancers.
2. **Reduced Mutation Burden:** Virus-positive tumors generally harbor fewer nonsynonymous somatic mutations compared to virus-negative tumors of the same cancer type.
3. **Frequent RNA Helicase Mutations:** Virus-positive tumors frequently harbor mutations in RNA helicases, such as DDX3X and EIF4A1, many of which appear to reduce protein function in a sex-specific manner. These mutations are recurrently observed across various virus-related tumors, including natural killer/T-cell lymphoma, aggressive NK-cell lymphoma, and Burkitt lymphoma (BL), among others. Notably, by aggregating data from 252 BL cases, we identified a significant association between DDX3X mutations and EBV-positive BL, an association previously reported but not statistically significant.
4. **Enhanced Immunotherapy Response:** Virus-positive tumors, particularly gastric cancer (GC) and head and neck squamous cell carcinoma (HNSCC), demonstrate higher response rates to immune checkpoint inhibitors. These tumors also exhibit increased CD8+ T-cell infiltration and show evidence of T-cell receptor clonal expansion.

Comment 2) “*DDX3* and *eIF4A* results are aggregated across different cancer types. This approach can be misleading because it masks the variability and specific associations that may be present in individual cancer types. For example, *eIF4A* is only significantly elevated odds of mutation in virus-positive tumors in GC ($q=0.04$) but not in other cancer types. However, when combining cancer types, the adjusted *P* value becomes much more significant ($q= 3.46E-05$). Aggregating data could lead to incorrect conclusions about the generalizability of these findings across all cancer types. It’s of interest to examine if the association between *eIF4A* mutation and virus status is still significant after adjusting different cancer types as the confounder.”

Response 2) We thank the reviewer for the suggestion. In our analysis of genes recurrently mutated in virus-positive versus virus-negative tumors, we pooled cases from multiple cancer types for each gene to compare the overall mutation rate between virus-positive and virus-negative cases. To ensure that each cancer type contributed meaningfully to the combined results and to provide a clearer statistical framework, we applied two additional criteria for analyzing and selecting genes of interest:

1. A cancer type was included in the analysis only if the gene was mutated in at least one sample (virus-positive or virus-negative).
2. A gene was considered a candidate of interest only if the final combined direction was consistently supported by at least half of the included cancer types.

As a result of these additional criteria, *STAT3* (associated with virus-positive tumors) and *TNFAIP3* (associated with virus-negative tumors) have been included to recurrently mutated genes in virus-positive versus virus-negative tumors, alongside *EIF4A1* and *DDX3X* (virus-positive-associated) and *TP53* and *CDKN2A* (virus-negative-associated). To clarify which cancer types and how many cases were considered for each gene, we included per-cancer-type statistics in Fig. S14 and Fig. S15 (Fig. R16 and Fig. R17) and in Table S14.

The rationale for performing a pan-cancer analysis by combining all cancer cases was to uncover trends that may remain undetected in individual cancer types due to limited sample sizes. For example, when we analyzed all 603 cases together for the *EIF4A1* gene, the resulting p-value was more significant than in any single cancer type, driven by a collective signal across GC, BL, PBL, and CC, all of which showed higher mutation rates in virus-positive cases (Fig. R16A). Combining data across cancer types increases the overall sample size, which enhances the statistical power and reduces the variance in estimates of mutation rates. This approach allows consistent patterns across cancer types to contribute cumulatively to the overall signal, resulting in a lower p-value compared to analyses confined to individual cancer types. In this way, pan-cancer analysis highlights broader, shared trends that might otherwise be missed in cancer-specific studies.

To address the reviewer’s suggestion of accounting for cancer-type variability as a potential confounder, we conducted a p-value combination analysis for each gene, both with and without weighting by sample size (using Fisher’s method and *wFisher*¹⁷, respectively). The results remained statistically significant for most genes while the results for *DDX3X* required further validation (Table R7). By incorporating an extended Burkitt lymphoma dataset, which specifically includes *DDX3X* mutation data aggregated from 252 cases in previously published studies, we observed a significant association between *DDX3X* mutations and EBV-positive BL ($p=3.78e-6$). This association was not significant in the original BL cohort ($n=120$, $p=0.19$) used for the initial *DDX3X* analysis. When using p-values from the extended BL cohort to calculate

the combined p-value for the *DDX3X* gene, both unweighted and weighted methods produced significant results ($p=0.001$ and $p=2.57e-4$, respectively; Table R8). These findings are now included in Table S14 of the revised manuscript.

Moreover, to address the reviewer’s concern on masking the variability and specific associations that may be present in individual cancer types, a heatmap has been added to the right side of Fig. 4A to illustrate the mutation rate trends across cancer types and clarify which cancer types were included in the analysis (Fig. R5).

Finally, we have updated the discussion section to acknowledge the limitations of our approach on identifying virus-associated genes, as follows:

Included in Discussion section) “Lastly, while our findings on *EIF4A1* and *DDX3X* remain robust and underscore the potential importance of these RNA helicases in viral positive tumors, they are derived from a limited number of cancer types. Future studies incorporating larger and more diverse cohorts will help expand our understanding and validate their roles across a broader range of malignancies.”

Fig. R16. Combined and individual cancer-type trends in virus-positive-associated genes. A–C) Boxplots of bootstrapped ($n = 10,000$) one-sided Fisher’s exact p-values (direction: higher mutation rates in virus-positive cases) for each cancer type included in the combined analysis of each gene. Number of samples used in the analysis for each cancer type are shown in parenthesis. D–F) Differences in mutation proportions between virus-positive and virus-negative cases for each cancer type. * MWU test, $p < 0.05$.

Fig. R17. Combined and individual cancer-type trends in virus-negative-associated genes. A–C) Boxplots of bootstrapped ($n = 10,000$) one-sided Fisher's exact p-values (direction: higher mutation rates in virus-negative cases) for each cancer type included in the combined analysis of each gene. Number of samples used in the analysis for each cancer type are shown in parenthesis. D–F) Differences in mutation proportions between virus-positive and virus-negative cases for each cancer type. * MWU test, $p < 0.05$.

Fig. R18. Combined $\text{log}_{10}(\text{odds ratio})$ of mutation in genes associated with virus-positive (top) and virus-negative (bottom) status ($q < 0.005$) from pooled data of 1,971 tumors across 9 virus-associated cancers. The heatmap on the right displays the cancer cohorts included in the pooled data for the calculation of each gene, with colors representing mutation rate trends in each cohort (red: higher in virus-positive; blue: higher in virus-negative) and shades indicating the two-sided Fisher's exact test p-value.

Gene	Cancer	Mutated Virus+	Mutated Virus-	Not Mutated Virus+	Not Mutated Virus-	OR (>1 towards virus +)	OR lower	OR upper	Fisher's Exact P value	Adjusted P value (BH)	Unweighted Fisher Combined P value	wFisher Combined P value
EIF4A1	Combined	15	0	246	342	43.073022	2.480079	701.388973	2.8e-06	1.7e-03	4.6e-02	7.4e-03
STAT3	Combined	25	12	412	1110	5.612864	2.794126	11.275170	4.8e-07	3.7e-04	6.9e-02	2.9e-02
DDX3X	Combined	60	32	393	1002	4.780534	3.064676	7.457071	2.1e-12	5.6e-09	0.53 (with Burkitt cohort), 0.001 (with extended BL cohort)	0.32 (with Burkitt cohort), 2.57e-04 (with extended BL cohort)
TNFAIP3	Combined	15	130	388	963	0.286380	0.165674	0.495030	3.7e-07	3.3e-04	3.1e-03	3.6e-04
TP53	Combined	74	593	501	640	0.159411	0.121881	0.208497	4.6e-52	2.5e-48	7.9e-41	2.1e-42
CDKN2A	Combined	7	112	516	1065	0.128997	0.059680	0.278826	9.4e-12	1.7e-08	1.3e-03	2.5e-04

Table R7. Statistics for each virus-positive and virus-negative associated genes from their respective pooled (Combined) datasets.

Gene	Cancer	Mutated Virus+	Mutated Virus-	Not Mutated Virus+	Not Mutated Virus-	OR (>1 towards virus +)	OR lower	OR upper	Fisher's Exact P value	Adjusted P value (BH)	Unweighted Fisher Combined P value	wFisher Combined P value
DDX3X	gastric	0	3	27	314	1.6	0.095	39.70	1			
DDX3X	cervical	4	0	146	4	0.28	0.0098	4.88	1			
DDX3X	PBL	1	1	30	12	0.4	0.023	6.93	0.51			
DDX3X	Burkitt	51	10	43	16	1.90	0.78	4.61	0.19			
DDX3X	CHL	1	15	85	284	0.22	0.029	1.71	0.14			
DDX3X	HNSCC	3	3	62	372	6	1.18	30.40	0.044			
DDX3X	extended BL cohort	68	25	68	91	3.64	2.09	6.35	3.78E-06			
DDX3X	Combined	60	32	393	1002	4.78	3.06	7.46	2.1e-12	5.6e-09	0.53 (with Burkitt cohort), 0.001 (with extended BL cohort)	0.32 (with Burkitt cohort), 2.57e-04 (with extended BL cohort)

Table R8. Statistics for each cancer type included in the pooled (combined) dataset for *DDX3X*. Note that the combined *DDX3X* statistics represent the pooled dataset, which includes the Burkitt cohort (n=120), except for the last two columns. These columns show the combined p-values under two conditions: (1) with the Burkitt cohort included and (2) with the extended BL cohort included.

Comment 3) “Investigate cancer subtypes. Cancer is a heterogeneous disease, and different subtypes can have distinct genomic characteristics and responses to viral infections. By not analyzing subtypes, the study may overlook important differences.”

Response 3) We thank the reviewer for the great suggestion. For each of the features discussed in the manuscript, including mutation load, mutational signatures, chromosomal instability signatures, and somatic mutations in six genes recurrently found in virus-positive and virus-negative cases, we examined subtype differences and discussed them throughout the manuscript. One observation on subtype differences was in gastric cancer, specifically regarding the SBS15/mismatch-repair-(MMR)-deficiency signature. We have found significantly lower SBS15 mutation fractions and counts in virus-positive compared to virus-negative GC. Upon further investigation into GC subtypes, we observed that the high SBS15 mutation signature was primarily driven by the MSI subtype (Fig. R19). The corresponding statistical results have been included in Tables S6, S8, S13, and S16 of the revised manuscript.

Fig. R19. Gastric cancer subtype differences in A) fractions (%) and B) $\log_2(\text{mutation counts}+1)$ of SBS15/MMR-deficiency signature. * MWU test, $p < 0.05$.

Modified in Results Section) “Additionally, we observed that several cancer subtypes and virus strains differed in mutation load (**Table S6**). The NS subtype, the most common subtype of cHL in our datasets (targeted, 84.6%, 204/241; WES, 73.4%, 47/64), exhibited the highest mutation load relative to other subtypes in both targeted and WES cohorts ($p = 1.6E-06$ and 0.010 , respectively). However, after adjusting for subtypes, sex, and age, only EBV status remained significantly associated with a lower mutation load in the targeted and WES cohorts, consistent with a previous report¹²⁸ (coefficient = -0.44 , -2.1 ; $p = 0.009$, 0.001). As expected²⁰, eBL had a higher total mutation load compared to sBL, reflecting a similar trend with EBV status ($p = 0.00029$). Since EBV-positivity was highly associated with eBL ($p = 7.68e-12$), when both subtype and EBV status were considered simultaneously, neither variable showed a significant independent association with total mutation load.

In CC, there was no significant difference in mutation load between squamous carcinoma and adenocarcinoma subtypes. However, age at diagnosis positively correlated with mutation load (Spearman’s $\rho = 0.31$, $p = 4.1E-05$, **Table S6**), consistent with a previous study¹³². Notably, multiple HPV strains in CC differed significantly in mutation load, with HPV31 exhibiting the highest levels (Fold change = 7.07 , $p = 0.02$, **Fig. S3**, **Table S6**). To account for age, HPV strains, viral status, on mutation load, we used generalized linear model regression model (GLM) and found that HPV31 and virus-positive status remained significantly associated with elevated mutation load (coefficient = 1.81 and 1.09 , $p = 0.013$ and < 0.001 , respectively).

In HNSCC, females exhibited higher driver gene mutation counts than males ($p = 0.0058$), and age correlated positively with both total and driver mutation counts (Spearman’s $\rho = 0.19$ and 0.27 , $p = 2.4E-05$ and $1.2E-09$, respectively; **Table S6**), consistent with previous findings¹⁴⁷. Unlike CC, there were no

significant differences in mutation load between HPV strains in HNSCC (**Fig. S4**), as reported in a prior study³⁴.

In HCC, no difference in mutation load was evident between HBV and HCV related HCC. Neither Activated B-cell (ABC) nor Germinal Center B-cell (GCB) subtypes of PCNSL differed in their mutation loads.”

*“In GC, the proportion of mutations associated with SBS15/mismatch-repair-(MMR)-deficiency was higher in EBV-negative than EBV-positive cases (**Fig. 3C,D** and **Table S7**). Microsatellite instability (MSI), as assessed by standard methods, is a defining characteristic of a GC subtype that is exclusively EBV-negative and comprised 73/406 (18%) of EBV-negative patients in the TCGA cohort. Accordingly, the relative contribution of mutations attributed to SBS15 and SBS20/MMR-deficiency were higher in the conventionally defined MSI subtype compared to other EBV-negative cases (**Fig. S6** and **Table S8**). This suggests that the difference in MMR signatures between EBV-positive and EBV-negative GC are largely driven by the MSI subtype.”*

*“Notably, we found that DDX3X mutation was strongly (though not exclusively) associated with EBV-positive status ($p=3.78e-6$), which has been discussed in previous studies but without statistical significance^{20,148-152}, and endemic subtype ($p=4.25e-7$, **Table S14,S16**).”*

Comment 4) *“Investigate WGS samples. The study seems limited to WES and targeted sequencing. WGS provides a more comprehensive view of the genome, including non-coding regions and large structural changes that WES and targeted sequencing might miss. Including WGS samples (e.g., ones from PCAWG [PMID: 32025001]) could examine chromosomal instability (CIN) and structural variants (SV) between virus-positive and virus-negative tumors.”*

Response 4) We thank the reviewer for the suggestion. We first performed structural variant and copy-number signature analyses using SigProfilerExtractor²⁶. This analysis was performed on WGS samples from HCC, HNSCC, CC, and GC obtained from PCAWG, as well as BL and cHL (Fig. R20–21). Among SV signatures, no significant differences were observed between virus-positive and virus-negative tumors, except for an enrichment of the SV2 signature, which consists of non-clustered translocations (COSMIC v3.4), in HPV-positive HNSCC ($q = 0.0025$, Fig. R11G). In the CN signature analysis, several de novo signatures that could not be decomposed into known COSMIC CN signatures were identified as differing in multiple cancer types (Fig. R21). Fig. R20-21 are now included as Fig. S9-10 in the modified manuscript. In addition, as suggested by the reviewer, we compared differences in chromosomal instability (CIN) signatures between virus-positive and virus-negative tumors using CIN signature data from a recent study on TCGA cohorts, including HCC, HNSCC, CC, and GC²⁷. Notably, the CX1 signature, which involves whole-arm or whole-chromosome changes and is associated with chromosome missegregation due to defective mitosis and/or telomere dysfunction²⁷, showed a significantly higher relative contribution in virus-positive cases compared to virus-negative cases in HNSCC, GC, and HCC ($p=7.9e-6$, $1.3e-4$, 0.0032 , respectively; Fig. R22–23). Although CC exhibited a similar trend, the difference was not statistically significant, likely due to the small number of HPV-negative cases ($n = 4$, Fig. R14). This observation is consistent with studies showing that HPV’s E6 and E7 proteins disrupt normal cell cycle regulation and mitotic progression, leading to chromosomal abnormalities²⁸. Fig. R22-23 are included as Fig. S11-12 in the modified manuscript.

Added in Results Section) *“To identify differences in structural variant (SV) and copy number (CN) signatures between virus-positive and virus-negative tumors, we conducted a signature analysis using SigProfilerExtractor²⁶, following the same methodology as in SBS analysis. This analysis was performed on WGS samples from HCC, HNSCC, CC, and GC obtained from PCAWG, as well as BL and cHL (Fig. S9–10 and Table S11). Among SV signatures, no significant differences were observed between virus-positive and virus-negative tumors, except for an enrichment of the SV2 signature, which consists of non-clustered translocations (COSMIC v3.4), in HPV-positive HNSCC ($q = 0.0025$, Fig. S9G). In the CN signature analysis, several de novo signatures that could not be decomposed into known COSMIC CN signatures were identified as differing in multiple cancer types (Fig. S10).*

Furthermore, we compared differences in chromosomal instability (CIN) signatures between virus-positive and virus-negative tumors using CIN signature data from a recent study on TCGA cohorts, including HCC, HNSCC, CC, and GC (Table S3)²⁷. Notably, the CX1 signature, which involves whole-arm or whole-chromosome changes and is associated with chromosome missegregation due to defective mitosis and/or telomere dysfunction²⁷, showed a significantly higher relative contribution in virus-positive cases compared to virus-negative cases in HNSCC, GC, and HCC ($p=7.9e-6$, $1.3e-4$, 0.0032 , respectively; Fig. S11–12, Table S12–13). Although CC exhibited a similar trend, the difference was not statistically significant, likely due to the small number of HPV-negative cases ($n = 4$, Fig. S12). This observation is consistent with studies showing that HPV’s E6 and E7 proteins disrupt normal cell cycle regulation and mitotic progression, leading to chromosomal abnormalities²⁸.”

A**B****C**

Fig. S20. Structure variant signatures in A) hepatocellular carcinoma (HCC), B) burkitt lymphoma (BL), C) classical hodgkin lymphoma (cHL), D) gastric cancer (GC), E) cervical cancer (CC), F) head and neck squamous cell carcinoma (HNSCC). G) Significant structure variant signature in HNSCC. Q value by MWU test.

Fig. R21. Copy number signatures in A) GC, B) CC, C) cHL, D) HCC, E) HNSCC, F) BL. G-I) Significant copy number signatures in G) HCC, H) HNSCC, and I) BL. Q value by MWU test.

Fig. R22. Chromosomal instability signatures in A) HNSCC. B-F) Significant chromosomal instability signatures in HNSCC. P value by MWU test.

F
Defective Mitosis

G
Impaired HR

H
Impaired NHEJ

I

Fig. R23. Chromosomal instability signatures in A) GC, B) HCC, C) CC. B-D) Significant chromosomal instability signatures in GC. F-H) Significant chromosomal instability signatures in HCC. J) Significant chromosomal instability signatures in CC. P value by MWU test.

Comment 5) “Mutational signatures were analyzed by Palimpsest. Would the same conclusions be hold when using the state-of-art methods SigProfilerExtractor [PMID: 36388765] or MuSiCal [PMID: 38361034].”

Response 5) We thank the reviewer for suggesting the use of more advanced methods for mutational signature analysis. In response, we employed SigProfilerExtractor (v.1.1.24)²⁶ to extract de novo mutational signatures and to decompose them into the COSMICv3.4 signatures. We compared these findings with those obtained using Palimpsest and observed that most of the signatures identified by Palimpsest were also detected by SigProfilerExtractor. Moreover, both the direction (i.e., whether virus-positive or -negative samples had a higher proportion of mutations in each signature) and the level of significance were very similar (Table R9). For consistency with our structural variation and copy number signature analyses, which were also conducted using SigProfilerExtractor, we have chosen to present and use the SigProfilerExtractor results for mutational signature analysis in our updated manuscript. Results, figures, and methods are updated accordingly.

Modified in Results Section) “To detect and quantify the relative contribution of COSMIC mutation signatures³⁰ within the virus-associated cancers, we applied SigProfilerExtractor²⁶ to extract de novo mutational signatures and decompose them into COSMIC signatures across 7 available cancers. Virus-positive tumors exhibited different activities of mutation signatures compared to virus-negative tumors of the same cancer type (Fig. 3, S5 and Table S7).”

Signature	Palimpsest		SigProfilerExtractor	
	direction	Wilcoxon_ranksum_p	direction	Wilcoxon_ranksum_p
SBS3_HNSCC	higher in virus negative	4.70831E-09		
DBS1_MCC	higher in virus negative	0.000542058	higher in virus negative	7.71082E-05
SBS15_GC	higher in virus negative	0.003387619	higher in virus negative	0.012578035
SBS7b_MCC	higher in virus negative	0.005107194	higher in virus negative	0.000907045
ID7_GC	higher in virus negative	0.0071948	higher in virus negative	0.02291929

ID1_GC	higher in virus negative	0.014997676	higher in virus negative	0.026676527
ID2_GC	higher in virus negative	0.032323036	higher in virus negative	0.060021893
SBS45_HNSCC	higher in virus negative	0.035164661	higher in virus negative	0.033055838
ID3_HNSCC	higher in virus negative	0.042422919		
SBS7a_MCC	higher in virus negative	0.070003179	higher in virus negative	0.013045501
SBS6_HCC	higher in virus negative	0.122073766		
SBS40_HCC	higher in virus negative	0.127387733		
DBS2_HNSCC	higher in virus negative	0.12977748	higher in virus negative	0.075468469
SBS20_GC	higher in virus negative	0.141163209	higher in virus negative	0.809561888
ID2_CC	higher in virus negative	0.152776644	higher in virus negative	0.330767687
DBS4_HNSCC	higher in virus negative	0.176184771	higher in virus negative	0.148040278
SBS21_GC	higher in virus negative	0.27633481	higher in virus negative	0.839367232
ID5_BL	higher in virus negative	0.289578841		
SBS2_MCC	higher in virus negative	0.292334552		
DBS9_GC	higher in virus negative	0.321689434		
SBS40_PBL	higher in virus negative	0.35453948		
DBS1_CC	higher in virus negative	0.362520759		
SBS5_BL	higher in virus negative	0.372755503	higher in virus negative	0.204121407
ID2_HNSCC	higher in virus negative	0.385303924		
DBS9_CC	higher in virus negative	0.415815157		
SBS6_BL	higher in virus negative	0.420781234		
ID8_PBL	higher in virus negative	0.48745329		
SBS15_CC	higher in virus negative	0.487942581	higher in virus negative	0.617183459
ID8_HCC	higher in virus negative	0.521408175	higher in virus negative	0.405462061
SBS84_BL	higher in virus negative	0.594782083	higher in virus negative	0.298396713
SBS17b_GC	higher in virus negative	0.676372773	higher in virus negative	0.871769236
DBS1_HNSCC	higher in virus negative	0.753610605	higher in virus positive	0.770202592
ID2_PBL	higher in virus negative	0.816961322	higher in virus negative	0.48745329
DBS2_HCC	higher in virus negative	0.855101298		
DBS9_BL	higher in virus negative	0.93126201		
SBS4_CC	higher in virus negative	0.938229358		
DBS5_BL	higher in virus negative	0.961779496		
SBS2_HNSCC	higher in virus positive	6.22405E-06	higher in virus positive	0.000181837
SBS5_MCC	higher in virus positive	0.000111101	higher in virus positive	0.002913912
SBS1_HNSCC	higher in virus positive	0.000600509	higher in virus positive	0.001460776
SBS1_MCC	higher in virus positive	0.000604362	higher in virus positive	0.035716058
SBS24_HCC	higher in virus positive	0.005741363	higher in virus positive	0.066016865
SBS40_GC	higher in virus positive	0.007178642		
ID5_GC	higher in virus positive	0.024258799	higher in virus positive	0.012366693
SBS17b_BL	higher in virus positive	0.06366359		
ID3_HCC	higher in virus positive	0.070310894	higher in virus positive	0.277983931
ID8_CC	higher in virus positive	0.100402599		
SBS26_HCC	higher in virus positive	0.179199558		
SBS2_CC	higher in virus positive	0.201011164	higher in virus positive	0.237282435
SBS1_GC	higher in virus positive	0.265874302	higher in virus positive	0.127314855
SBS46_BL	higher in virus positive	0.266241925		
SBS13_MCC	higher in virus positive	0.279062458	higher in virus positive	0.99061269
SBS6_CC	higher in virus positive	0.338532109		
SBS1_CC	higher in virus positive	0.391817781	higher in virus positive	0.442959274
SBS22_HCC	higher in virus positive	0.392605584		
ID8_HNSCC	higher in virus positive	0.439306092		
SBS13_CC	higher in virus positive	0.478272364	higher in virus positive	0.542962124
SBS15_PBL	higher in virus positive	0.48745329	higher in virus positive	0.417886964
DBS11_BL	higher in virus positive	0.502291337	higher in virus positive	0.125162496
ID8_GC	higher in virus positive	0.551612202	higher in virus positive	0.528281601
ID15_GC	higher in virus positive	0.562704236		
DBS7_BL	higher in virus positive	0.618220052		
DBS4_HCC	higher in virus positive	0.631081031		
ID1_BL	higher in virus positive	0.722879775	higher in virus positive	0.961779496
DBS1_BL	higher in virus positive	0.773711285		
ID17_GC	higher in virus positive	0.78118271	higher in virus positive	0.867041473
SBS10b_GC	higher in virus positive	0.850532803	higher in virus negative	0.74455586
DBS4_GC	higher in virus positive	0.86054859	higher in virus positive	0.854065121
ID2_BL	higher in virus positive	0.923646643	higher in virus negative	0.431927028
DBS9_HCC	higher in virus positive	0.937465197		
DBS5_HCC	higher in virus positive	0.943923037		
SBS7b_HNSCC	higher in virus positive	0.983353352	higher in virus negative	0.511313572

Table R9. Mutational signature identification and comparison results using the Palimpsest method and the SigProfilerExtractor method. The proportion of mutations attributed to each signature and cohort is compared between virus-positive and virus-negative cases using both methods. The direction of the statistical test is indicated in red (higher in virus-positive) or blue (higher in virus-negative). Empty rows under SigProfilerExtractor indicate signatures identified exclusively through the Palimpsest method and not by SigProfilerExtractor. Signatures highlighted in yellow correspond to those shown in Fig. 3.

Comment 6) *“Viral sub-strains. Is it possible to stratify the analysis by viral sub-strains.”*

Response 6) We appreciate the reviewer’s insightful suggestion. Along with tumor subtypes, we examined viral strain differences for each of the features discussed in the manuscript, including mutation load, mutational signatures, chromosomal instability signatures, and somatic mutations in six genes recurrently found in virus-positive and virus-negative cases. The corresponding statistical results have been included in Tables S6, S8, S13, and S16 of the revised manuscript.

The analysis on HPV strains showed HPV strain-specific differences in mutation patterns across CC and HNSCC, with consistent trends observed in both cancers. APOBEC signatures were strongly correlated with total mutation load in both cancers^{35,36}, and HPV strains exhibited distinct APOBEC-associated mutation loads (Fig. R26, R27). In particular, HPV33 consistently demonstrated lower APOBEC-associated mutation loads compared to HPV16 in both HNSCC³⁴ and CC (Fig. R26, R27). In CC, the strain-specific differences in APOBEC signature levels appeared to be largely driven by total mutation load (Fig. R25, R27). In contrast, in HNSCC, HPV33 retained significantly lower APOBEC-associated mutation counts³⁴ even after adjusting for total mutations (Fig. R24, R26), suggesting additional strain-specific mechanisms beyond overall mutational burden. These findings indicate that APOBEC-mediated mutagenesis plays a central role in shaping the mutational landscape of HPV-positive CC and HNSCC, with HPV strain-specific effects potentially reflecting shared viral-host interaction mechanisms across these cancer types. However, the small sample size for certain HPV strains, such as HPV31 (n = 2) in CC, limits the generalizability of these findings. Larger studies are needed to confirm these strain-specific effects and to determine their potential implications for tumor progression and clinical outcomes. Larger studies are needed to confirm these strain-specific effects and to determine their potential implications for tumor progression and clinical outcomes. Fig. R24-27 are included as Fig. S3, S4, S7, S8 in the revised manuscript.

Fig. R24. HNSCC HPV strain differences in A) total nonsynonymous mutation load, B) driver gene mutation load, and C) age at disease onset. * MWU test, $p < 0.05$.

Fig. R25. Cervical cancer HPV strain differences in A) total nonsynonymous mutation load, B) driver gene mutation load, and C) age at disease onset. Refer to Table S6 for significant associations between age and HPV strains. * MWU test, $p < 0.05$.

Fig. R26. HNSCC HPV strain differences in A,C) fractions (%) and B,D) log₂(mutation counts+1) of SBS2 and SBS13 (both APOBEC associated signatures). * MWU test, p < 0.05.

Fig. R27. Cervical cancer HPV strain differences in A,C) fractions (%) and B,D) log2(mutation counts+1) of SBS2 and SBS13 (both APOBEC associated signatures). * MWU test, $p < 0.05$.

Modified in Results Section) “Additionally, we observed that several cancer subtypes and virus strains differed in mutation load (**Table S6**). The NS subtype, the most common subtype of cHL in our datasets (targeted, 84.6%, 204/241; WES, 73.4%, 47/64), exhibited the highest mutation load relative to other subtypes in both targeted and WES cohorts ($p = 1.6E-06$ and 0.010 , respectively). However, after adjusting for subtypes, sex, and age, only EBV status remained significantly associated with a lower mutation load in the targeted and WES cohorts, consistent with a previous report¹²⁸ (coefficient = -0.44 , -2.1 ; $p = 0.009$, 0.001). As expected²⁰, eBL had a higher total mutation load compared to sBL, reflecting a similar trend with EBV status ($p = 0.00029$). Since EBV-positivity was highly associated with eBL ($p = 7.68e-12$), when both subtype and EBV status were considered simultaneously, neither variable showed a significant independent association with total mutation load.

In CC, there was no significant difference in mutation load between squamous carcinoma and adenocarcinoma subtypes. However, age at diagnosis positively correlated with mutation load (Spearman’s $\rho = 0.31$, $p = 4.1E-05$, **Table S6**), consistent with a previous study¹³². Notably, multiple HPV strains in CC differed significantly in mutation load, with HPV31 exhibiting the highest levels (Fold change = 7.07 , $p = 0.02$, **Fig. S3**, **Table S6**). To account for age, HPV strains, viral status, on mutation load, we used generalized linear model regression model (GLM) and found that HPV31 and virus-positive status remained significantly associated with elevated mutation load (coefficient = 1.81 and 1.09 , $p = 0.013$ and < 0.001 , respectively).

*In HNSCC, females exhibited higher driver gene mutation counts than males ($p = 0.0058$), and age correlated positively with both total and driver mutation counts (Spearman's $\rho = 0.19$ and 0.27 , $p = 2.4E-05$ and $1.2E-09$, respectively; **Table S6**), consistent with previous findings¹⁴⁷. Unlike CC, there were no significant differences in mutation load between HPV strains in HNSCC (**Fig. S4**), as reported in a prior study³⁴.*

In HCC, no difference in mutation load was evident between HBV and HCV related HCC. Neither Activated B-cell (ABC) nor Germinal Center B-cell (GCB) subtypes of PCNSL differed in their mutation loads."

*"In HNSCC, HPV-positive HNSCC had higher relative contribution of SBS2/APOBEC mutations (**Fig. 3E,F** and **Table S7**), as reported in previous study¹. This is consistent with the hypothesis that HPV oncoproteins may increase APOBEC3A and APOBEC3B expression and mutagenic activity^{32,33}. In a multivariate analysis of HPV-positive HNSCC, which accounted for sex, HPV strain, age, and total mutation count, HPV33-associated HNSCCs showed significantly lower SBS2/APOBEC mutation counts, while total mutation count was positively associated with SBS2/APOBEC mutation counts, both findings consistent with previous reports^{34,35} (coefficient = -1.37 and 0.012 , $p = 0.003$ and <0.001 , respectively; **Table S9, Fig. S7**).*

*The mutation counts associated with the SBS5/clock-like signature were higher in HPV-negative cervical cancer (CC) compared to HPV-positive CC ($p=0.013$, **Table S7**), consistent with our observation that total mutation counts are positively correlated with age and tend to be higher in HPV-negative cases. Notably, the levels of SBS2/13/APOBEC-associated signatures varied between HPV strains, with HPV31 significantly exhibiting the highest mutation counts (**Fig. S8** and **Table S8**). A similar trend observed in HNSCC was also apparent in CC, where the HPV33 strain exhibited lower APOBEC signature levels compared to HPV16, although this difference was not statistically significant in CC. These strain-specific differences aligned with our findings on total mutation load. Importantly, these differences appear to be primarily driven by variations in total mutation load, as HPV strain-specific effects were no longer significant in a multivariable model that accounted for virus strain, virus status, total nonsynonymous mutation load, and age. In this model, virus-positive status and total nonsynonymous mutation load were positively associated with APOBEC signature mutations (**Table S10**), consistent with previous findings³⁶"*

Modified in Discussion Section) *"Our analysis noted HPV strain-specific differences in mutation patterns across CC and HNSCC, with consistent trends observed in both cancers. APOBEC signatures were strongly correlated with total mutation load in both cancers^{35,36}, and HPV strains exhibited distinct APOBEC-associated mutation loads. In particular, HPV33 consistently demonstrated lower APOBEC-associated mutation loads compared to HPV16 in both HNSCC³⁴ and CC. In CC, the strain-specific differences in APOBEC signature levels appeared to be largely driven by total mutation load. In contrast, in HNSCC, HPV33 retained significantly lower APOBEC-associated mutation counts³⁴ even after adjusting for total mutations, suggesting additional strain-specific mechanisms beyond overall mutational burden. These findings indicate that APOBEC-mediated mutagenesis plays a central role in shaping the mutational landscape of HPV-positive CC and HNSCC, with HPV strain-specific effects potentially reflecting shared viral-host interaction mechanisms across these cancer types. However, the small sample size for certain HPV strains, such as HPV31 ($n = 2$) in CC, limits the generalizability of these findings. Larger studies are needed to confirm these strain-specific effects and to determine their potential implications for tumor progression and clinical outcomes."*

Minor:

Minor Comment 1) “Fig1 a. show confidence interval.”

Response Minor 1) We appreciate the reviewer’s comments and have revised Fig. 1A for clarity by separating it into two sub-figures: Fig. R1A now compares the sex incidence of virus-associated versus non-virus-associated cancers, while Fig. R1B focuses specifically on the sex incidence of virus-positive and virus-negative cases within virus-associated cancers. We have adjusted the layout of Fig. R1B so that virus-positive and virus-negative data points within each cancer type appear adjacent for easier comparison. Additionally, we went through each cohort in Fig. R1A retrieving the sample sizes and merged them into one point per cancer type with confidence interval lines to increase clarity and statistical power. The cohorts with no sample size information were taken out. The updated figures show significant difference of male-to-female ratio between virus-associated versus non-virus-associated cancers (Fisher’s Exact $P = 2.2e-16$, Fig. R1A) and between virus-positive and virus-negative cases within virus-associated cancers (Wilcoxon Rank Sum Test $P = 5.23e-08$, Fig. R1B). We included Fig. R28 as Fig. 1A-B in the revised manuscript.

Fig. R28. Epidemiological trends of virus-associated cancers. A) Incidence ratios of virus-associated and non-virus-associated cancers, analyzed using Fisher's Exact Test. Error bars represent 95% confidence intervals for each M/F incidence ratio, and B) Virus-positive and virus-negative tumors in virus-associated cancers in males compared to females (M/F) reported in selected published studies, analyzed using Wilcoxon Rank Sum Test. Each point corresponds to an incidence ratio reported in a published study in Table S2. Virus-associated cancers: ANKL, aggressive NK-cell leukemia; GC, gastric cancer; HCC, hepatocellular carcinoma, HL, Hodgkin lymphoma; HNSCC, head and neck squamous cell carcinoma, MCC, Merkel cell carcinoma; NKTCL, Natural killer/T-cell lymphoma; NPC, nasopharyngeal carcinoma; PBL, plasmablastic lymphoma; PCNSL, primary central nervous system lymphoma. Non-virus associated cancers: ALL, acute lymphoblastic leukemia; BCC, basal cell carcinoma; CLL, chronic lymphocytic leukemia. Virus-positive tumors: BL, Burkitt lymphoma.

Minor Comment 2) “Define GC in the main text.”

Response Minor 2) We have defined gastric cancer (GC) in the main text.

Minor Comment 3) “The title is “Genomic landscape of virus-associated cancers”. But the first section is about epidemiological trends which seems unrelated to the major theme of the manuscript.”

Response Minor 3) One of the manuscript's primary objectives is to identify the genomic differences of virus-associated cancers. Among our main findings, we identified recurrent mutations in the *DDX3X* gene, located on the sex chromosome, which the truncating mutations occur predominantly in males and are more frequent in virus-positive cases (Figures 4A, C, and D). Although it may not be the sole factor, these observations suggest that virus-associated, sex chromosome-linked mutations could be associated to the observed male-biased incidence of these cancers.

Therefore, we considered it as important to highlight the differences in incidence between males and females (M/F ratio) in these malignancies, as well as variations based on geographic location. For instance, in gastric cancer, the known discrepancy can be striking: EBV-positive gastric tumors have a mean M/F ratio of 2.7 (n=33 studies), in contrast to EBV-negative tumors, which have a mean M/F ratio of 0.93 (n=31 studies; Wilcoxon p-value = 1.0e-10; Fig. R1B). In a meta-analysis by Lee et al.⁵, encompassing multiple populations from Chile, Colombia, The Netherlands, China, and Japan (totaling 9,738 patients), the M/F ratio in EBV-positive cohorts reached as high as 12.67 (van Beek et al. cohort), while the highest M/F ratio in EBV-negative groups was only 2.0 (Moritani et al. cohort)⁵. A similar trend is observed in head and neck squamous cell carcinoma (HNSCC), where the M/F ratio is 8.37 in HPV-positive cases⁶, compared to 3.22 in HPV-negative cases⁷. These pronounced sex differences in virus-positive cancers underscore the significant influence of viral factors on gender disparities in cancer incidence.

Reviewer #3 (Remarks to the Author): Expert in virus-associated cancers, genetics, immunology, immunotherapy

Reviewer's summary) *"This study provides a comprehensive analysis of the genomic changes associated with virus driven cancers. It confirms previous epidemiological analyses showing gender and geographical variations in the incidence rates of these virus-associated tumours. The overall conclusion of fewer mutations and deletions in virus-positive versus virus-negative cases, particularly in certain genes such as p53, is consistent with previous studies where individual virus-associated cancers have been examined. The novel observation of increased mutations in the RNA helicases DDX3X and EIF4A is interesting and provides an important focus for more detailed molecular studies."*

Response) We appreciate the reviewer for the summary of our manuscript.

Comment 1) *"Some of the references are old and not relevant. For instance, more recent references covering the different patterns of virus gene expression in EBV-associated malignancies should be included."*

Response 1) We thank the reviewer for pointing out the irrelevancy in the references. We removed the old reference and added new references to our introduction section as below:

Modified in Introduction Section) *"Type I latency is characterized by the expression of EBV nuclear antigen 1 (EBNA1), essential for viral DNA replication and potentially inhibiting apoptosis, along with*

EBV-encoded small RNAs (EBERs) and BamHI-A rightward transcript (BART) miRNAs¹⁵³. In contrast, type II latency involves the expression of latent membrane proteins (LMP1 and LMP2), which activate the NF- κ B and PI3K/AKT pathways, in addition to the markers of latency I¹⁵³. Latency I is commonly observed in Burkitt lymphoma (BL), while latency II is seen in classical Hodgkin lymphoma (cHL) and nasopharyngeal carcinoma, with EBV-positive gastric cancer being associated with either latency type I or II^{153,154}.”

Comment 2) *“The finding that EBV-positive Hodgkin lymphoma (HL) occurs ‘most frequently in North Africa, the Middle East and South America’ is surprising and raises concerns about the methodology used to ascribe EBV positivity in the included studies. It would also be useful to include information about the age profile of these tumours in the different countries.”*

Response 2) The database used for showing the global incidence of Hodgkin Lymphoma employed EBER and LMP-1 in situ hybridization for the reliable detection of EBV positivity¹⁵⁵ which are widely recognized as gold-standard methods for identifying EBV-associated HL.

Regarding the age profile, we have included the following information to the results section:

Modified in Results Section) *“EBV-positive HL is more frequently observed in cases with mixed cellular histology, males, children, older adults, and in developing countries. In these regions, HL incidence shows an earlier peak, primarily in children under 15 years, and is characterized by a high prevalence of EBV positivity. In contrast, young-adult onset nodular sclerosis HL, typical of the 15 to 39 age group in industrialized nations, is usually EBV-negative¹⁵⁶⁻¹⁵⁸.”*

Comment 3) *“Previous studies have also found that the total mutation count is lower in virus-positive compared to virus-negative tumours while highlighting important differences in the epigenetic landscape in these tumours. While understandingly not the focus of the current manuscript, it is important to mention the contribution of epigenetic alterations to the pathogenesis of virus-associated cancers.”*

Response 3) We appreciate the reviewer’s suggestion. While this study does not focus on epigenetic alterations, previous studies have highlighted significant differences in the epigenetic landscapes of virus-associated cancers. For example, HPV-positive CC exhibit higher promoter methylation and increased gene silencing compared to virus-negative cases³⁶. Similarly, HPV-positive HNSCC is widely reported to be hypermethylated compared to HPV-negative cases^{159,160}, with genes such as *CDKN2A*, *RASSF1*, and *CCNA1*, which are involved in cell cycle regulation and apoptosis, frequently affected^{159,161,162}. In GC, EBV-positive cancers are characterized by DNA hypermethylation. These prior findings underscore the important role of epigenetic alterations in the pathogenesis of virus-associated cancers. We have included this paragraph in the discussion section of the manuscript.

Added in Discussion Section) *“While this study does not focus on epigenetic alterations, previous studies have highlighted significant differences in the epigenetic landscapes of virus-associated cancers. For example, HPV-positive CC exhibit higher promoter methylation and increased gene silencing compared to virus-negative cases³⁶. Similarly, HPV-positive HNSCC is widely reported to be hypermethylated compared to HPV-negative cases^{159,160}, with genes such as *CDKN2A*, *RASSF1*, and *CCNA1*, which are involved in*

cell cycle regulation and apoptosis, frequently affected^{159,161,162}. In GC, EBV-positive cancers are characterized by DNA hypermethylation¹⁸. These prior findings underscore the important role of epigenetic alterations in the pathogenesis of virus-associated cancers.”

Comment 4) *“Mutations in DDX3X appear to be more common in virus-associated lymphomas rather than carcinomas. How is this explained?”*

Response 4) We thank the reviewer for raising this important point. Mutations in *DDX3X* do appear to be more common in virus-associated lymphomas than in carcinomas, though the exact reason remains unclear. *DDX3X* is expressed in most cell types and plays a critical role in lymphomagenesis and virus-associated immunity, which may explain its higher mutation frequency in lymphomas^{152,163-166}. In natural killer T-cell lymphoma (NKTCL), mutated *DDX3X* activates oncogenic pathways such as NF-κB and MAPK while disrupting RNA unwinding and cell cycle regulation, facilitating lymphomagenesis¹⁶⁷. In Burkitt lymphoma (BL), *DDX3X* loss-of-function mutations moderate MYC-driven translation, further promoting lymphoid malignancies¹⁵². Additionally, *DDX3X* has been reported to promote type I interferon (IFN) responses, which are essential for antiviral immunity, and viruses like EBV may exploit *DDX3X* mutations to suppress IFN production and evade immune detection^{164,166}.

Comment 5) *“These results suggest that EBV-positive status may be a positive prognostic marker for patients undergoing ICI therapy for GC and HNSCC, which may be correlated with PD-L1 expression in GC but may represent an independent marker in HNSCC.’ This statement lines 275-278 is confusing. What EBV-positive HNSCC is being referred to? Comparisons with EBV-positive NPC are difficult as all these tumours are virus positive.”*

Response 5) We apologize for the mistake and thank the reviewer for catching the error. We were referring to HPV-positive HNSCC and EBV-positive GC. We agree with the comment on the difficulty to include NPC when evaluating the efficacy of ICI in viral-associated cancers, and do not claim to make any direct comparisons of immunotherapy response across cancer types. We will however note that immunotherapy has been shown to be a promising avenue for NPC treatment¹⁶⁸, in line with the underlying hypothesis of our work.

Comment 6) *“The response to ICI therapy could be examined in the context of the tumour microenvironment as the TCGA includes transcriptional data related to infiltrating lymphocytes etc.”*

Response 6) We thank the reviewer for suggesting this analysis. We compared estimates of CD8+ T cell infiltration in TCGA viral-positive and viral-negative GC, HNSCC and HCC tumors¹³³. Using CIBERSORT¹³⁴ results on bulk RNA-seq, we found increased CD8+ T cell scores in EBV-positive versus EBV-negative GC, and in HPV-positive versus HPV-negative HNSCC (Wilcoxon rank-sum test, p-values = 7.4e-09 and 4.3e-10 respectively; Fig. R29).

This increase in CD8+ T cell infiltration in viral-positive GC and HNSCC was significant across all deconvolution algorithms we used: TIMER¹³³ (p-values = 2.0e-09 and 1.3e-04), quanTIseq¹³⁶ (p-values = 7.2e-08 and 1.8e-10), MCP-counter¹³⁷ (p-values = 1.2e-07 and 2.3e-13), EPIC¹³⁸ (p-values = 4.1e-07 and 1.3e-11), and xCell¹³⁸ (p-values = 1.7e-09 and 2.4e-06). On the other hand, none of the deconvolution

algorithms detected a significant difference in CD8+ T cell infiltration between HBV/HCV+ and HBV/HCV- HCC (Fig. R31).

This analysis highlights a possible mechanism for the higher ICI response in viral positive GC and HNSCC tumors. The presence of immunogenic viral antigens might recruit more CD8+ T cells to the tumor. These CD8+ T cells could then, through therapeutic intervention, be reactivated into an anti-tumor response. These results are also in accordance with another study¹⁶⁹, which showed elevated levels of CD8A and CD8B expression in HPV+ versus HPV- HNSCC, and EBV+ versus EBV- GC.

In the same cohorts, we analyzed T cell receptor (TCR) clonality, focusing on alpha and beta CDR3 sequences extracted from bulk RNA-seq with TRUST4¹³⁹. We used the two metrics used in the original TRUST4 TCGA publication¹⁴⁰: clonotypes per thousand CDR3 reads (CPK), and clonality, defined as (1 – normalized Shannon entropy). We found that both GC and HNSCC viral-positive tumors displayed evidence of TCR beta clonal selection as compared to viral-negative GC and HNSCC (Fig. R29-31). Analysis of TCR alpha showed corresponding clonal selection for viral-positive GC and HNSCC.

Finally, we looked at TCR alpha and beta repertoire overlap, by measuring the pairwise Jaccard index within the viral-positive and the viral-negative subgroups of GC, HNSCC and HCC TCGA samples. We found a significantly higher overlap of TCR alpha and beta clones for viral-positive versus viral-negative GC, and a similar trend in HNSCC, perhaps highlighting evolutionary convergence of TCR clones across patients due to exposure to shared viral antigens (Table R10).

Fig. R29, R30, and R31 is included in the revised manuscript as Fig. 5C-D and Fig. S19, respectively.

Fig. R29. CIBERSORT CD8+ T cell infiltration score in TCGA studies of GC (TCGA-STAD), HCC (TCGA-LIHC), and HNSCC (TCGA-HNSC). $p < 0.05$, MWU test. Lines show median and quartile distributions (upper: 75th, lower: 25th percentiles).

Fig. R30. TCR β clonotypes per thousand reads (CPK), versus viral status of tumors in TCGA studies of GC (TCGA-STAD), HCC (TCGA-LIHC), and HNSCC (TCGA-HNSC). $p < 0.05$, MWU test. Lines show median and quartile distributions (upper: 75th, lower: 25th percentiles).

Fig. R31. Analysis of TCR alpha and beta CDR3 diversity metrics in virus positive and negative GC, HNSCC and HCC TCGA tumors. A) TCR α clonotypes per thousand reads (CPK). B) TCR α and TCR β clonality, defined as 1 - normalized Shannon entropy. $p < 0.05$, MWU test.

Cancer Type:	Chain:	Jaccard index (viral negative):	Jaccard index (viral positive):	Wilcoxon statistic:	Wilcoxon p-value:
GC	TRA	0.000809	0.002780	4.08	4.45e-05
HNSCC	TRA	0.000134	0.000519	1.52	0.13
HCC	TRA	0.000549	0.000451	0.03	0.97
GC	TRB	0.000209	0.001236	4.37	1.27e-05
HNSCC	TRB	0.000213	0.001127	1.58	0.11
HCC	TRB	0.000163	0.000030	-0.06	0.95

Table R10. TCR alpha and beta repertoire overlap within the virus-positive and the virus-negative subgroups of GC, HNSCC and HCC TCGA.

Added in Results Section) *“To determine the relationship between virus positivity and other prognostic markers of ICI therapy response, we compared the expression of PD-L1 [CD274] in TCGA’s studies of GC¹⁸, HCC²⁴, and HNSCC¹⁷⁰. CD274 expression was higher in virus-positive GC compared to virus-negative GC (median 6.74 and 5.21, $p=1.6e-07$), but this was not the case in HCC or HNSCC (Fig. 5B). Using results of CIBERSORT^{133,134} deconvolution of the same TCGA samples, we found evidence of higher CD8+ T cell infiltration in EBV-positive versus EBV-negative GC (median 0.19 and 0.09, $p=7.4e-09$), and in HPV-positive versus HPV-negative HNSCC (median 0.13 and 0.07, $p=4.3e-10$), but not between virus-positive and virus-negative HCC (Fig. 5C). These results were replicated in all other deconvolution approaches tested¹³⁵⁻¹³⁸ (Fig. S18). Similarly, analysis of TRUST4^{139,140}-extracted T cell receptors showed higher T cell β receptor clonal selection, measured by counts per clonotypes per thousand reads (CPK), in both virus-positive GC (median 185.16 and 310.73, $p=7.8e-10$) and HNSCC (median 345.72 and 444.44, $p=0.017$) but not in HCC (Fig. 5D). We observed similar trends for the α chain, as well as with the metric of clonality (Fig. S19). The effect of virus positivity appears independent from TMB, as virus-positive GC and HNSCC have fewer mutations than virus-negative GC and HNSCC, respectively (Fig. 2A), while in HCC virus-positive and negative cases have similar mutation load ($p=0.12$). These results suggest that virus-positive status may be a positive prognostic marker for patients undergoing ICI therapy for GC and HNSCC, which may be correlated with higher CD8+ T cell infiltration, T cell receptor clonal selection, and T cell exhaustion (for GC only).”*

Added in Discussion Section) *“In both GC and HNSCC, we observe a marked increase in CD8+ T cell infiltration as well as an increase in T cell receptor clonality in virus-positive tumors. The high immunogenicity of viral antigens expressed in these tumors might elicit a bigger and more clonal T cell response, which could be reactivated through ICI therapy.”*

References

- 1 Zapatka, M. *et al.* The landscape of viral associations in human cancers. *Nat Genet* **52**, 320-330 (2020). <https://doi.org/10.1038/s41588-019-0558-9>
- 2 Nault, J. C. *et al.* Recurrent AAV2-related insertional mutagenesis in human hepatocellular carcinomas. *Nature Genetics* **47**, 1187-+ (2015). <https://doi.org/10.1038/ng.3389>
- 3 Cobbs, C. S. *et al.* Human cytomegalovirus infection and expression in human malignant glioma. *Cancer Research* **62**, 3347-3350 (2002).
- 4 Mundo, L. *et al.* Frequent traces of EBV infection in Hodgkin and non-Hodgkin lymphomas classified as EBV-negative by routine methods: expanding the landscape of EBV-related lymphomas. *Mod Pathol* **33**, 2407-2421 (2020). <https://doi.org/10.1038/s41379-020-0575-3>
- 5 Lee, J. H. *et al.* Clinicopathological and molecular characteristics of Epstein-Barr virus-associated gastric carcinoma: a meta-analysis. *J Gastroenterol Hepatol* **24**, 354-365 (2009). <https://doi.org/10.1111/j.1440-1746.2009.05775.x>
- 6 Wu, Q. *et al.* HPV Positive Status Is a Favorable Prognostic Factor in Non-Nasopharyngeal Head and Neck Squamous Cell Carcinoma Patients: A Retrospective Study From the Surveillance, Epidemiology, and End Results Database. *Front Oncol* **11**, 688615 (2021). <https://doi.org/10.3389/fonc.2021.688615>
- 7 Li, H., Park, H. S., Osborn, H. A. & Judson, B. L. Sex differences in patients with high risk HPV-associated and HPV negative oropharyngeal and oral cavity squamous cell carcinomas. *Cancers Head Neck* **3**, 4 (2018). <https://doi.org/10.1186/s41199-018-0031-y>
- 8 Cancer, L. F. I. A. f. R. o. in *IARC Monographs on the Evaluation of Carcinogenic Risks to Humans* Vol. No. 100B. *EPSTEIN-BARR VIRUS* (2012).
- 9 Chabay, P. A. & Preciado, M. V. EBV primary infection in childhood and its relation to B-cell lymphoma development: a mini-review from a developing region. *Int J Cancer* **133**, 1286-1292 (2013). <https://doi.org/10.1002/ijc.27858>
- 10 Devkota, K., He, M., Liu, M. Y., Li, Y. & Zhang, Y. W. Increasing Epstein-Barr virus infection in Chinese children: A single institutional based retrospective study. *Fl000Res* **7**, 1211 (2018). <https://doi.org/10.12688/fl000research.15544.2>
- 11 Bouvard, V. *et al.* A review of human carcinogens-Part B: biological agents. *Lancet Oncol* **10**, 321-322 (2009). [https://doi.org/10.1016/S1470-2045\(09\)70096-8](https://doi.org/10.1016/S1470-2045(09)70096-8)
- 12 Kwok, H., Chan, K. W., Chan, K. H. & Chiang, A. K. S. Distribution, Persistence and Interchange of Epstein-Barr Virus Strains among PBMC, Plasma and Saliva of Primary Infection Subjects. *Plos One* **10** (2015). [https://doi.org/ARTN e0120710](https://doi.org/10.1371/journal.pone.0120710)
- 13 Smith, N. A. *et al.* Differences in the Epstein-Barr Virus gp350 IgA Antibody Response Are Associated With Increased Risk for Coinfection With a Second Strain of Epstein-Barr Virus. *J Infect Dis* **219**, 955-963 (2019). <https://doi.org/10.1093/infdis/jiy601>
- 14 Lung, M. L., Li, S. B. & Chang, R. S. Study of Epstein-Barr-Virus (Ebv) Transmission by Ebv Genotyping. *J Infect Dis* **164**, 213-214 (1991).
- 15 Neves, M., Marinho-Dias, J., Ribeiro, J. & Sousa, H. Epstein-Barr Virus Strains and Variations: Geographic or Disease-Specific Variants? *J Med Virol* **89**, 373-387 (2017). <https://doi.org/10.1002/jmv.24633>
- 16 Tsao, S. W., Tsang, C. M. & Lo, K. W. Epstein-Barr virus infection and nasopharyngeal carcinoma. *Philos Trans R Soc Lond B Biol Sci* **372**, 20160270 (2017). <https://doi.org/10.1098/rstb.2016.0270>

- 17 Yoon, S., Baik, B., Park, T. & Nam, D. Powerful p-value combination methods to detect incomplete association. *Sci Rep* **11**, 6980 (2021). <https://doi.org:10.1038/s41598-021-86465-y>
- 18 The Cancer Genome Atlas Research Network. Comprehensive molecular characterization of gastric adenocarcinoma. *Nature* **513**, 202-209 (2014). <https://doi.org:10.1038/nature13480>
- 19 Amber, K., McLeod, M. P. & Nouri, K. The Merkel cell polyomavirus and its involvement in Merkel cell carcinoma. *Dermatol Surg* **39**, 232-238 (2013). <https://doi.org:10.1111/dsu.12079>
- 20 Grande, B. M. *et al.* Genome-wide discovery of somatic coding and noncoding mutations in pediatric endemic and sporadic Burkitt lymphoma. *Blood* **133**, 1313-1324 (2019). <https://doi.org:10.1182/blood-2018-09-871418>
- 21 Ramis-Zaldivar, J. E. *et al.* MAPK and JAK-STAT pathways dysregulation in plasmablastic lymphoma. *Haematologica* **106**, 2682-2693 (2021). <https://doi.org:10.3324/haematol.2020.271957>
- 22 Amaddeo, G. *et al.* Integration of tumour and viral genomic characterisations in HBV-related hepatocellular carcinomas. *Gut* **64**, 820 (2015). <https://doi.org:10.1136/gutjnl-2013-306228>
- 23 Weniger, M. A. & Kuppers, R. Molecular biology of Hodgkin lymphoma. *Leukemia* **35**, 968-981 (2021). <https://doi.org:10.1038/s41375-021-01204-6>
- 24 The Cancer Genome Atlas Research Network. Comprehensive and Integrative Genomic Characterization of Hepatocellular Carcinoma. *Cell* **169**, 1327-1341 e1323 (2017). <https://doi.org:10.1016/j.cell.2017.05.046>
- 25 Mork, J. *et al.* Human Papillomavirus Infection as a Risk Factor for Squamous-Cell Carcinoma of the Head and Neck. *New England Journal of Medicine* **344**, 1125-1131 (2001). <https://doi.org:doi:10.1056/NEJM200104123441503>
- 26 Islam, S. M. A. *et al.* Uncovering novel mutational signatures by de novo extraction with SigProfilerExtractor. *Cell Genom* **2**, None (2022). <https://doi.org:10.1016/j.xgen.2022.100179>
- 27 Drews, R. M. *et al.* A pan-cancer compendium of chromosomal instability. *Nature* **606**, 976-983 (2022). <https://doi.org:10.1038/s41586-022-04789-9>
- 28 Mallick, S., Choi, Y., Taylor, A. M. & Cosper, P. F. Human Papillomavirus-Induced Chromosomal Instability and Aneuploidy in Squamous Cell Cancers. *Viruses* **16** (2024). <https://doi.org:10.3390/v16040501>
- 29 Schulze, K. *et al.* Exome sequencing of hepatocellular carcinomas identifies new mutational signatures and potential therapeutic targets. *Nat Genet* **47**, 505-511 (2015). <https://doi.org:10.1038/ng.3252>
- 30 Alexandrov, L. B. *et al.* The repertoire of mutational signatures in human cancer. *Nature* **578**, 94-101 (2020). <https://doi.org:10.1038/s41586-020-1943-3>
- 31 Starrett, G. J. *et al.* Clinical and molecular characterization of virus-positive and virus-negative Merkel cell carcinoma. *Genome Med* **12**, 30 (2020). <https://doi.org:10.1186/s13073-020-00727-4>
- 32 Henderson, S., Chakravarthy, A., Su, X., Boshoff, C. & Fenton, T. R. APOBEC-mediated cytosine deamination links PIK3CA helical domain mutations to human papillomavirus-driven tumor development. *Cell reports* **7**, 1833-1841 (2014).

- 33 Warren, C. J., Westrich, J. A., Van Doorslaer, K. & Pyeon, D. Roles of APOBEC3A and APOBEC3B in human papillomavirus infection and disease progression. *Viruses* **9**, 233 (2017).
- 34 Chatfield-Reed, K., Gui, S., O'Neill, W. Q., Teknos, T. N. & Pan, Q. HPV33+ HNSCC is associated with poor prognosis and has unique genomic and immunologic landscapes. *Oral Oncology* **100**, 104488 (2020).
<https://doi.org/10.1016/j.oraloncology.2019.104488>
- 35 Gillison, M. L. *et al.* Human papillomavirus and the landscape of secondary genetic alterations in oral cancers. *Genome Res* **29**, 1-17 (2019).
<https://doi.org/10.1101/gr.241141.118>
- 36 The Cancer Genome Atlas Research Network. Integrated genomic and molecular characterization of cervical cancer. *Nature* **543**, 378-384 (2017).
<https://doi.org/10.1038/nature21386>
- 37 Chu, Y.-J. *et al.* Aflatoxin B1 exposure increases the risk of hepatocellular carcinoma associated with hepatitis C virus infection or alcohol consumption. *European Journal of Cancer* **94**, 37-46 (2018).
- 38 Chu, Y. J. *et al.* Aflatoxin B1 exposure increases the risk of cirrhosis and hepatocellular carcinoma in chronic hepatitis B virus carriers. *International journal of cancer* **141**, 711-720 (2017).
- 39 Sung, H. *et al.* Global Cancer Statistics 2020: GLOBOCAN Estimates of Incidence and Mortality Worldwide for 36 Cancers in 185 Countries. *CA Cancer J Clin* **71**, 209-249 (2021). <https://doi.org/10.3322/caac.21660>
- 40 Abdulamir, A., Hafidh, R., Abdulmuhammen, N., Abubakar, F. & Abbas, K. The distinctive profile of risk factors of nasopharyngeal carcinoma in comparison with other head and neck cancer types. *BMC public health* **8**, 1-16 (2008).
- 41 Adham, M. *et al.* Nasopharyngeal carcinoma in Indonesia: epidemiology, incidence, signs, and symptoms at presentation. *Chinese journal of cancer* **31**, 185 (2012).
- 42 Agelli, M. & Clegg, L. X. Epidemiology of primary Merkel cell carcinoma in the United States. *Journal of the American Academy of Dermatology* **49**, 832-841 (2003).
- 43 Ajila, V., Shetty, H., Babu, S., Shetty, V. & Hegde, S. Human papilloma virus associated squamous cell carcinoma of the head and neck. *Journal of sexually transmitted diseases* **2015**, 791024 (2015).
- 44 Aka, P. *et al.* Incidence and trends in Burkitt lymphoma in northern Tanzania from 2000 to 2009. *Pediatric blood & cancer* **59**, 1234-1238 (2012).
- 45 Akhtar, S., Oza, K. K. & Wright, J. Merkel cell carcinoma: report of 10 cases and review of the literature. *Journal of the American Academy of Dermatology* **43**, 755-767 (2000).
- 46 Alipov, G. *et al.* Epstein-Barr virus-associated gastric carcinoma in Kazakhstan. *World Journal of Gastroenterology* **11**, 27 (2005).
- 47 Andres, C., Belloni, B., Puchta, U., Sander, C. A. & Flaig, M. J. Prevalence of MCPyV in Merkel cell carcinoma and non-MCC tumors. *Journal of cutaneous pathology* **37**, 28-34 (2010).
- 48 Bassig, B. A. *et al.* Subtype-specific incidence rates of lymphoid malignancies in Hong Kong compared to the United States, 2001-2010. *Cancer Epidemiology* **42**, 15-23 (2016).
- 49 Bosch, F. X., Ribes, J., Cléries, R. & Díaz, M. Epidemiology of hepatocellular carcinoma. *Clinics in liver disease* **9**, 191-211 (2005).

- 50 Carrascal, E. *et al.* Epstein-Barr virus-associated gastric carcinoma in Cali, Colombia. *Oncology reports* **10**, 1059-1062 (2003).
- 51 Castillo, J. J., Bibas, M. & Miranda, R. N. The biology and treatment of plasmablastic lymphoma. *Blood, The Journal of the American Society of Hematology* **125**, 2323-2330 (2015).
- 52 Chang, M.-H. *et al.* Hepatitis B vaccination and hepatocellular carcinoma rates in boys and girls. *Jama* **284**, 3040-3042 (2000).
- 53 Chang, M. S., Lee, H. S., Kim, C. W., Kim, Y. I. & Kim, W. H. Clinicopathologic characteristics of Epstein-Barr virus-incorporated gastric cancers in Korea. *Pathology-Research and Practice* **197**, 395-400 (2001).
- 54 Chen, W. *et al.* Esophageal cancer incidence and mortality in China, 2009. *Journal of thoracic disease* **5**, 19 (2013).
- 55 Chong, J. M. *et al.* Expression of CD44 variants in gastric carcinoma with or without Epstein-Barr virus. *International journal of cancer* **74**, 450-454 (1997).
- 56 Claviez, A. *et al.* Impact of latent Epstein-Barr virus infection on outcome in children and adolescents with Hodgkin's lymphoma. *Journal of clinical oncology* **23**, 4048-4056 (2005).
- 57 Conte, S. *et al.* Population-Based Study detailing cutaneous melanoma incidence and mortality trends in Canada. *Frontiers in medicine* **9**, 830254 (2022).
- 58 Corvalan, A. *et al.* Epstein-Barr virus in gastric carcinoma is associated with location in the cardia and with a diffuse histology: a study in one area of Chile. *International journal of cancer* **94**, 527-530 (2001).
- 59 Czopek, J. P. *et al.* EBV-positive gastric carcinomas in Poland. *Polish Journal of Pathology: Official Journal of the Polish Society of Pathologists* **54**, 123-128 (2003).
- 60 Deo, S. *et al.* Colorectal Cancers in Low-and Middle-Income Countries—Demographic Pattern and Clinical Profile of 970 Patients Treated at a Tertiary Care Cancer Center in India. *JCO Global Oncology* **7**, 1110-1115 (2021).
- 61 Diepstra, A. *et al.* Latent Epstein-Barr virus infection of tumor cells in classical Hodgkin's lymphoma predicts adverse outcome in older adult patients. *J Clin Oncol* **27**, 3815-3821 (2009).
- 62 Divaris, K. *et al.* Oral health and risk for head and neck squamous cell carcinoma: the Carolina Head and Neck Cancer Study. *Cancer Causes & Control* **21**, 567-575 (2010).
- 63 Enblad, G., Sandvej, K., Sundstrom, C., Pallesen, G. & Glimelius, B. Epstein-Barr virus distribution in Hodgkin's disease in an unselected Swedish population. *Acta Oncologica* **38**, 425-429 (1999).
- 64 Fedder, M. & Gonzalez, M. F. Nasopharyngeal carcinoma. Brief review. *The American journal of medicine* **79**, 365-369 (1985).
- 65 Galetsky, S. A. *et al.* Epstein-Barr-virus-associated gastric cancer in Russia. *International Journal of Cancer* **73**, 786-789 (1997).
- 66 Gulley, M. L., Pulitzer, D. R., Eagan, P. A. & Schneider, B. G. Epstein-Barr virus infection is an early event in gastric carcinogenesis and is independent of bcl-2 expression and p53 accumulation. *Human pathology* **27**, 20-27 (1996).
- 67 Hao, Z. *et al.* The Epstein-Barr virus-associated gastric carcinoma in Southern and Northern China. *Oncology reports* **9**, 1293-1298 (2002).
- 68 Harn, H.-J. *et al.* Epstein-Barr virus-associated gastric adenocarcinoma in Taiwan. *Human pathology* **26**, 267-271 (1995).

- 69 Herrera-Goepfert, R. *et al.* Epstein-Barr virus-associated gastric carcinoma: Evidence of age-dependence among a Mexican population. *World journal of gastroenterology* **11**, 6096 (2005).
- 70 Hjalgrim, H., Friberg, J. & Melbye, M. in *Human Herpesviruses: Biology, Therapy, and Immunoprophylaxis* (eds Ann Arvin *et al.*) Ch. 53, (Cambridge University Press, 2007).
- 71 Hsu, J. L. & Glaser, S. L. Epstein-Barr virus-associated malignancies: epidemiologic patterns and etiologic implications. *Critical reviews in oncology/hematology* **34**, 27-53 (2000).
- 72 Iscovich, J., Boffetta, P., Franceschi, S., Azizi, E. & Sarid, R. Classic Kaposi sarcoma: epidemiology and risk factors. *Cancer* **88**, 500-517 (2000).
- 73 Jarrett, R. *et al.* The Scotland and Newcastle epidemiological study of Hodgkin's disease: impact of histopathological review and EBV status on incidence estimates. *Journal of clinical pathology* **56**, 811-816 (2003).
- 74 Johansson, S. L. & Cohen, S. M. Epidemiology and etiology of bladder cancer. *Seminars in surgical oncology* **13**, 291-298 (1997).
- 75 Kang, G. H. *et al.* Epstein-barr virus-positive gastric carcinoma demonstrates frequent aberrant methylation of multiple genes and constitutes CpG island methylator phenotype-positive gastric carcinoma. *The American journal of pathology* **160**, 787-794 (2002).
- 76 Karim, N. & Pallesen, G. Epstein-Barr virus (EBV) and gastric carcinoma in Malaysian patients. *The Malaysian journal of pathology* **25**, 45-47 (2003).
- 77 Kassem, A. *et al.* Frequent detection of Merkel cell polyomavirus in human Merkel cell carcinomas and identification of a unique deletion in the VP1 gene. *Cancer research* **68**, 5009-5013 (2008).
- 78 Keegan, T. H. *et al.* Epstein-Barr virus as a marker of survival after Hodgkin's lymphoma: a population-based study. *Journal of Clinical Oncology* **23**, 7604-7613 (2005).
- 79 Koriyama, C. *et al.* Epstein-Barr virus-associated gastric carcinoma in Japanese Brazilians and non-Japanese Brazilians in Sao Paulo. *Japanese journal of cancer research* **92**, 911-917 (2001).
- 80 Kume, T. *et al.* Low rate of apoptosis and overexpression of bcl-2 in Epstein-Barr virus-associated gastric carcinoma. *Histopathology* **34**, 502-509 (1999).
- 81 Lopes, L. *et al.* Epstein-Barr virus infection and gastric carcinoma in São Paulo State, Brazil. *Brazilian Journal of Medical and Biological Research* **37**, 1707-1712 (2004).
- 82 Lu, S. N. *et al.* Secular trends and geographic variations of hepatitis B virus and hepatitis C virus-associated hepatocellular carcinoma in Taiwan. *International journal of cancer* **119**, 1946-1952 (2006).
- 83 Mbulaiteye, S. M. *et al.* Trimodal age-specific incidence patterns for Burkitt lymphoma in the United States, 1973-2005. *International journal of cancer* **126**, 1732-1739 (2010).
- 84 McGlynn, K. A. & London, W. T. Epidemiology and natural history of hepatocellular carcinoma. *Best practice & research Clinical gastroenterology* **19**, 3-23 (2005).
- 85 McNeil, D. E., Coté, T. R., Clegg, L. & Mauer, A. SEER update of incidence and trends in pediatric malignancies: acute lymphoblastic leukemia. *Medical and pediatric oncology* **39**, 554-557 (2002).
- 86 Mimi, C. Y. & Yuan, J.-M. Epidemiology of nasopharyngeal carcinoma. *Seminars in cancer biology* **12**, 421-429 (2002).
- 87 Molica, S. Sex differences in incidence and outcome of chronic lymphocytic leukemia patients. *Leukemia & lymphoma* **47**, 1477-1480 (2006).

- 88 Moritani, S., Kushima, R., Sugihara, H. & Hattori, T. Phenotypic characteristics of Epstein-Barr-virus-associated gastric carcinomas. *Journal of cancer research and clinical oncology* **122**, 750-756 (1996).
- 89 Murphy, G., Pfeiffer, R., Camargo, M. C. & Rabkin, C. S. Meta-analysis shows that prevalence of Epstein-Barr virus-positive gastric cancer differs based on sex and anatomic location. *Gastroenterology* **137**, 824-833 (2009).
- 90 Nogueira, C. *et al.* Prevalence and characteristics of Epstein-Barr virus-associated gastric carcinomas in Portugal. *Infectious agents and cancer* **12**, 1-8 (2017).
- 91 Ogwang, M. D., Bhatia, K., Biggar, R. J. & Mbulaiteye, S. M. Incidence and geographic distribution of endemic Burkitt lymphoma in northern Uganda revisited. *International journal of cancer* **123**, 2658-2663 (2008).
- 92 Ojima, H., Fukuda, T., Nakajima, T., Takenoshita, S. & Nagamachi, Y. Discrepancy between clinical and pathological lymph node evaluation in Epstein-Barr virus-associated gastric cancers. *Anticancer research* **16**, 3081-3084 (1996).
- 93 Pallagani, L. *et al.* Epidemiology and clinicopathological profile of renal cell carcinoma: a review from tertiary care referral centre. *Journal of Kidney Cancer and VHL* **8**, 1 (2021).
- 94 Qiu, K. *et al.* Epstein-Barr virus in gastric carcinoma in Suzhou, China and Osaka, Japan: Association with clinico-pathologic factors and HLA-subtype. *International journal of cancer* **71**, 155-158 (1997).
- 95 Ragin, C., Modugno, F. & Gollin, S. The epidemiology and risk factors of head and neck cancer: a focus on human papillomavirus. *Journal of dental research* **86**, 104-114 (2007).
- 96 Rahbari, R., Zhang, L. & Kebebew, E. Thyroid cancer gender disparity. *Future Oncology* **6**, 1771-1779 (2010).
- 97 Randi, G., Franceschi, S. & La Vecchia, C. Gallbladder cancer worldwide: geographical distribution and risk factors. *International journal of cancer* **118**, 1591-1602 (2006).
- 98 Rawla, P. & Barsouk, A. Epidemiology of gastric cancer: global trends, risk factors and prevention. *Gastroenterology Review/Przegląd Gastroenterologiczny* **14**, 26-38 (2019).
- 99 Rowlands, D. *et al.* Epstein-Barr virus and carcinomas: rare association of the virus with gastric adenocarcinomas. *British journal of cancer* **68**, 1014-1019 (1993).
- 100 Sakuma, K. *et al.* Cancer risk to the gastric corpus in Japanese, its correlation with interleukin-1 β gene polymorphism (+ 3953* T) and Epstein-Barr virus infection. *International journal of cancer* **115**, 93-97 (2005).
- 101 Sellam, F. *et al.* Delayed diagnosis of pancreatic cancer reported as more common in a population of North African young adults. *Journal of gastrointestinal oncology* **6**, 505 (2015).
- 102 Shibata, D. & Weiss, L. Epstein-Barr virus-associated gastric adenocarcinoma. *The American journal of pathology* **140**, 769 (1992).
- 103 Shin, W. S. *et al.* Epstein-Barr virus-associated gastric adenocarcinomas among Koreans. *American journal of clinical pathology* **105**, 174-181 (1996).
- 104 Souza, E. M. *et al.* Impact of Epstein-Barr virus in the clinical evolution of patients with classical Hodgkin's lymphoma in Brazil. *Hematological Oncology* **28**, 137-141 (2010).
- 105 Takano, Y. *et al.* The role of the Epstein-Barr virus in the oncogenesis of EBV (+) gastric carcinomas. *Virchows Archiv* **434**, 17-22 (1999).
- 106 Tamási, L. *et al.* Age and Gender Specific Lung Cancer Incidence and Mortality in Hungary: Trends from 2011 Through 2016. *Pathology and Oncology Research*, 88 (2021).

- 107 Tamimi, A. F. & Juweid, M. Epidemiology and outcome of glioblastoma. *Exon Publications*, 143-153 (2017).
- 108 Tavakoli, A. *et al.* Association between Epstein-Barr virus infection and gastric cancer: a systematic review and meta-analysis. *BMC cancer* **20**, 1-14 (2020).
- 109 Tokunaga, M. *et al.* Epstein-Barr virus in gastric carcinoma. *The American journal of pathology* **143**, 1250 (1993).
- 110 van Beek, J. *et al.* EBV-positive gastric adenocarcinomas: a distinct clinicopathologic entity with a low frequency of lymph node involvement. *Journal of Clinical Oncology* **22**, 664-670 (2004).
- 111 Venook, A. P., Papandreou, C., Furuse, J. & Ladrón de Guevara, L. The incidence and epidemiology of hepatocellular carcinoma: a global and regional perspective. *The oncologist* **15**, 5-13 (2010).
- 112 Villano, J., Koshy, M., Shaikh, H., Dolecek, T. & McCarthy, B. Age, gender, and racial differences in incidence and survival in primary CNS lymphoma. *British journal of cancer* **105**, 1414-1418 (2011).
- 113 Wang, X. m. *et al.* Clinical analysis of 1629 newly diagnosed malignant lymphomas in current residents of Sichuan province, China. *Hematological oncology* **34**, 193-199 (2016).
- 114 Wang, Y. *et al.* Quantitative methylation analysis reveals gender and age differences in p16 INK 4a hypermethylation in hepatitis B virus-related hepatocellular carcinoma. *Liver International* **32**, 420-428 (2012).
- 115 Wei, K.-R. *et al.* Nasopharyngeal carcinoma incidence and mortality in China in 2010. *Chinese journal of cancer* **33**, 381 (2014).
- 116 Wu, M. S. *et al.* Epstein–Barr virus—associated gastric carcinomas: relation to H. pylori infection and genetic alterations. *Gastroenterology* **118**, 1031-1038 (2000).
- 117 Wu, S., Han, J., Li, W.-Q., Li, T. & Qureshi, A. A. Basal-cell carcinoma incidence and associated risk factors in US women and men. *American journal of epidemiology* **178**, 890-897 (2013).
- 118 Yanai, H. *et al.* Endoscopic and pathologic features of Epstein-Barr virus–associated gastric carcinoma. *Gastrointestinal endoscopy* **45**, 236-242 (1997).
- 119 Yoshiwara, E. *et al.* Epstein-Barr virus-associated gastric carcinoma in Lima, Peru. *J Exp Clin Cancer Res* **24**, 49-54 (2005).
- 120 Zhou, L. *et al.* Global, regional, and national burden of Hodgkin lymphoma from 1990 to 2017: estimates from the 2017 Global Burden of Disease study. *Journal of hematology & oncology* **12**, 1-13 (2019).
- 121 Zhu, Z.-Z. *et al.* Sex-related differences in DNA copy number alterations in hepatitis B virus-associated hepatocellular carcinoma. *Asian Pacific Journal of Cancer Prevention* **13**, 225-229 (2012).
- 122 Murphy, G., Pfeiffer, R., Camargo, M. C. & Rabkin, C. S. Meta-analysis shows that prevalence of Epstein-Barr virus-positive gastric cancer differs based on sex and anatomic location. *Gastroenterology* **137**, 824-833 (2009).
<https://doi.org/10.1053/j.gastro.2009.05.001>
- 123 Lee, J. H., Kim, Y., Choi, J. W. & Kim, Y. S. Prevalence and prognostic significance of Epstein-Barr virus infection in classical Hodgkin's lymphoma: a meta-analysis. *Arch Med Res* **45**, 417-431 (2014). <https://doi.org/10.1016/j.arcmed.2014.06.001>

- 124 Dozzo, M. *et al.* Burkitt lymphoma in adolescents and young adults: management challenges. *Adolescent Health, Medicine and Therapeutics* **Volume 8**, 11-29 (2016). <https://doi.org:10.2147/ahmt.s94170>
- 125 Schrama, D. *et al.* Merkel cell polyomavirus status is not associated with clinical course of Merkel cell carcinoma. *J Invest Dermatol* **131**, 1631-1638 (2011). <https://doi.org:10.1038/jid.2011.115>
- 126 Rismiller, K. & Knackstedt, T. J. Aggressive Digital Papillary Adenocarcinoma: Population-Based Analysis of Incidence, Demographics, Treatment, and Outcomes. *Dermatol Surg* **44**, 911-917 (2018). <https://doi.org:10.1097/DSS.0000000000001483>
- 127 Harms, P. W. *et al.* The Distinctive Mutational Spectra of Polyomavirus-Negative Merkel Cell Carcinoma. *Cancer Res* **75**, 3720-3727 (2015). <https://doi.org:10.1158/0008-5472.CAN-15-0702>
- 128 Tiacci, E. *et al.* Pervasive mutations of JAK-STAT pathway genes in classical Hodgkin lymphoma. *Blood* **131**, 2454-2465 (2018).
- 129 Kaulen, L. D. *et al.* Integrated genetic analyses of immunodeficiency-associated Epstein-Barr virus- (EBV) positive primary CNS lymphomas. *Acta Neuropathol* **146**, 499-514 (2023). <https://doi.org:10.1007/s00401-023-02613-w>
- 130 Ghasemi, F. *et al.* Mutational analysis of head and neck squamous cell carcinoma stratified by smoking status. *JCI Insight* **4** (2019). <https://doi.org:10.1172/jci.insight.123443>
- 131 Panda, A. *et al.* Immune Activation and Benefit From Avelumab in EBV-Positive Gastric Cancer. *J Natl Cancer Inst* **110**, 316-320 (2018). <https://doi.org:10.1093/jnci/djx213>
- 132 Zhao, X., Fan, X., Lin, X., Guo, B. & Yu, Y. Deciphering age-specific molecular features in cervical cancer and constructing an angio-immune prognostic model. *Medicine (Baltimore)* **103**, e37717 (2024). <https://doi.org:10.1097/MD.00000000000037717>
- 133 Li, T. *et al.* TIMER2.0 for analysis of tumor-infiltrating immune cells. *Nucleic Acids Res* **48**, W509-W514 (2020). <https://doi.org:10.1093/nar/gkaa407>
- 134 Newman, A. M. *et al.* Robust enumeration of cell subsets from tissue expression profiles. *Nat Methods* **12**, 453-457 (2015). <https://doi.org:10.1038/nmeth.3337>
- 135 Li, B. *et al.* Comprehensive analyses of tumor immunity: implications for cancer immunotherapy. *Genome Biol* **17**, 174 (2016). <https://doi.org:10.1186/s13059-016-1028-7>
- 136 Finotello, F. *et al.* Molecular and pharmacological modulators of the tumor immune contexture revealed by deconvolution of RNA-seq data. *Genome Med* **11**, 34 (2019). <https://doi.org:10.1186/s13073-019-0638-6>
- 137 Becht, E. *et al.* Estimating the population abundance of tissue-infiltrating immune and stromal cell populations using gene expression. *Genome Biol* **17**, 218 (2016). <https://doi.org:10.1186/s13059-016-1070-5>
- 138 Aran, D., Hu, Z. & Butte, A. J. xCell: digitally portraying the tissue cellular heterogeneity landscape. *Genome Biol* **18**, 220 (2017). <https://doi.org:10.1186/s13059-017-1349-1>
- 139 Song, L. *et al.* TRUST4: immune repertoire reconstruction from bulk and single-cell RNA-seq data. *Nat Methods* **18**, 627-630 (2021). <https://doi.org:10.1038/s41592-021-01142-2>
- 140 Song, L. *et al.* Comprehensive Characterizations of Immune Receptor Repertoire in Tumors and Cancer Immunotherapy Studies. *Cancer Immunol Res* **10**, 788-799 (2022). <https://doi.org:10.1158/2326-6066.CIR-21-0965>
- 141 Özay, Z. İ., Sütcüoğlu, O., Özdemir, N. & Yazıcı, O. Review of Immunotherapy Efficacy in Virus-associated Cancers. *EJMO* **6** (2022). <https://doi.org:10.14744/ejmo.2022.86807>

- 142 Miliotis, C. N. & Slack, F. J. Multi-layered control of PD-L1 expression in Epstein-Barr virus-associated gastric cancer. *J Cancer Metastasis Treat* **6** (2020). <https://doi.org:10.20517/2394-4722.2020.12>
- 143 Ukpo, O. C., Thorstad, W. L. & Lewis, J. S., Jr. B7-H1 expression model for immune evasion in human papillomavirus-related oropharyngeal squamous cell carcinoma. *Head Neck Pathol* **7**, 113-121 (2013). <https://doi.org:10.1007/s12105-012-0406-z>
- 144 Lyford-Pike, S. *et al.* Evidence for a role of the PD-1:PD-L1 pathway in immune resistance of HPV-associated head and neck squamous cell carcinoma. *Cancer Res* **73**, 1733-1741 (2013). <https://doi.org:10.1158/0008-5472.CAN-12-2384>
- 145 Kim, H. S. *et al.* Association Between PD-L1 and HPV Status and the Prognostic Value of PD-L1 in Oropharyngeal Squamous Cell Carcinoma. *Cancer Res Treat* **48**, 527-536 (2016). <https://doi.org:10.4143/crt.2015.249>
- 146 Badoual, C. *et al.* PD-1-expressing tumor-infiltrating T cells are a favorable prognostic biomarker in HPV-associated head and neck cancer. *Cancer Res* **73**, 128-138 (2013). <https://doi.org:10.1158/0008-5472.CAN-12-2606>
- 147 Stransky, N. *et al.* The mutational landscape of head and neck squamous cell carcinoma. *Science* **333**, 1157-1160 (2011). <https://doi.org:10.1126/science.1208130>
- 148 López, C. *et al.* Genomic and transcriptomic changes complement each other in the pathogenesis of sporadic Burkitt lymphoma. *Nature Communications* **10** (2019). <https://doi.org:10.1038/s41467-019-08578-3>
- 149 Abate, F. *et al.* Distinct Viral and Mutational Spectrum of Endemic Burkitt Lymphoma. *PLOS Pathogens* **11**, e1005158 (2015). <https://doi.org:10.1371/journal.ppat.1005158>
- 150 Zhou, P. X. *et al.* Sporadic and endemic Burkitt lymphoma have frequent FOXO1 mutations but distinct hotspots in the AKT recognition motif. *Blood Advances* **3**, 2118-2127 (2019). <https://doi.org:10.1182/bloodadvances.2018029546>
- 151 Kaymaz, Y. *et al.* Comprehensive Transcriptome and Mutational Profiling of Endemic Burkitt Lymphoma Reveals EBV Type-Specific Differences. *Molecular Cancer Research* **15**, 563-576 (2017). <https://doi.org:10.1158/1541-7786.mcr-16-0305-t>
- 152 Gong, C. *et al.* Sequential inverse dysregulation of the RNA helicases DDX3X and DDX3Y facilitates MYC-driven lymphomagenesis. *Mol Cell* **81**, 4059-4075 e4011 (2021). <https://doi.org:10.1016/j.molcel.2021.07.041>
- 153 Shechter, O., Sausen, D. G., Gallo, E. S., Dahari, H. & Borenstein, R. Epstein-Barr Virus (EBV) Epithelial Associated Malignancies: Exploring Pathologies and Current Treatments. *Int J Mol Sci* **23** (2022). <https://doi.org:10.3390/ijms232214389>
- 154 Yang, J., Liu, Z., Zeng, B., Hu, G. & Gan, R. Epstein-Barr virus-associated gastric cancer: A distinct subtype. *Cancer Lett* **495**, 191-199 (2020). <https://doi.org:10.1016/j.canlet.2020.09.019>
- 155 de Martel, C., Georges, D., Bray, F., Ferlay, J. & Clifford, G. M. Global burden of cancer attributable to infections in 2018: a worldwide incidence analysis. *Lancet Glob Health* **8**, e180-e190 (2020). [https://doi.org:10.1016/S2214-109X\(19\)30488-7](https://doi.org:10.1016/S2214-109X(19)30488-7)
- 156 Jarrett, A., Armstrong, A. & Alexander, E. Epidemiology of EBV and Hodgkin's lymphoma. *Annals of oncology* **7**, S5-S10 (1996).
- 157 Vockerodt, M., Cader, F. Z., Shannon-Lowe, C. & Murray, P. Epstein-Barr virus and the origin of Hodgkin lymphoma. *Chin J Cancer* **33**, 591-597 (2014). <https://doi.org:10.5732/cjc.014.10193>

- 158 Massini, G., Siemer, D. & Hohaus, S. EBV in Hodgkin Lymphoma. *Mediterr J Hematol Infect Dis* **1**, e2009013 (2009). <https://doi.org/10.4084/MJHID.2009.013>
- 159 Sartor, M. A. *et al.* Genome-wide methylation and expression differences in HPV(+) and HPV(-) squamous cell carcinoma cell lines are consistent with divergent mechanisms of carcinogenesis. *Epigenetics* **6**, 777-787 (2011). <https://doi.org/10.4161/epi.6.6.16216>
- 160 van Kempen, P. M. *et al.* Differences in methylation profiles between HPV-positive and HPV-negative oropharynx squamous cell carcinoma: a systematic review. *Epigenetics* **9**, 194-203 (2014).
- 161 Ekanayake Weeramange, C. *et al.* DNA Methylation Changes in Human Papillomavirus-Driven Head and Neck Cancers. *Cells* **9** (2020). <https://doi.org/10.3390/cells9061359>
- 162 Nakagawa, T. *et al.* DNA Methylation and HPV-Associated Head and Neck Cancer. *Microorganisms* **9** (2021). <https://doi.org/10.3390/microorganisms9040801>
- 163 Gadek, M., Sherr, E. H. & Floor, S. N. The variant landscape and function of DDX3X in cancer and neurodevelopmental disorders. *Trends in Molecular Medicine* **29**, 726-739 (2023). <https://doi.org/10.1016/j.molmed.2023.06.003>
- 164 Ryan, C. S. & Schröder, M. The human DEAD-box helicase DDX3X as a regulator of mRNA translation. *Frontiers in Cell and Developmental Biology* **10** (2022). <https://doi.org/10.3389/fcell.2022.1033684>
- 165 Lacroix, M. *et al.* The X-Linked Helicase DDX3X Is Required for Lymphoid Differentiation and MYC-Driven Lymphomagenesis. *Cancer Research* **82**, 3172-3186 (2022). <https://doi.org/10.1158/0008-5472.can-21-2454>
- 166 Soulat, D. *et al.* The DEAD-box helicase DDX3X is a critical component of the TANK-binding kinase 1-dependent innate immune response. *The EMBO Journal* **27**, 2135-2146 (2008). <https://doi.org/10.1038/emboj.2008.126>
- 167 Jiang, L. *et al.* Exome sequencing identifies somatic mutations of
in natural killer/T-cell lymphoma. *Nature Genetics* **47**, 1061-+ (2015).
<https://doi.org/10.1038/ng.3358>
- 168 Liu, X. *et al.* Immunotherapy for recurrent or metastatic nasopharyngeal carcinoma. *NPJ Precis Oncol* **8**, 101 (2024). <https://doi.org/10.1038/s41698-024-00601-1>
- 169 Cao, S. *et al.* Dynamic host immune response in virus-associated cancers. *Commun Biol* **2**, 109 (2019). <https://doi.org/10.1038/s42003-019-0352-3>
- 170 The Cancer Genome Atlas Network. Comprehensive genomic characterization of head and neck squamous cell carcinomas. *Nature* **517**, 576-582 (2015). <https://doi.org/10.1038/nature14129>

Point-by-Point Response to Reviewer Comments

Reviewer #1

Comment) *“The authors performed an in-depth revision of their manuscript and satisfactorily answered to all my comments.*

They considerably improved the clarity of the manuscript by providing many additional details and improving the clarity of the figures.

They added new data (PCAWG data sets) and analyses (structural variant and copy-number signatures). The manuscript now clearly states which findings are novel and how the results compare with previous work.

I find the revised manuscript very convincing and I now support its acceptance for publication in Nature Communications.”

Response) We thank the reviewer for their time and input, which have helped to improve our manuscript.

Reviewer #2

Comment 1) *“The authors have addressed most of my previous comments. I have two minor comments for further improvement:*

1. Clarification of Novel Findings:

The revised manuscript has cited more prior works and highlights four key findings: higher male incidence, reduced mutation burden, frequent RNA helicase mutations, and an enhanced immunotherapy response in virus-associated cancers. It would be beneficial to clarify in the Discussion which of these findings confirm previous reports and which represent novel observations. Clearly distinguishing these points will help readers understand the significance of the findings and identify areas for further study.”

Response 1) We thank the reviewer for this helpful suggestion. At the start of the revised Discussion, we added the following paragraph to summarize our main findings and relate them to previous studies. Because Nature Communications advises against using words such as “new,” “novel,” and “first,” it may not be entirely clear which observations build upon prior work; nonetheless, we have made every effort to highlight these distinctions:

“This study builds upon and extends prior research while providing insights into the epidemiological, somatic, and immune components commonly implicated in the pathogenesis of oncovirus-associated cancers. The observed higher male incidence of virus-associated cancers confirms earlier epidemiological findings in certain malignancies, such as EBV-positive HL and GC¹⁻³. Likewise, the finding of a reduced mutation burden in virus-positive tumors relative to virus-negative counterparts is consistent with previous reports in individual tumor types⁴⁻⁹, including HNSCC⁷ and PCNSL⁶. We also identified frequent RNA helicase mutations (DDX3X and EIF4A1) in virus-positive tumors spanning multiple cancer types; although DDX3X mutation have been reported in HNSCC¹⁰, our results demonstrate their broader relevance across a pan-cancer cohort. Moreover, by aggregating data from 252 BL cases, we found a significant association between DDX3X mutations and EBV-positive BL, an association that had been noted

previously but not shown to be statistically significant¹¹⁻¹⁶. Finally, our analysis of immunotherapy response suggests that virus-positive status is associated with enhanced response to checkpoint blockade in gastric cancer and head and neck squamous cell carcinoma.”

Comment 2) *“2. Specification of Data Sources:*

Although the paper cites 23 studies involved in this analysis and the supplementary table provides links to the corresponding publications, it is not clear where the actual datasets used in this analysis were obtained. For enhanced transparency and reproducibility, please specify the direct links or identifiers for the datasets employed in this study.”

Response 2) We appreciate the reviewer’s suggestion. The 23 studies listed in Supplementary Table S1 are those used to create Figure 1A, which compares the incidence rates of males and females in virus-associated and non-virus-associated cancer types. The male and female counts in Supplementary Table S1 were derived directly from the respective published articles and/or their supplementary materials. We have updated the Data Availability section of our manuscript to provide further details on the additional datasets used in other parts of the analysis.

Updated Data Availability Section) *“The DNA sequencing data of Kaposi sarcoma generated in this study have been deposited in the European Nucleotide Archive (ENA) at EMBL-EBI database under accession code PRJEB76508 [<https://www.ebi.ac.uk/ena/browser/view/PRJEB76508>]. The TCGA (HNSCC, CC and GC) cohort’s clinical data, mutation and copy number alteration calls are available at cBioportal (<https://www.cbioportal.org/>), raw and normalized chromosomal instability (CIN) signature activities are available within the supplementary information of Drews et al.’s publication¹⁷. SV and CN data of PCAWG WGS cases are available from The International Cancer Genome Consortium Accelerating Research in Genomic Oncology (ICGC ARGO) data platform (Legacy ICGC 25K Data, <https://platform.icgc-argo.org/>). Clinical and genomic data including SNV, CN, and/or SV calls of other cancer types are available within the supplementary information of their published studies^{11,1,5,8,18-30}. EBV status of the Hodgkin lymphoma cohort from Alig et al.²⁹ has been acquired from the authors directly. Source data are provided with this paper.”*

Reviewer #3

Comment) *“The authors have comprehensively addressed my comments and have revised their manuscript accordingly.”*

Response) We thank the reviewer for the thoughtful review.

References

- 1 The Cancer Genome Atlas Research Network. Comprehensive molecular characterization of gastric adenocarcinoma. *Nature* **513**, 202-209 (2014). <https://doi.org/10.1038/nature13480>
- 2 Murphy, G., Pfeiffer, R., Camargo, M. C. & Rabkin, C. S. Meta-analysis shows that prevalence of Epstein-Barr virus-positive gastric cancer differs based on sex and anatomic location. *Gastroenterology* **137**, 824-833 (2009). <https://doi.org/10.1053/j.gastro.2009.05.001>
- 3 Lee, J. H., Kim, Y., Choi, J. W. & Kim, Y. S. Prevalence and prognostic significance of Epstein-Barr virus infection in classical Hodgkin's lymphoma: a meta-analysis. *Arch Med Res* **45**, 417-431 (2014). <https://doi.org/10.1016/j.arcmed.2014.06.001>
- 4 Harms, P. W. *et al.* The Distinctive Mutational Spectra of Polyomavirus-Negative Merkel Cell Carcinoma. *Cancer Res* **75**, 3720-3727 (2015). <https://doi.org/10.1158/0008-5472.CAN-15-0702>
- 5 Tiacci, E. *et al.* Pervasive mutations of JAK-STAT pathway genes in classical Hodgkin lymphoma. *Blood* **131**, 2454-2465 (2018).
- 6 Kaulen, L. D. *et al.* Integrated genetic analyses of immunodeficiency-associated Epstein-Barr virus- (EBV) positive primary CNS lymphomas. *Acta Neuropathol* **146**, 499-514 (2023). <https://doi.org/10.1007/s00401-023-02613-w>
- 7 Ghasemi, F. *et al.* Mutational analysis of head and neck squamous cell carcinoma stratified by smoking status. *JCI Insight* **4** (2019). <https://doi.org/10.1172/jci.insight.123443>
- 8 Ramis-Zaldivar, J. E. *et al.* MAPK and JAK-STAT pathways dysregulation in plasmablastic lymphoma. *Haematologica* **106**, 2682-2693 (2021). <https://doi.org/10.3324/haematol.2020.271957>
- 9 Panda, A. *et al.* Immune Activation and Benefit From Avelumab in EBV-Positive Gastric Cancer. *J Natl Cancer Inst* **110**, 316-320 (2018). <https://doi.org/10.1093/jnci/djx213>
- 10 Seiwert, T. Y. *et al.* Integrative and comparative genomic analysis of HPV-positive and HPV-negative head and neck squamous cell carcinomas. *Clin Cancer Res* **21**, 632-641 (2015). <https://doi.org/10.1158/1078-0432.CCR-13-3310>
- 11 Grande, B. M. *et al.* Genome-wide discovery of somatic coding and noncoding mutations in pediatric endemic and sporadic Burkitt lymphoma. *Blood* **133**, 1313-1324 (2019). <https://doi.org/10.1182/blood-2018-09-871418>
- 12 López, C. *et al.* Genomic and transcriptomic changes complement each other in the pathogenesis of sporadic Burkitt lymphoma. *Nature Communications* **10** (2019). <https://doi.org/10.1038/s41467-019-08578-3>
- 13 Abate, F. *et al.* Distinct Viral and Mutational Spectrum of Endemic Burkitt Lymphoma. *PLOS Pathogens* **11**, e1005158 (2015). <https://doi.org/10.1371/journal.ppat.1005158>
- 14 Zhou, P. X. *et al.* Sporadic and endemic Burkitt lymphoma have frequent FOXO1 mutations but distinct hotspots in the AKT recognition motif. *Blood Advances* **3**, 2118-2127 (2019). <https://doi.org/10.1182/bloodadvances.2018029546>
- 15 Kaymaz, Y. *et al.* Comprehensive Transcriptome and Mutational Profiling of Endemic Burkitt Lymphoma Reveals EBV Type-Specific Differences. *Molecular Cancer Research* **15**, 563-576 (2017). <https://doi.org/10.1158/1541-7786.mcr-16-0305-t>
- 16 Gong, C. *et al.* Sequential inverse dysregulation of the RNA helicases DDX3X and DDX3Y facilitates MYC-driven lymphomagenesis. *Mol Cell* **81**, 4059-4075 e4011 (2021). <https://doi.org/10.1016/j.molcel.2021.07.041>

- 17 Drews, R. M. *et al.* A pan-cancer compendium of chromosomal instability. *Nature* **606**, 976-983 (2022). <https://doi.org:10.1038/s41586-022-04789-9>
- 18 Wienand, K. *et al.* Genomic analyses of flow-sorted Hodgkin Reed-Sternberg cells reveal complementary mechanisms of immune evasion. *Blood Adv* **3**, 4065-4080 (2019). <https://doi.org:10.1182/bloodadvances.2019001012>
- 19 Grande, B. M. *et al.* Genome-wide discovery of somatic coding and noncoding mutations in pediatric endemic and sporadic Burkitt lymphoma. *Blood* **133**, 1313-1324 (2019).
- 20 Starrett, G. J. *et al.* Clinical and molecular characterization of virus-positive and virus-negative Merkel cell carcinoma. *Genome Med* **12**, 30 (2020). <https://doi.org:10.1186/s13073-020-00727-4>
- 21 The Cancer Genome Atlas Network. Comprehensive genomic characterization of head and neck squamous cell carcinomas. *Nature* **517**, 576-582 (2015). <https://doi.org:10.1038/nature14129>
- 22 The Cancer Genome Atlas Research Network. Integrated genomic and molecular characterization of cervical cancer. *Nature* **543**, 378-384 (2017). <https://doi.org:10.1038/nature21386>
- 23 Zhang, L. *et al.* Genomic Analysis of Nasopharyngeal Carcinoma Reveals TME-Based Subtypes. *Mol Cancer Res* **15**, 1722-1732 (2017). <https://doi.org:10.1158/1541-7786.MCR-17-0134>
- 24 Xiong, J. *et al.* Genomic and Transcriptomic Characterization of Natural Killer T Cell Lymphoma. *Cancer Cell* **37**, 403-419 e406 (2020). <https://doi.org:10.1016/j.ccell.2020.02.005>
- 25 Liu, Z. *et al.* Genomic characterization of HIV-associated plasmablastic lymphoma identifies pervasive mutations in the JAK-STAT pathway. *Blood Cancer Discov* **1**, 112-125 (2020). <https://doi.org:10.1158/2643-3230.BCD-20-0051>
- 26 Gandhi, M. K. *et al.* EBV-associated primary CNS lymphoma occurring after immunosuppression is a distinct immunobiological entity. *Blood* **137**, 1468-1477 (2021). <https://doi.org:10.1182/blood.2020008520>
- 27 Kataoka, K. *et al.* Integrated molecular analysis of adult T cell leukemia/lymphoma. *Nat Genet* **47**, 1304-1315 (2015). <https://doi.org:10.1038/ng.3415>
- 28 Dufva, O. *et al.* Aggressive natural killer-cell leukemia mutational landscape and drug profiling highlight JAK-STAT signaling as therapeutic target. *Nat Commun* **9**, 1567 (2018). <https://doi.org:10.1038/s41467-018-03987-2>
- 29 Alig, S. K. *et al.* Distinct Hodgkin lymphoma subtypes defined by noninvasive genomic profiling. *Nature* **625**, 778-787 (2024). <https://doi.org:10.1038/s41586-023-06903-x>
- 30 Maura, F. *et al.* Molecular Evolution of Classic Hodgkin Lymphoma Revealed Through Whole-Genome Sequencing of Hodgkin and Reed Sternberg Cells. *Blood Cancer Discov* **4**, 208-227 (2023). <https://doi.org:10.1158/2643-3230.BCD-22-0128>